# Towards Efficient and Optimal Covariance-Adaptive Algorithms for Combinatorial Semi-Bandits

**Julien Zhou**

Criteo AI Lab
Paris, France

Univ. Grenoble Alpes, Inria,
CNRS, Grenoble INP, LJK,
38000 Grenoble, France

`julien.zhou@inria.fr`

**Pierre Gaillard**

Univ. Grenoble Alpes, Inria,
CNRS, Grenoble INP, LJK,
38000 Grenoble, France

**Thibaud Rahier**

Criteo AI Lab,
Paris, France

**Houssam Zenati**

Université Paris-Saclay, Inria,
Palaiseau, France

**Julyan Arbel**

Univ. Grenoble Alpes, Inria,
CNRS, Grenoble INP, LJK,
38000 Grenoble, France

## Abstract

We address the problem of stochastic combinatorial semi-bandits, where a player selects among $P$ actions from the power set of a set containing $d$ base items. Adaptivity to the problem's structure is essential in order to obtain optimal regret upper bounds. As estimating the coefficients of a covariance matrix can be manageable in practice, leveraging them should improve the regret. We design "optimistic" covariance-adaptive algorithms relying on online estimations of the covariance structure, called `OLS-UCB-C` and `COS-V` (only the variances for the latter). They both yield improved gap-free regret. Although `COS-V` can be slightly suboptimal, it improves on computational complexity by taking inspiration from Thompson Sampling approaches. It is the first sampling-based algorithm satisfying a $O(\sqrt{T})$ gap-free regret (up to poly-logs). We also show that in some cases, our approach efficiently leverages the semi-bandit feedback and outperforms bandit feedback approaches, not only in exponential regimes where $P \gg d$ but also when $P \leq d$, which is not covered by existing analyses.

## 1 Introduction

In sequential decision-making, the bandit framework has been extensively studied and was instrumental to several applications, e.g. A/B testing (Guo et al., 2020), online advertising and recommendation services (Zeng et al., 2016), network routing (Tabei et al., 2023), demand-side management (Brégère et al., 2019), etc. Its popularity stems from its relative simplicity, allowing it to model and analyze a wide range of challenging real-world settings. Reference books like Bubeck and Cesa-Bianchi (2012) or Lattimore and Szepesvári (2020) offer a wide perspective on the subject.

In this framework, a *decision-maker* or *player* must make choices and receives associated rewards, but it lacks prior knowledge of its environment. This naturally leads to an exploration-exploitation trade-off: the player must explore different actions to determine the best one, but an inefficient exploration strategy may harm the cumulative rewards. Efficient algorithms rely on exploiting the

38th Conference on Neural Information Processing Systems (NeurIPS 2024).

environment's structure, such as estimating the parameters of a reward function instead of exploring every action.

This paper focuses on the stochastic combinatorial semi-bandit framework. At each round, the player chooses a subset of *base items* and receives a feedback for each item chosen. The action set is included in the base items' power set, and can therefore be exponentially big and difficult to explore. The main challenge in this framework is to effectively combine the information collected through different actions (that may share common base items).

**Problem formulation.**    We consider a set of $d \in \mathbb{N}^*$ *base items*, each item $i \in [d] = \{1, \ldots, d\}$ yielding stochastic rewards. A *player* accesses these rewards through a set $\mathcal{A} \subseteq \{0, 1\}^d$ of $P \in \mathbb{N}^*$ *actions*, each corresponding to a subset of at most $m \geq 5$ items[a]. We refer to actions $a \in \mathcal{A}$ using their components vector $a = (a_i)_{i \in [d]} \in \{0, 1\}^d$ where for all $j \in [d]$, $a_j = 1$ if and only if action $a$ contains base item $j$.

The player interacts with an *environment* over a sequence of $T \in \mathbb{N}^*$ *rounds*. At each round $t \in [T]$, the player chooses an action $A_t \in \mathcal{A}$, the environment samples a reward vector $Y_t \in \mathbb{R}^d$, the player observes the realization for every item contained in $A_t$, and receives their sum. The interactions between the player and the environment are summarized in Framework 1.

---

**Framework 1** Stochastic Combinatorial Semi-Bandit

For each $t \in \{1, \ldots, T\}$:
- The player chooses an action $A_t \in \mathcal{A}$.
- The environment samples a vector of rewards $Y_t \in \mathbb{R}^d$ from a fixed unknown distribution.
- The player receives the reward $\langle A_t, Y_t \rangle = \sum_i A_{t,i} Y_{t,i}$.
- The player observes $Y_{t,i}$ for all $i \in [d]$ s.t. $A_{t,i} = 1$.

---

**Assumptions.**    We make the following assumptions. For all $t \in [T]$, $Y_t$ is independent of the past rewards and the player's decision $\sigma(A_1, Y_1, \ldots, A_{t-1}, Y_{t-1}, A_t)$. There exists a mean reward vector $\mathbb{E}[Y_t] = \mu \in \mathbb{R}^d$ and a second order moment matrix $\mathbf{S} = \mathbb{E}[Y_t Y_t^\top] \in M_d(\mathbb{R})$. The positive semi-definite covariance matrix is denoted $\mathbf{\Sigma} \in M_d(\mathbb{R})$, with $\mathbf{\Sigma} = \mathbf{S} - \mu\mu^\top$. There exists a known *vector* $B \in \mathbb{R}_+^d$ such that for all $t \in [T]$ and $i \in [d]$, $|Y_{t,i}| \leq B_i/2$ almost surely (and $|Y_{t,i} - \mu_i| \leq B_i$).

The objective of the decision-maker is to minimize the expected cumulative pseudo-regret defined as:

$$\mathbb{E}[R_T] = \mathbb{E}\left[\sum_{t=1}^T \langle a^* - A_t, \mu \rangle\right] = \sum_{t=1}^T \Delta_{A_t}, \tag{1}$$

where $\langle \cdot, \cdot \rangle$ denotes the usual inner product in $\mathbb{R}^d$, $a^* \in \arg\max_{a \in \mathcal{A}} \langle a, \mu \rangle$ is an optimal action, and $\Delta_a = \langle a^* - a, \mu \rangle$ is the *sub-optimality gap* for action $a \in \mathcal{A}$.

## 1.1    Existing work and limitations

Combinatorial semi-bandit problems have been extensively studied by the bandit community since their introduction by Chen et al. (2013). Here, we only highlight key earlier works related to this paper. For a comprehensive introduction to this literature, we refer the interested reader to the monograph by Lattimore and Szepesvári (2020).

A first line of works considers deterministic algorithms based on the optimistic principle and upper confidence bounds (UCBs). Chen et al. (2013) first designed CUCB, computing UCBs for the items' average rewards, converting these into UCBs for the actions' rewards, and choosing the action with the highest one. It was later analyzed by Kveton et al. (2015), who proved a regret upper bound uniform over all possible covariance matrices $\mathbf{\Sigma}$ (hence, paying the worst-case). Combes et al. (2015) highlighted the importance of designing $\mathbf{\Sigma}$-adaptive algorithms by showing that the regret could be improved by a factor of $m$ when the items' average rewards are independent. Subsequently, Degenne and Perchet (2016) developed OLS-UCB, an algorithm intended to leverage the covariance structure. However, OLS-UCB requires prior knowledge of a positive semi-definite covariance-proxy matrix $\mathbf{\Gamma}$, such that for all $t \geq 1$ and for all $u \in \mathbb{R}^d$, $\mathbb{E}[\exp(\langle u, Y_t - \mu \rangle)] \leq \exp(\frac{1}{2}\|u\|_{\mathbf{\Gamma}}^2)$. Estimating $\mathbf{\Gamma}$ in

---

[a]Throughout the paper, the term *item* (or *base item*) refers to an element in the set $[d]$, while an *action* denotes a subset of base items in $\mathcal{A}$.

practice is challenging and leads to regret bounds depending on it instead of the "true" covariance matrix $\Sigma$, potentially resulting in significantly looser bounds. This issue was addressed by Perrault et al. (2020b), who proposed a covariance-adaptive algorithm, ESCB-C, with asymptotically optimal gap-dependent regret upper bounds. Yet, it suffers from an additive constant of order $\Delta_{\min}^{-2}$, which prevents its conversion into an $\tilde{O}(\sqrt{T})$[b] gap-free bound. Thus, none of the above works proposes a $\tilde{O}(\sqrt{T})$ gap-free and covariance-adaptive regret bound, which is one of the key contributions of this paper. A common drawback of these works is also their potentially prohibitive computational complexity, due to the need to solve a maximization step over a large action set $\mathcal{A} \subseteq \{0,1\}^d$ that can be exponentially large. Some works, such as Cuvelier et al. (2021) or Liu et al. (2022), propose solutions to achieve polynomial time complexity, for example by applying UCB at the item level only rather than the action level. However, these approaches only work for independent rewards or under specific assumptions on their distribution, making the analysis for generic and unknown distributions extremely challenging. Another approach to tackle the computational burden in combinatorial semi-bandits is to resort to sampling algorithms, which we detail below.

A second line of works for stochastic combinatorial semi-bandits considers randomized algorithms inspired by Thompson Sampling (TS) for multi-armed bandits (Thompson, 1933). These algorithms involve sampling a random vector $\tilde{\mu}_t \in \mathbb{R}^d$ at each round $t + 1 \in [T - 1]$ from a distribution representing a "belief" over the parameter $\mu$, taking a decision $A_{t+1} \in \arg\max_{a \in \mathcal{A}} \langle a, \tilde{\mu}_t \rangle$, and updating the belief distribution using the observations. The main appeal of these approaches lies in their computational complexity, especially when solving a linear maximization problem in particular action spaces (such as matroids). Recent works have designed and analyzed such algorithms. Notably, Wang and Chen (2018) consider independent item's rewards. Perrault et al. (2020a) refine it and assume a known variance-proxy $\Gamma$ and therefore suffers from the same drawbacks as Degenne and Perchet (2016). Their technical analysis also yields a gap-dependent regret bound with an undesirable $\Delta_{\min}^{-m}$ term, preventing a $\tilde{O}(\sqrt{T})$ gap-free rate. A central contribution in our paper is the combination of the computational efficiency for sampling algorithms with the covariance-adaptivity $\tilde{O}(\sqrt{T})$ gap-free from our UCB approach.

Besides, the literature concerning our setting has historically mostly focused on cases where the action set is exponentially large, namely $P \gg d$, and the way to get quasi-optimal regret rates in these instances. However, outside of these regimes, the commonly derived regret bounds are too rough and fail to show the benefit of the semi-bandit feedback. While the conventional stochastic combinatorial semi-bandit regret upper bound grows as $\tilde{O}(\sqrt{mdT})$ (Kveton et al., 2015), a $\tilde{O}(\sqrt{mPT})$ could be achieved using bandit feedback only (Auer et al., 2002b). Intriguingly, the latter appears to outperform the semi-bandit rate as soon as $P < d$, making the extra information obtained through a richer feedback seemingly useless. Fine-grained analyses, clearly taking the structure into account, are therefore needed.

## 1.2 Contributions

**A new deterministic optimism-based algorithm (Section 2).** We present OLS-UCB-C (Online Least Squares Upper Confidence Bound with Covariance estimation), relying on the optimism principle. The analysis of OLS-UCB-C sketched in Section 5.2 shows the following properties:

- *First optimal gap-free regret upper bound.* OLS-UCB-C yields a similar gap-dependent regret bound as ESCB-C from Perrault et al. (2020b) up to logarithmic factors, and *the first optimal covariance-adaptative gap-free $\tilde{O}(\sqrt{T})$ regret bound* (Theorem 1).
- *Improved performance over UCB in all regimes of $P/d$.* Under some conditions on the covariance matrix $\Sigma$, we prove that OLS-UCB has a uniformly better regret than UCB, showing that properly leveraging semi-bandit feedback indeed consistently offers an advantage on (simple) bandit algorithms, which is not straightforward from existing analyses.
- *Improved complexity over concurrent algorithms.* OLS-UCB-C circumvents the convex optimization problem that ESCB-C requires to solve at each round and is therefore more efficient, despite suffering in the very large $P$ regime as many other deterministic algorithms.

**The first stochastic optimism-based algorithm (Section 3).** We introduce COS-V (Combinatorial Optimistic Sampling with Variance estimation), a TS-inspired algorithm exploiting the "frequentist" confidence regions derived in Section 4. It satisfies the following:

---

[b]We denote $\tilde{O}$ for $O$ when $T \to \infty$, up to poly-logarithmic terms.

- *Improved complexity for $P \gg 1$ compared to other deterministic semi-bandit algorithms.* COS-V can be efficient in the very large $P$ regime, which is the main blind spot of OLS-UCB-C.
- *First gap-free $\tilde{O}(\sqrt{T})$ regret upper bound for a sampling algorithm.* The analysis we provide in Section 5.3 exploits the common structure of the OLS-UCB-C and COS-V algorithms. It enables the derivation of a gap-dependent bound for COS-V that does not involve the $\Delta^{-m}$ term we typically find in the analysis for other TS algorithms (Wang and Chen, 2018; Perrault et al., 2020a), consequently leading to a new $\tilde{O}(\sqrt{T})$ variance-adaptive gap-free regret upper bound for a sampling algorithm.

**A novel gap-free lower bound (Section 2.2).** We show a gap-free lower bound on the regret for stochastic combinatorial semi-bandits, explicitly involving the structure of the problem (the items forming each action) and the covariance matrix $\mathbf{\Sigma}$. This lower bound highlights the optimality of the gap-free upper bound we establish for OLS-UCB-C.

Technical details are deferred to Section 4, Section 5, and the Appendix.

Table 1: Asymptotic $\tilde{O}(\cdot)$ regret bounds and per-round time complexities up to poly-logarithmic terms in $d$, for the following deterministic algorithms: UCB (Auer et al., 2002a), UCBV (Audibert et al., 2009), CUCB (Kveton et al., 2015), OLS-UCB-C (Degenne and Perchet, 2016), ESCB-C (Perrault et al., 2020b), and OLS-UCB-C (ours); as well as the two stochastic algorithms: CTS-Gaussian (Perrault et al., 2020a) and COS-V (ours).
Notations: $a$ refers to actions; $i$ and $j$ refer to items; $m$ denotes the maximum number of items per action; $B$ is a vector of bounds on the items' rewards; $\mathbf{\Gamma}$ is a covariance-proxy matrix; $\gamma$ is the maximum of "correlations-proxy"; we abbreviate $\max\{x, 0\}$ to $(x)_+$ for any $x \in \mathbb{R}$ ; $C_{1/T}^{\text{opt}}$ refers to the complexity of the optimisation step needed in ESCB-C.

| Fdbck. | Algorithm | Info. | Time Complexity | Gap-Free Asymptotic Regret |
|---|---|---|---|---|
| Bndt. | UCB | $B$ | $P$ | $\left(T \sum_a (a^\top B)^2\right)^{1/2}$ |
| | UCBV | $\varnothing$ | $P$ | $\left(T \sum_a a^\top \mathbf{\Sigma} a\right)^{1/2}$ |
| S-Bndt. | CUCB | $B$ | $mP$ | $(Tmd)^{1/2}\|B\|_\infty$ |
| | OLS-UCB[a] | $\mathbf{\Gamma}$ | $m^2 + Pd^2$ | $\varnothing$ |
| | ESCB-C | $\varnothing$ | $m^2 + P\,C_{1/T}^{\text{opt}}$ | $\varnothing$ |
| | OLS-UCB-C | $\varnothing$ | $m^2 + Pd^2$ | $\left(T \sum_i \max_{a/i \in a} \sum_{j \in a} (\mathbf{\Sigma}_{i,j})_+\right)^{1/2}$ |
| S-Bndt. | CTS-Gaussian[b] | $\mathbf{\Gamma}$ | $\text{poly}(d)$ | $\varnothing$ |
| | COS-V[b] | $\varnothing$ | $\text{poly}(d)$ | $\left(Tm \sum_{i \in [d]} \mathbf{\Sigma}_{i,i}\right)^{1/2}$ |

## 2 Covariance-adaptative deterministic algorithm: OLS-UCB-C

In this section, we design a new algorithm that efficiently leverages the semi-bandit feedback. It approximates the coefficients of the covariance matrix $\mathbf{\Sigma}$ online. The approximation is symmetric by construction and yields a coefficient-wise upper bound of $\mathbf{\Sigma}$, but it is not necessarily positive semi-definite, a constraint that can be challenging to impose in practice.

### 2.1 Algorithm: OLS-UCB-C

We present OLS-UCB-C described in Alg. 2 and detail below the successive steps it performs.

**Initial exploration.** The algorithm first explores by choosing every base item $i \in [d]$ and every "reachable" couple $(i, j) \in [d]^2$ at least once.

---

[a]Note that OLS-UCB incur a $\Delta^{-2}$ term in its gap-dependent bound. This was overlooked by the authors but yields a $T^{2/3}$ gap-free bound.

[b]Assuming $\mathcal{A}$ has matroid structure, the computational complexity is improved compared to a $O(P)$ for large $P$.

**Rewards means estimation.** At each round $t + 1 \in [T-1]$, the algorithm uses an empirical mean $\hat{\mu}_t$ for $\mu$ defined as

$$\hat{\mu}_t = \mathbf{N}_t^{-1} \sum_{s=1}^{t} \mathbf{d}_{A_s} Y_s, \tag{2}$$

where $\mathbf{d}_a = \text{diag}(a) \in M_d(\mathbb{R})$ is the diagonal matrix of the elements in $a \in \mathcal{A}$; $n_{t,(i,j)}$ is the number of times items $i$ and $j$ (with possibly $i = j$) have been chosen together; $\mathbf{N}_t = \text{diag}((n_{t,(i,i)})_{i\in[d]}) \in M_d(\mathbb{R})$ is the diagonal matrix of item counts.

**Rewards covariances estimation.** The covariances are estimated by

$$\hat{\chi}_{t,(i,j)} = \hat{\mathbf{S}}_{t,(i,j)} - \hat{\mu}_{t,i}\hat{\mu}_{t,j}, \tag{3}$$

where $\hat{\mathbf{S}}_{t,(i,j)} = \frac{1}{n_{t,(i,j)}} \sum_{s=1}^{t} A_{s,i} A_{s,j} Y_{s,i} Y_{s,j}$.

---

**Algorithm 2** `OLS-UCB-C`

---

**Input** $\delta > 0$, $B \in \mathbb{R}^d_+$.
**for** $t = 1, \ldots, T$ **do**
  **if** $\{a \in \mathcal{A} \,|\, \min_{(i,j)\in a} n_{t,(i,j)} < 1\} \neq \emptyset$
  **then**
    Choose any $A_t$ in the above set.
  **else**
    Compute $\hat{\mu}_{t-1}$ from (2).
    Compute $\hat{\mathbf{\Sigma}}_{t-1}$ from (3) and (4).
    Compute $\hat{\mathbf{Z}}_{t-1}$ from (5).
    Choose $A_t \in \mathcal{A}$ from (6).
    Environment samples $Y_t \in \mathbb{R}^d$.
    Receive reward $\langle A_t, Y_t \rangle = \sum_i A_{t,i} Y_{t,i}$.
  **end if**
**end for**

---

The algorithm uses $\hat{\mathbf{\Sigma}}_t$, a coefficient-wise upper-confidence bound of $\mathbf{\Sigma}$ whose coefficients are defined for a fixed $\delta > 0$ as

$$\hat{\mathbf{\Sigma}}_{t,(i,j)} = \hat{\chi}_{t,(i,j)} + \frac{B_i B_j}{4}\left( \frac{5h_{t,\delta}}{\sqrt{n_{t,(i,j)}}} + \frac{h_{t,\delta}^2}{n_{t,(i,j)}} + \frac{1}{n_{t,(i,j)}^2} \right), \tag{4}$$

where $h_{t,\delta} = \left( 1 + 2\log(1/\delta) + 2\log\left( t\log(t)^2 d(d+1) \right) + \log(1+t) \right)^{1/2}$.

**Optimistic action choice.** Following the 'optimistic' principle of `UCB`-like algorithms, the estimated rewards $(\langle\hat{\mu}_t, a\rangle)_{a\in\mathcal{A}}$ are inflated by bonuses, yielding corresponding upper confidence bounds. The bonuses involve the history through a *regularized empirical design matrix* (with empirical covariances):

$$\hat{\mathbf{Z}}_t = \sum_{s=1}^{t} \mathbf{d}_{A_s} \hat{\mathbf{\Sigma}}_t \mathbf{d}_{A_s} + \mathbf{d}_{\hat{\mathbf{\Sigma}}_t} \mathbf{N}_t + \|B\|^2 \mathbf{I}, \tag{5}$$

where $\mathbf{d}_{\hat{\mathbf{\Sigma}}_t} = \text{diag}(\hat{\mathbf{\Sigma}}_t) \in M_d(\mathbb{R})$, $\mathbf{I}$ is the identity matrix and $\hat{\mathbf{\Sigma}}_t$ is the coefficient-wise upper bound for the covariance matrix defined in (4). Formally, `OLS-UCB-C` chooses

$$A_{t+1} \in \arg\max_{a\in\mathcal{A}} \left\{ \langle a, \hat{\mu}_t \rangle + f_{t,\delta} \left\| \mathbf{N}_t^{-1} a \right\|_{\hat{\mathbf{Z}}_t} \right\}, \tag{6}$$

where $f_{t,\delta} = 6\log(1/\delta) + 6\Big( \log(t) + (d+2)\log(\log(t)) \Big) + 3d\Big( 2\log(2) + \log(1+e) \Big)$.

**Efficiency improvement.** While Perrault et al. (2020b) use an axis-realignment technique to derive their confidence regions, our approach builds ellipsoidal confidence regions. This simplifies the computation of an upper confidence bound for each action as we have a closed-form expression. In comparison, Perrault et al. (2020b) need to solve linear programs in convex sets at each iteration.

## 2.2 Regret upper bounds

**Theorem 1.** *Let $T \in \mathbb{N}^*$ and $\delta > 0$.*

*Then,* `OLS-UCB-C` *(Alg. 2) satisfies the* gap-dependent *regret upper bound*

$$\mathbb{E}[R_T] = \tilde{O}\left( \log(m)^2 \sum_{i=1}^{d} \max_{a\in\mathcal{A}/i\in a, \Delta_a > 0} \frac{\sigma_{a,i}^2}{\Delta_a} \right),$$

*where $\sigma_{a,i}^2 = \sum_{j\in a} \max\{\mathbf{\Sigma}_{i,j}, 0\}$, and the* gap-free *regret upper bound*

$$\mathbb{E}[R_T] = \tilde{O}\left( \log(m)\sqrt{T} \sqrt{\sum_{i=1}^{d} \max_{a\in\mathcal{A}/i\in a} \sigma_{a,i}^2} \right).$$

The proof is outlined in Section 5 and the specific details are presented in Appendix E.

**Optimal gap-free bound.** This result shows that `OLS-UCB-C` yields the same gap-dependent regret upper bound as `ESCB-C` (Perrault et al., 2020b) (up to poly-logarithmic factors) and more importantly yields a novel covariance-adaptive and optimal $O(\sqrt{T})$ gap-free bound, as shown by the following lower-bound proven in Appendix A. Unfortunately, only the positive coefficients of $\Sigma$ are considered in our bound but the inclusion of negative correlations could be advantageous to reduce the rate at which the regret increases. However, it could complicate the analysis greatly and is thus deferred to future research.

**Theorem 2.** *Let $d, m \in \mathbb{N}^*$ such that $d/m \geq 2$ is an integer, $T \in \mathbb{N}^*$, and $\Sigma \succeq 0$ a covariance matrix. Then, there exists a stochastic combinatorial semi-bandit with $d$ base items, and a reward distribution with covariance matrix $\Sigma$ on which for any policy $\pi$, the pseudo regret satisfies*

$$\mathbb{E}[R_T] \geq \frac{1}{8}\left(T \sum_{i \in [d]} \max_{a \in \mathcal{A}, i \in a} \sum_{j \in a} \Sigma_{i,j}\right)^{1/2}.$$

**Improvement over `CUCB`.** Our gap-free and gap-dependent bounds outperform those of `CUCB` (Kveton et al., 2015) no matter the covariance structure, as $(Tmd)^{1/2}\|B\|_\infty \gtrsim (\mathrm{Tr}(\Sigma)T)^{1/2}$.[c] Besides, in the particular case of a diagonal $\Sigma$, our gap-free upper bound gains a factor at least $\sqrt{m}$ over the one of `CUCB`. In this scenario, $\sigma_{a,i}^2 = \Sigma_{i,i}$ for all $a \in \mathcal{A}$ and $i \in a$. Our gap-dependent and gap-free upper bounds are then roughly bounded as

$$\sum_{i=1}^d \frac{\Sigma_{i,i}}{\min_{a \in \mathcal{A}/i \in a} \Delta_a} \text{ and } \sqrt{\mathrm{Tr}(\Sigma)T},$$

respectively.

**Improvement over `UCBV`.** Assuming that $\Sigma_{i,j} \geq 0$ for all $i, j$, our upper-bound uniformly improves the one of `UCBV` of order $\left(T \sum_a a^\top \Sigma a\right)^{1/2}$, since in this case $\sum_{i=1}^d \max_{a \in \mathcal{A}\setminus i \in a} \sigma_{a,i}^2 \leq \sum_a \|a\|_\Sigma$. Existing semi-bandit analyses could only leverage semi-bandit feedback in the regime $P \gg d$, which is natural in combinatorial bandits but not systematic in real-world applications.

## 3 New sampling algorithm for combinatorial semi-bandits: `COS-V`

In this section, we introduce a randomized algorithm inspired from `TS`, enabling to get potentially computational complexity at the cost of not leveraging off-diagonal covariances.

The difficulty in designing and analysing `TS` algorithms generally stems from controlling the random exploration. To that end, we parametrize the exploration distribution using the same estimators as `OLS-UCB-C`.

### 3.1 Algorithm: `COS-V`

We propose a sampling strategy using "frequentist" estimators, `COS-V`, described in Algorithm 3.

The algorithm begins with the same exploration phase as `OLS-UCB-C`. Thereafter at each round $t + 1 \in [T - 1]$, we sample parameters $(\tilde{\mu}_{i,t})_{i \in [d]}$ using 1-dimensional normal distributions biased toward the positive orthant. Formally, for all $i \in [d]$, we sample

---

**Algorithm 3 `COS-V`**

**Input** $\delta > 0$, $B \in \mathbb{R}_+^d$.
**for** $t \in [T]$ **do**
  **if** $\left\{a \in \mathcal{A} \text{ s.t } \min_{(i,j) \in a} n_{t,(i,j)} < 1\right\} \neq \emptyset$
  **then**
    Choose $A_t$ in the above set.
  **else**
    Compute $\hat{\mu}_{t-1}$ (2).
    Compute $(\hat{\Sigma}_{t-1,(i,i)})_{i \in [d]}$ (4).
    Compute $(\hat{\mathbf{Z}}_{t-1,(i,i)})_{i \in [d]}$ from (5).
    Sample $\tilde{\mu}_{t-1}$ from (7)
    Choose $A_t \in \arg\max_{a \in \mathcal{A}}\langle a, \tilde{\mu}_{t-1}\rangle$.
    Environment samples $Y_t \in \mathbb{R}^d$.
    Receive reward $\langle A_t, Y_t\rangle = \sum_i A_{t,i} Y_{t,i}$.
  **end if**
**end for**

---

$$\tilde{\mu}_{t,i} \sim \mathcal{N}\left(\hat{\mu}_{t,i} + (1 + g_{t,\delta})f_{t,\delta}\frac{\hat{\mathbf{Z}}_{t,(i,i)}^{1/2}}{n_{t,(i,i)}}, \; f_{t,\delta}^2 \frac{\hat{\mathbf{Z}}_{t,(i,i)}}{n_{t,(i,i)}^2}\right), \tag{7}$$

where $g_{t,\delta} = \left(1 + 2\log\left(2dt\log(t)^2/\delta\right)\right)^{1/2}$ and $f_{t,\delta}$ is the same as for `OLS-UCB-C`.

---

[c]We denote $\gtrsim$ for $\geq$ up to a constant factor.

## 3.2 Regret upper bound

**Theorem 3.** *Let $T \in \mathbb{N}^*$, and $\delta > 0$.*

*Then, COS-V (Alg. 3) satisfies the gap-dependent regret upper bound*

$$\mathbb{E}[R_T] = \tilde{O}\left( \log(m)^2 \sum_{i=1}^{d} \frac{m\mathbf{\Sigma}_{i,i}}{\Delta_{i,\min}} \right), \tag{8}$$

*where $\Delta_{i,\min} = \min\{\Delta_a, \ a \in \mathcal{A} \ such \ that \ i \in a\}$, and the gap-free regret upper bound*

$$\mathbb{E}[R_T] = \tilde{O}\left( \log(m)\sqrt{T}\sqrt{m \sum_{i=1}^{d} \mathbf{\Sigma}_{i,i}} \right). \tag{9}$$

The proof is outlined in Section 5 and the specific details can be found in Appendix F.

**Novel variance-dependent bound.** Theorem 3 presents the first variance-dependent bound for a sampling-based semi-bandit algorithm. Unfortunately, integrating the covariances $\mathbf{\Sigma}_{i,j}$ in the leading term is still an open problem. Possible leads include exploring other biasing strategies for sampling, or using oversampling approaches like Abeille and Lazaric (2017) which inflate the confidence regions in the linear bandits setting.

**Novel gap-free regret bound.** An important novelty of our gap-dependent bound Eq. (16) is the absence of $\Delta_{\min}^{-m}$ terms present in the previous analyses of CTS (Wang and Chen, 2018; Perrault et al., 2020a). In particular, this improvement yields the first $\tilde{O}(\sqrt{T})$ gap-free regret upper bound for a sampling strategy.

## 4 Mean and covariance estimation

In this section, we present concentration results for $\hat{\mu}_t$ (rewards means) and $\hat{\mathbf{\Sigma}}_t$ (rewards covariances, estimated with $\hat{\chi}_t$) used in OLS-UCB-C and COS-V, which are central to prove Theorem 1 and Theorem 3 (sketched in Section 5).

### 4.1 Covariance-aware confidence region for the average reward

**Average reward estimation.** Let $a \in \mathcal{A}$, $t \geq d(d+1)/2$, as introduced in Section 2.1, the least square estimator for the mean reward vector $\mu$ using all the data gathered after round $t$ is

$$\hat{\mu}_t = \mathbf{N}_t^{-1} \sum_{s=1}^{t} \mathbf{d}_{A_s} Y_s = \mu + \mathbf{N}_t^{-1} \sum_{s=1}^{t} \mathbf{d}_{A_s} \eta_s,$$

where the $\eta_s$ denote the deviations $Y_s - \mu$.

**Confidence region design.** We design confidence regions inspired from LinUCB literature (Rusmevichientong and Tsitsiklis, 2010; Filippi et al., 2010; Abbasi-Yadkori et al., 2011) and the work of Degenne and Perchet (2016). Major differences with those works include using Bernstein's style concentration inequalities involving the covariance matrix $\mathbf{\Sigma}$, assuming a multidimensional noise term, and combining them with a covering argument to relax dependence in $d$ (peeling trick from Degenne and Perchet, 2016). We introduce the *regularized design matrix* defined by

$$\mathbf{Z}_t = \mathbf{V}_t + \mathbf{N}_t \mathbf{d}_{\mathbf{\Sigma}} + \|B\|^2 \mathbf{I},$$

where $\mathbf{V}_t = \sum_{s=1}^{t} \mathbf{d}_{A_s} \mathbf{\Sigma} \mathbf{d}_{A_s}$ is the design matrix (of which the OLS-UCB-C and COS-V use an empirical version). Let $S_t = \mathbf{N}_t(\hat{\mu}_t - \mu)$, the deviations of $\langle a, \hat{\mu}_t \rangle$ are bounded as

$$|\langle a, \hat{\mu}_t - \mu \rangle| \leq \|\mathbf{N}_t^{-1} a\|_{\mathbf{Z}_t} \|S_t\|_{\mathbf{Z}_t^{-1}}. \tag{10}$$

Designing a confidence region for $\|S_t\|_{\mathbf{Z}_t^{-1}}$ therefore allows to control the deviations $|\langle a, \hat{\mu}_t - \mu \rangle|$ uniformly on $\mathcal{A}$. Let $\delta > 0$, we define the event

$$\mathcal{G}_t = \left\{ \|S_t\|_{\mathbf{Z}_t^{-1}} \leq f_{t,\delta} \right\}, \tag{11}$$

with $f_{t,\delta} = 6\log(1/\delta) + 6[\log(t) + (d+2)\log(\log(t))] + 3d[2\log(2) + \log(1+e)]$.

This event can also be written $\mathcal{G}_t = \left\{ \|\hat{\mu}_t - \mu\|_{\mathbf{N}_t \mathbf{Z}_t^{-1} \mathbf{N}_t} \leq f_{t,\delta} \right\}$ and is therefore equivalent to $\hat{\mu}_t$ belonging to an ellipsoid around the true reward mean vector $\mu$.

**Confidence region probability.** The following result proven in Appendix B presents an upper bound for $\mathbb{P}(\mathcal{G}_t^c)$.

**Proposition 1.** *Let $t \geq d(d+1)/2$ and $\delta > 0$. Then, $\mathbb{P}(\mathcal{G}_t^c) \leq \delta/(t \log(t)^2)$.*

Proving this result relies on an argument adapted from Faury et al. (2020) and a covering trick from Degenne and Perchet (2016).

### 4.2 Confidence interval for covariances estimator

**Rewards covariances estimator.** Let $t \geq d(d+1)/2$ and a "reachable" couple $(i,j) \in [d]^2$ . The coefficients of $\boldsymbol{\Sigma}$ can be estimated online by $\hat{\chi}_t$ as introduced in Section 2.1

$$\hat{\chi}_{t,(i,j)} = \hat{\mathbf{S}}_{t,(i,j)} - \hat{\mu}_{t,i}\hat{\mu}_{t,j} .$$

**Rewards covariances upper confidence bound.** Let $\delta > 0$. We use the following coefficient-wise upper estimates of $\boldsymbol{\Sigma}$ in our algorithms

$$\hat{\boldsymbol{\Sigma}}_{t,(i,j)} = \hat{\chi}_{t,(i,j)} + \frac{B_i B_j}{4}\left(\frac{5 h_{t,\delta}}{\sqrt{n_{t,(i,j)}}} + \frac{h_{t,\delta}^2}{n_{t,(i,j)}} + \frac{1}{n_{t,(i,j)}^2}\right),$$

with $h_{t,\delta} = \left(1 + 2\log(1/\delta) + 2\log\left(t\log(t)^2 d(d+1)\right) + \log(1+t)\right)^{1/2}$.

**Favorable event design.** We define $\mathcal{C}_t$ as the event where all the coefficients of $\hat{\boldsymbol{\Sigma}}_t$ are indeed upper bounding those of $\boldsymbol{\Sigma}$:

$$\mathcal{C}_t = \left\{\forall (i,j) \in [d]^2 \text{ "reachable", } \hat{\boldsymbol{\Sigma}}_{t,(i,j)} \geq \boldsymbol{\Sigma}_{i,j}\right\}. \tag{12}$$

**Favorable event probability.** The following result proven in Appendix C presents an upper bound for $\mathbb{P}(\mathcal{C}_t^c)$.

**Proposition 2.** *Let $t \geq d(d+1)/2$ and $\delta > 0$. Then, $\mathbb{P}(\mathcal{C}_t^c) \leq \delta/(t \log(t)^2)$.*

## 5 Regret upper bounds

In this section, we provide a sketch of the proof for Theorem 1 and Theorem 3. For both `OLS-UCB-C` and `COS-V`, the idea to bound the regret is to find a sequence of *favorable events* $(\mathcal{E}_t)_{t \geq d(d+1)/2}$ that are true with high probability, and under which the regret grows logarithmically with time.

### 5.1 Template bound

Let $(\mathcal{E}_t)_{t \in [T]}$ be a sequence of events, then for both `OLS-UCB-C` and `COS-V` standard derivations yield

$$\mathbb{E}[R_T] \leq \Delta_{\max}\left(d(d+1)/2 + \sum_{t=d(d+1)/2}^{T-1} \mathbb{P}(\mathcal{E}_t^c)\right) + \mathbb{E}\left[\sum_{t=d(d+1)/2}^{T-1} \mathbb{1}\{\mathcal{E}_t\}\Delta_{A_{t+1}}\right]. \tag{13}$$

Assuming that the sequence of events $(\mathcal{E}_t)_{t \geq d(d+1)/2}$ happens with high enough probability, it is sufficient to control what happens conditionally to it. In particular, Proposition 6 in Appendix D states that if we can bound $\Delta_{A_{t+1}}^2$ with a linear combination of terms evolving as $n_{t,(i,j)}^{-k}$ for every couple $(i,j) \in A_{t+1}$ and different $k \geq 1$, then we can infer a worst-case behaviour, which yields Theorem 1 and Theorem 3.

In the following, we will refer to the term $\sum_{t=d(d+1)/2}^{T-1} \mathbb{P}(\mathcal{E}_t^c)$ as the *unfavorable event probability* and to the term $\mathbb{E}\left[\sum_{t=d(d+1)/2}^{T} \mathbb{1}\{\mathcal{E}_t\}\Delta_{A_{t+1}}\right]$ as the *high-probability regret*.

### 5.2 Regret of `OLS-UCB-C`

For `OLS-UCB-C` we consider the sequence of events $\mathcal{E}_t = \{\mathcal{G}_t \cap \mathcal{C}_t\}$ for all $t \geq d(d+1)/2$, corresponding the confidence regions of $(\hat{\mu}_t)_{t \geq d(d+1)/2}$ and of $(\hat{\boldsymbol{\Sigma}}_{t,(i,j)})_{t \geq d(d+1)/2, (i,j) \in [d]^2}$ defined in Section 4. Under these events, we can upper-bound the high-probability regret from Eq. (13) with the following proposition (proven in Appendix E.1).

**Proposition 3.** *Let $\delta > 0$. Then,* `OLS-UCB-C` *yields*

$$\mathbb{E}\left[\sum_{t=d(d+1)/2}^{T-1} \Delta_{A_{t+1}} \mathbb{1}\{\mathcal{G}_t \cap \mathcal{C}_t\}\right] = O\left(\log(T)^2 \log(m)^2 \sum_{i=1}^{d} \max_{a \in \mathcal{A}/i \in a} \frac{\sigma_{a,i}^2}{\Delta_a}\right),$$

*as $T \to \infty$, where $\sigma_{a,i}^2 = \sum_{j \in a}(\mathbf{\Sigma}_{i,j})_+$.*

**Conclusion of the proof.** Injecting results from Proposition 3 (high-probability regret) as well as Proposition 1 and Proposition 2 (unfavorable event probability) into the template bound (13), we get

$$\mathbb{E}[R_T] = O\left(\log(T)^2 \log(m)^2 \sum_{i \in [d]} \max_{a \in \mathcal{A}/i \in a} \frac{\sigma_{a,i}^2}{\Delta_a}\right), \tag{14}$$

for `OLS-UCB-C` as $T \to \infty$. This provides the gap-dependent bound of Theorem 1. The gap-free bound is detailed in Appendix E.4. It is enabled by the fact that our gap-dependent bound does not incur any term in $\Delta_{\min}^{-2}$, unlike Perrault et al. (2020b); Degenne and Perchet (2016).

### 5.3  Regret of `COS-V`

For the analysis of our stochastic algorithm `COS-V`, we need to consider events related to the sampling distributions in addition to the events $\mathcal{G}_t'$ and $\mathcal{C}_t$ introduced in the precedent section. For this purpose, we denote the event $\mathcal{H}_t$ defined as

$$\mathcal{H}_t = \left\{\forall i \in [d], \left|\left(\hat{\mu}_{t,i} + (1 + g_{t,\delta})f_{t,\delta}\frac{(\hat{\mathbf{Z}}_{t,i})^{1/2}}{n_{t,i}}\right) - \tilde{\mu}_{t,i}\right| \leq g_{t,\delta} f_t \frac{(\hat{\mathbf{Z}}_{t,i})^{1/2}}{n_{t,i}}\right\}. \tag{15}$$

The high-level idea of the event $\mathcal{H}_t$ to ensure that the sampled rewards $\tilde{\mu}_{t,i}$ upper-bound the true mean $\mu_i$ while not being too far for all the items $i \in a^*$. Showing that the event $\mathcal{H}_t$ indeed occurs with high-probability (Lemma 7 in Appendix F) and setting the events $\mathcal{E}_t = \{\mathcal{G}_t \cap \mathcal{C}_t \cap \mathcal{H}_t\}$, we can upper-bound the high-probability regret in the following proposition (proof is in Appendix F.2).

**Proposition 4.** *Let $\delta > 0$. Then* `COS-V` *yields*

$$\mathbb{E}\left[\sum_{t=d(d+1)}^{T-1} \Delta_{A_{t+1}} \mathbb{1}\{\mathcal{G}_t \cap \mathcal{C}_t \cap \mathcal{H}_t\}\right] = O\left(\log(T)^3 \log(m)^2 \left(\sum_{i=1}^{d} \frac{m\mathbf{\Sigma}_{i,i}}{\Delta_{i,\min}}\right)\right).$$

**Conclusion of the proof.** Injecting results from Proposition 4 (high-probability regret) as well as Lemma 7, Proposition 1 and Proposition 2 (unfavorable event probability) into the template bound (13) yields

$$\mathbb{E}[R_T] = O\left(\log(T)^3 \log(m)^2 \sum_{i \in [d]} \frac{m\mathbf{\Sigma}_{i,i}}{\Delta_{i,\min}}\right), \tag{16}$$

as $T \to \infty$. This provides the gap-dependent bound of Theorem 3. As it does not incur any term in $\Delta_{\min}^{-m}$ as in Wang and Chen (2018); Perrault et al. (2020a), this result can be used to derive a $\tilde{O}(\sqrt{T})$ gap-free bound for a sampling-based combinatorial semi-bandit algorithm.

## 6  Concluding remarks

We propose and analyze two algorithms for combinatorial semi-bandits. `OLS-UCB-C` is a deterministic, covariance-adaptive algorithm. Compared to other existing approaches, our algorithm is typically less computationally demanding and yields the first $\tilde{O}(\sqrt{T})$ gap-free regret rate that explicitly depends on the covariance of the base item rewards and the structure. `COS-V` is a variance-adaptive, `TS`-like algorithm. Its complexity is significantly lower under certain types of constraints, but its regret is suboptimal as it assumes worst-case correlations. However, leveraging the analysis of `OLS-UCB-C`, it also yields the first $\tilde{O}(\sqrt{T})$ gap-free regret upper bound among sampling-based approaches.

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

# Appendix

The Supplementary is organized as follows:

- Appendix A proves the lower bound from Theorem 2,
- Appendix B outlines proofs concerning the concentration of the average estimator (Propositions 1)
- Appendix C presents those for the covariance estimator (Proposition 2),
- Appendix D establishes general propositions used to upper-bound the number of times each item is chosen,
- Appendix E and Appendix F detail proofs for OLS-UCB-C and COS-V,
- Appendix G presents some experimental results.

## A   Proof of the lower bound (Theorem 2)

**Theorem 2.** *Let $d, m \in \mathbb{N}^*$ such that $d/m \geq 2$ is an integer, $T \in \mathbb{N}^*$, and $\boldsymbol{\Sigma} \succeq 0$ a covariance matrix. Then, there exists a stochastic combinatorial semi-bandit with $d$ base items, and a reward distribution with covariance matrix $\boldsymbol{\Sigma}$ on which for any policy $\pi$, the pseudo regret satisfies*

$$\mathbb{E}[R_T] \geq \frac{1}{8}\left(T \sum_{i \in [d]} \max_{a \in \mathcal{A}, i \in a} \sum_{j \in a} \boldsymbol{\Sigma}_{i,j}\right)^{1/2}.$$

*Proof.* We follow the methodology of Auer et al. (2002b), modifying it to account for the different variances among actions.

Let $d, m \in \mathbb{N}^*$ such that $d/m \geq 2$ is an integer, $T \in \mathbb{N}^*$, and a covariance matrix $\boldsymbol{\Sigma} \succeq 0$. We consider the structure where $\mathcal{A} = \{a_1, \ldots, a_{d/m}\} \subset \{0,1\}^d$ contains $d/m$ disjoint actions each having $m$ base elements. We consider that for all $p \in [d/m]$, $(a_p)_{i \in [d]} = \left(\mathbb{1}\{(p-1)m < i \leq pm\}\right)_{i \in [d]}$. Let $\pi$ be a policy. As all the actions are disjoints, we can reduce ourselves to a multi-armed bandit with $d/m$ actions, where for all $p \in [d/m]$ the variance of the $p$-th action is $\langle a_p, \boldsymbol{\Sigma}_p \rangle$.

Let $\boldsymbol{\Sigma}' \in M_{d/m}(\mathbb{R})$ be the diagonal matrix where for all $p \in [d/m]$, $\boldsymbol{\Sigma}'_{p,p} = a_p^\top \boldsymbol{\Sigma} a_p$. Let $c > 0$, and

$$\Delta = 2c\sqrt{\boldsymbol{\Sigma}'_{\min} \frac{\sum_{k=1}^{d/m} \boldsymbol{\Sigma}'_{k,k}}{T}}, \tag{17}$$

where $\boldsymbol{\Sigma}'_{\min} = \min_{p \in [d/m]} \boldsymbol{\Sigma}'_{p,p}$.

We denote $G_0 \sim \mathcal{N}(0, \boldsymbol{\Sigma}')$ a $(d/m)$-dimensional centered Gaussian distribution with covariance matrix $\boldsymbol{\Sigma}'$. Let $p \in [d/m]$, we consider the mean vector $\mu^{(p)} \in \mathbb{R}^{d/m}$ having coordinate 0 everywhere and $\Delta$ at coordinate $p$, for all $i \in [d/m]$, $\mu_i^{(p)} = \Delta \mathbb{1}\{i = p\}$. We introduce the Gaussian reward distributions $G_p \sim \mathcal{N}(\mu^{(p)}, \boldsymbol{\Sigma}')$ and denote $T_p = \sum_{t=1}^T \mathbb{1}\{A_t = p\}$. Then, using policy $\pi$, and considering the reward distributions $G_p$ and $G_0$, the average number of times action $p$ has been chosen satisfies

$$\left|\mathbb{E}_{\pi, G_p}[T_p] - \mathbb{E}_{\pi, G_0}[T_p]\right| \leq T \, \mathrm{TV}\left((\pi, G_0), (\pi, G_p)\right) \leq T\sqrt{\frac{1}{2}\mathrm{KL}\left((\pi, G_0), (\pi, G_p)\right)}, \tag{18}$$

where TV denotes the total variation distance, KL denotes the Kullback–Leibler divergence and the last inequality uses Pinsker's inequality. Then, using the divergence decomposition between multi-armed bandits (Lemma 15.1 in Lattimore and Szepesvári, 2020),

$$\mathrm{KL}\left((\pi, G_0), (\pi, G_p)\right) = \sum_{k=1}^{d/m} \mathbb{E}_{\pi, G_0}\left[T_k\right] \mathrm{KL}\left(\mathcal{N}(0, \boldsymbol{\Sigma}'), \mathcal{N}(\mu^{(p)}, \boldsymbol{\Sigma}')\right)$$

$$= \sum_{k=1}^{d/m} \mathbb{E}_{\pi, G_0}\left[T_k\right] \frac{\left(\mu_k^{(p)}\right)^2}{2\boldsymbol{\Sigma}'_{k,k}}.$$

Reinjecting this expression into Eq. (18), we get

$$\mathbb{E}_{\pi,G_p}[T_p] \leq \mathbb{E}_{\pi,G_0}[T_p] + \frac{T}{2}\sqrt{\sum_{k=1}^{d/m} \frac{\left(\mu_k^{(p)}\right)^2}{\boldsymbol{\Sigma}'_{k,k}} \mathbb{E}_{\pi,G_0}[T_k]}$$

$$= \mathbb{E}_{\pi,G_0}[T_p] + \frac{T}{2}\sqrt{\frac{1}{\boldsymbol{\Sigma}'_{p,p}}\Delta^2 \mathbb{E}_{\pi,G_0}[T_p]}$$

$$= \mathbb{E}_{\pi,G_0}[T_p] + c\sqrt{T\mathbb{E}_{\pi,G_0}[T_p]\frac{\boldsymbol{\Sigma}'_{\min}}{\boldsymbol{\Sigma}'_{p,p}}\sum_{k=1}^{d/m}\boldsymbol{\Sigma}'_{k,k}} \qquad \leftarrow \text{reinjecting Eq. (17)}$$

$$\leq \mathbb{E}_{\pi,G_0}[T_p] + c\sqrt{T\mathbb{E}_{\pi,G_0}[T_p]\sum_{k=1}^{d/m}\boldsymbol{\Sigma}'_{k,k}} \,.$$

Now, summing over the actions $p$,

$$\sum_{p=1}^{d/m}\mathbb{E}_{\pi,G_p}[T_p] \leq \sum_{p=1}^{d/m}\mathbb{E}_{\pi,G_0}[T_p] + c\sqrt{T\sum_{k=1}^{d/m}\boldsymbol{\Sigma}'_{k,k}}\sum_{p=1}^{d/m}\sqrt{\mathbb{E}_{\pi,G_0}[T_p]}$$

$$\leq T + c\sqrt{T\sum_{k=1}^{d/m}\boldsymbol{\Sigma}'_{k,k}}\sqrt{\frac{d}{m}}\sqrt{\sum_{p=1}^{d/m}\mathbb{E}_{\pi,G_0}[T_p]} \qquad \leftarrow \text{Cauchy–Schwarz}$$

$$\leq T + cT\sqrt{\frac{d}{m}\sum_{k=1}^{d/m}\boldsymbol{\Sigma}'_{k,k}} \,. \tag{19}$$

We denote $R_T^{(p)}$ the average cumulative regret incurred with the reward distribution $G_p$, then

$$\sum_{p=1}^{d/m}R_T^{(p)} = \Delta\sum_{p=1}^{d/m}(T - \mathbb{E}_{\pi,G_p}[T_p])$$

$$= 2c\sqrt{\boldsymbol{\Sigma}'_{\min}\frac{\sum_{k=1}^{d/m}\boldsymbol{\Sigma}'_{k,k}}{T}}\left(\frac{d}{m}T - \sum_{p=1}^{d/m}\mathbb{E}_{\pi,G_p}[T_p]\right) \qquad \leftarrow \text{reinjecting Eq. (17)}$$

$$\geq 2c\sqrt{\boldsymbol{\Sigma}'_{\min}\frac{\sum_{k=1}^{d/m}\boldsymbol{\Sigma}'_{k,k}}{T}}\left(\frac{d}{m}T - T - cT\sqrt{\frac{d}{m}\sum_{k=1}^{d/m}\boldsymbol{\Sigma}'_{k,k}}\right) \qquad \leftarrow \text{from Eq. (19)}$$

$$= 2c\sqrt{\boldsymbol{\Sigma}'_{\min}}\frac{d}{m}\sqrt{T\sum_{k=1}^{d/m}\boldsymbol{\Sigma}'_{k,k}}\left(1 - \frac{m}{d} - c\sqrt{\frac{1}{d/m}\sum_{k=1}^{d/m}\boldsymbol{\Sigma}'_{k,k}}\right)$$

$$\geq 2c\sqrt{\boldsymbol{\Sigma}'_{\min}}\frac{d}{m}\sqrt{T\sum_{k=1}^{d/m}\boldsymbol{\Sigma}'_{k,k}}\left(1 - \frac{m}{d} - c\sqrt{\boldsymbol{\Sigma}'_{\min}}\right) \,.$$

Taking $c = \frac{1}{2}\frac{1}{\sqrt{\boldsymbol{\Sigma}'_{\min}}}(1 - \frac{m}{d})$,

$$\sum_{p=1}^{d/m}R_T^{(p)} \geq \frac{d}{m}\sqrt{T\sum_{k=1}^{d/m}\boldsymbol{\Sigma}'_{k,k}}\frac{1}{2}\left(1 - \frac{m}{d}\right)^2$$

$$\geq \frac{1}{8}\frac{d}{m}\sqrt{T\sum_{k=1}^{d/m}\boldsymbol{\Sigma}'_{k,k}} \leftarrow \text{as } m/d \leq 1/2 \,.$$

Therefore, there exists at least one instance $p^* \in [d/m]$ such that

$$R_T^{(p^*)} \geq \frac{1}{8} \sqrt{T \sum_{k=1}^{d/m} \boldsymbol{\Sigma}'_{k,k}} \, .$$

Now, decomposing

$$\sum_{k=1}^{d/m} \boldsymbol{\Sigma}'_{k,k} = \sum_{k=1}^{d/m} \Big( \sum_{i \in a_k} \sum_{j \in a_k} \boldsymbol{\Sigma}_{i,j} \Big) = \sum_{i \in [d]} \max_{a \in \mathcal{A}, i \in a} \sum_{j \in a} \boldsymbol{\Sigma}_{i,j} \, ,$$

we get

$$R_T^{(p^*)} \geq \frac{1}{8} \sqrt{T \sum_{i \in [d]} \max_{a \in \mathcal{A}, i \in a} \sum_{j \in a} \boldsymbol{\Sigma}_{i,j}} \, .$$

$\square$

## B Concentration of the average rewards estimations (Proposition 1)

**Proposition 1.** *Let $t \geq d(d+1)/2$ and $\delta > 0$. Then, $\mathbb{P}(\mathcal{G}_t^c) \leq \delta/(t \log(t)^2)$.*

*Proof.* Let $t \geq d(d+1)/2$ and $\delta > 0$.

We have

$$f_{t,\delta} = 6 \log(1/\delta) + 6 \Big( \log(t) + (d+2) \log(\log(t)) \Big) + 3d \Big( 2 \log(2) + \log(1+e) \Big)$$

$$= 6 \log \left( \frac{t \log(t)^2}{\delta} \left( \frac{\log(t)}{\log(1 + (e-1))} \right)^d + \Big( 6d \log(2) + 3d \log(2 + (e-1)) \Big) \right).$$

**Covering argument (*Peeling trick*).** The peeling trick consists in separating the space of trajectories up to round $t$ into an exponentially large number of parts, each having an exponentially small probability.

Formally, let $0 < \epsilon < 1$. For each $p \in \mathbb{N}^d$ we associate the set

$$\mathcal{D}_p = \left\{ x \in \mathbb{R}^d \text{ s.t. } \forall i \in [d], \ (1+\epsilon)^{p_i} \leq x_i < (1+\epsilon)^{p_i+1} \right\}. \tag{20}$$

As an abuse of notation, we denote by $(t \in \mathcal{D}_p)$ the event $\left( (n_{t,(i,i)} + 1)_{i \in [d]} \in \mathcal{D}_p \right)$.

Setting $P_{t,\epsilon} = \left\lfloor \frac{\log(t)}{\log(1+\epsilon)} \right\rfloor$, we define for each $p \in [P_{t,\epsilon}]^d$

$$\tilde{\mathbf{N}}_p = \mathrm{diag}\Big( \big((1+\epsilon)^{p_i}\big)_{i \in [d]} \Big) \in M_d(\mathbb{R}) \, ,$$

$$\mathbf{Z}_{t,p} = \mathbf{V}_t + \tilde{\mathbf{N}}_p \mathbf{d}_{\boldsymbol{\Sigma}} + \|B\|^2 \mathbf{I} \, . \tag{21}$$

In particular, under the event $(t \in \mathcal{D}_p)$, $\mathbf{N}_t \preceq (1+\epsilon) \tilde{\mathbf{N}}_p$.

Using this covering, we decompose

$$\mathbb{P}(\mathcal{G}_t^c) = \mathbb{P}\left( \left\| \sum_{s=1}^t \mathbf{d}_{A_s} \eta_s \right\|_{\mathbf{Z}_t^{-1}} > f_{t,\delta} \right)$$

$$= \sum_{p \in [P_{t,\epsilon}]^d} \mathbb{P}\left( \left( \|S_t\|_{\mathbf{Z}_t^{-1}} > f_{t,\delta} \right) \cap (t \in \mathcal{D}_p) \right)$$

$$\leq \sum_{p \in [P_{t,\epsilon}]^d} \mathbb{P}\left( \left( \|S_t\|_{\mathbf{Z}_{t,p}^{-1}} > f_{t,\delta} \right) \cap (t \in \mathcal{D}_p) \right).$$

We now apply the following Lemmas.

**Lemma 1.** *Let $t \geq d(d+1)/2$, $0 < \epsilon < 1$, $p \in [P_{t,\epsilon}]^d$, and $\delta > 0$. Then,*

$$\mathbb{P}\left( \|S_t\|_{\mathbf{Z}_{t,p}^{-1}} > 6\log\left(\frac{\mathrm{Norm}_p}{\mathrm{Norm}_{t,p}}\right) + 6\log(1/\delta) \right) \leq \delta\,, \tag{22}$$

*where*

$$\mathrm{Norm}_p = \int_{\lambda \in \mathbb{R}^d} \mathbb{1}\left\{ \|\mathbf{Z}_{0,p}^{1/2}\lambda\| \leq \frac{1}{2} \right\} \exp\{-\|\lambda\|^2_{\mathbf{Z}_{0,p}}\} d\lambda\,,$$

$$\mathrm{Norm}_{t,p} = \int_{\lambda \in \mathbb{R}^d} \mathbb{1}\left\{ \|\mathbf{Z}_{t,p}^{1/2}\lambda\| \leq \frac{1}{2} \right\} \exp\{-\|\lambda\|^2_{\mathbf{Z}_{t,p}}\} d\lambda\,.$$

**Lemma 2.** *Let $t \geq d(d+1)/2$, $0 < \epsilon < 1$ and $p \in [P_{t,\epsilon}]^d$. Then,*

$$\log\left(\frac{\mathrm{Norm}_p}{\mathrm{Norm}_{p,t}}\right) \leq d\log(2) + \frac{1}{2}\log\left(\frac{\det(\mathbf{Z}_{t,p})}{\det(\mathbf{Z}_{0,p})}\right)\,. \tag{23}$$

*Moreover, under event $(t \in \mathcal{D}_p)$,*

$$\log\left(\frac{\mathrm{Norm}_p}{\mathrm{Norm}_{p,t}}\right) \leq d\log(2) + \frac{1}{2}d\log(2+\epsilon)\,. \tag{24}$$

In our case, setting $\epsilon = e - 1$, they yield

$$\mathbb{P}(\mathcal{G}_t^c) \leq \sum_{p \in [P_{t,\epsilon}]^d} \mathbb{P}\Bigg( \left\{ \|S_t\|_{\mathbf{Z}_{t,p}^{-1}} > 6\log\left(\frac{t\log(t)^2}{\delta}\log(t)^d\right) + \left(6d\log(2) + 3d\log(1+e)\right) \right\}$$

$$\cap\, (t \in \mathcal{D}_p) \Bigg)$$

$$\leq \sum_{p \in [P_{t,\epsilon}]^d} \mathbb{P}\Bigg( \left\{ \|S_t\|_{\mathbf{Z}_{t,p}^{-1}} > 6\log\left(\frac{t\log(t)^2}{\delta}\log(t)^d\right) + 6\log\left(\frac{\mathrm{Norm}_p}{\mathrm{Norm}_{t,p}}\right) \right\}$$

$$\cap\, (t \in \mathcal{D}_p) \Bigg) \;\leftarrow\; \text{Lemma } 2$$

$$\leq \sum_{p \in [P_{t,\epsilon}]^d} \mathbb{P}\Bigg( \|S_t\|_{\mathbf{Z}_{t,p}^{-1}} > 6\log\left(\frac{t\log(t)^2}{\delta}\log(t)^d\right) + 6\log\left(\frac{\mathrm{Norm}_p}{\mathrm{Norm}_{t,p}}\right) \Bigg)$$

$$\leq \sum_{p \in [P_{t,\epsilon}]^d} \delta \frac{1}{t\log(t)^2}\frac{1}{\log(t)^d} \;\leftarrow\; \text{Lemma } 1$$

$$= \delta \frac{1}{t\log(t)^2}\frac{1}{\log(t)^d}\log(t)^d$$

$$= \frac{\delta}{t\log(t)^2}\,.$$

$\square$

## B.1 Proof of Lemma 1

**Lemma 1.** *Let $t \geq d(d+1)/2$, $0 < \epsilon < 1$, $p \in [P_{t,\epsilon}]^d$, and $\delta > 0$. Then,*

$$\mathbb{P}\left( \|S_t\|_{\mathbf{Z}_{t,p}^{-1}} > 6\log\left(\frac{\mathrm{Norm}_p}{\mathrm{Norm}_{t,p}}\right) + 6\log(1/\delta) \right) \leq \delta\,, \tag{22}$$

*where*

$$\mathrm{Norm}_p = \int_{\lambda \in \mathbb{R}^d} \mathbb{1}\left\{ \|\mathbf{Z}_{0,p}^{1/2}\lambda\| \leq \frac{1}{2} \right\} \exp\{-\|\lambda\|^2_{\mathbf{Z}_{0,p}}\} d\lambda\,,$$

$$\mathrm{Norm}_{t,p} = \int_{\lambda \in \mathbb{R}^d} \mathbb{1}\left\{ \|\mathbf{Z}_{t,p}^{1/2}\lambda\| \leq \frac{1}{2} \right\} \exp\{-\|\lambda\|^2_{\mathbf{Z}_{t,p}}\} d\lambda\,.$$

*Proof.* We adapt the proofs from Faury et al. (2020), which adapts Abbasi-Yadkori et al. (2011) itself. Let $t \geq d(d+1)/2$, $0 < \epsilon < 1$, $p \in [P_{t,\epsilon}]^d$, and $\delta > 0$.

Let $\lambda \in \mathbb{R}^d$ such that $\|\lambda\| \leq \frac{1}{2\|B\|}$ and $s \in [t]$. We denote $\mathcal{F}'_{t-1} = \sigma(A_1, Y_1, \ldots, A_{t-1}, Y_{t-1}, A_t)$.

Then, $\|\lambda^\top \mathbf{d}_{A_s} \eta_s\| \leq 1/2$ and

$$\mathbb{E}\left[\exp\left(\lambda^\top \mathbf{d}_{A_s} \eta_s - \lambda^\top \mathbf{d}_{A_s} \boldsymbol{\Sigma} \mathbf{d}_{A_s} \lambda\right) \Big| \mathcal{F}'_{s-1}\right] \leq 1$$

which yields that $\left(M_k(\lambda)\right)_{k \in \mathbb{N}^*} = \left(\exp\left(\lambda^\top S_k - \|\lambda\|_{V_k}^2\right)\right)_{k \in \mathbb{N}^*}$ is a $\mathcal{F}'_k$-supermartingale.

Let $p \in [P_{t,\epsilon}]^d$, we consider the density $g_p$ of a $d$-dimensional Gaussian with covariance matrix $\frac{1}{2}(\tilde{\mathbf{N}}_p \mathbf{d}_{\boldsymbol{\Sigma}} + \|B\|^2\mathbf{I})^{-1} = \frac{1}{2}\mathbf{Z}_{0,p}^{-1}$, truncated in the ellipsoid $\{x \in \mathbb{R}^d, \|\mathbf{Z}_{0,p}^{1/2}x\| \leq \frac{1}{2}\}$,

$$g_p(x) = \frac{\mathbb{1}\{x \in \mathbb{R}^d, \|\mathbf{Z}_{0,p}^{1/2}x\| \leq \frac{1}{2}\}}{\text{Norm}_p} \exp\left(-\|x\|_{\mathbf{Z}_{0,p}}^2\right),$$

where $\text{Norm}_p$ is the normalisation constant.

We integrate $\left(M_k(\lambda)\right)_{k \in \mathbb{N}^*}$ for $\lambda \sim g_p$, and define $(\bar{M}_{p,k})_{k \in \mathbb{N}^*}$ as

$$\bar{M}_{p,k} = \int_{\lambda \in \mathbb{R}^d} M_k(\lambda) d\lambda = \int_{\lambda \in \mathbb{R}^d} \frac{\mathbb{1}\{\|\mathbf{Z}_{0,p}^{1/2}\lambda\| \leq \frac{1}{2}\}}{\text{Norm}_p} \exp\left(\lambda^\top S_k - \|\lambda\|_{\mathbf{Z}_{k,p}}^2\right) d\lambda,$$

which is still a supermartingale.

Let $\lambda_{t,p}^* \in \arg\max_{\{\mathbf{Z}_{0,p}^{1/2}\|\lambda\| \leq \frac{1}{4}\}}(\lambda^\top S_t - \|\lambda\|_{\mathbf{Z}_{t,p}}^2)$. Then

$$
\begin{aligned}
\bar{M}_{p,k} &= \frac{\exp(\lambda_{t,p}^{*\top}S_k - \|\lambda_{t,p}^*\|_{\mathbf{Z}_{k,p}}^2)}{\text{Norm}_p} \int_{\|\mathbf{Z}_{0,p}^{1/2}\lambda\| \leq \frac{1}{2}} \exp\left((\lambda - \lambda_{t,p}^*)^\top S_t - \|\lambda\|_{\mathbf{Z}_{k,p}}^2 + \|\lambda_{t,p}^*\|_{\mathbf{Z}_{k,p}}^2\right) d\lambda \\
&= \frac{\exp(\lambda_{t,p}^{*\top}S_k - \|\lambda_{t,p}^*\|_{\mathbf{Z}_{t,p}}^2)}{\text{Norm}_p} \int_{\|\mathbf{Z}_{0,p}^{1/2}\lambda + \mathbf{Z}_{0,p}^{1/2}\lambda_{t,p}^*\| \leq \frac{1}{2}} \exp\left(\lambda^\top S_k - \|\lambda\|_{\mathbf{Z}_{k,p}}^2 - 2\lambda^\top \mathbf{Z}_{k,p}\lambda_{t,p}^*\right) d\lambda \\
&\geq \frac{\exp(\lambda_{t,p}^{*\top}S_k - \|\lambda_{t,p}^*\|_{\mathbf{Z}_{t,p}}^2)}{\text{Norm}_p} \int_{\|\mathbf{Z}_{0,p}^{1/2}\lambda\| \leq \frac{1}{4}} \exp\left(\lambda^\top S_k - \|\lambda\|_{\mathbf{Z}_{k,p}}^2 - 2\lambda^\top \mathbf{Z}_{k,p}\lambda_{t,p}^*\right) d\lambda \\
&= \frac{\exp(\lambda_{t,p}^{*\top}S_k - \|\lambda_{t,p}^*\|_{\mathbf{Z}_{k,p}}^2)}{\text{Norm}_p} \int_{\|\mathbf{Z}_{0,p}^{1/2}\lambda\| \leq \frac{1}{4}} \exp\left(\lambda^\top\left(S_k - 2\mathbf{Z}_{k,p}\lambda_{t,p}^*\right) - \|\lambda\|_{\mathbf{Z}_{k,p}}^2\right) d\lambda \\
&= \frac{\exp(\lambda_{t,p}^{*\top}S_k - \|\lambda_{t,p}^*\|_{\mathbf{Z}_{k,p}}^2)}{\text{Norm}_p} \text{Norm}_{k,p} \\
&\quad \int_{\lambda \in \mathbb{R}^d} \frac{\mathbb{1}\{\|\mathbf{Z}_{0,p}^{1/2}\lambda\| \leq \frac{1}{4}\}}{\text{Norm}_{k,p}} \exp\left(\lambda^\top\left(S_k - 2\mathbf{Z}_{k,p}\lambda_{t,p}^*\right) - \|\lambda\|_{\mathbf{Z}_{k,p}}^2\right) d\lambda,
\end{aligned}
$$

where we can recognize $g_{k,p}$ the density of a $d$-dimensional Gaussian with covariance matrix $\frac{1}{2}\mathbf{Z}_{k,p}^{-1}$, truncated in the ellipsoid $\{x \in \mathbb{R}^d, \|\mathbf{Z}_{0,p}^{1/2}x\| \leq \frac{1}{4}\}$,

$$g_{k,p}(x) = \frac{\mathbb{1}\{\|\mathbf{Z}_{0,p}^{1/2}x\| \leq \frac{1}{4}\}}{\text{Norm}_{k,p}} \exp\left(-\|x\|_{\mathbf{Z}_{k,p}}^2\right),$$

with $\text{Norm}_{k,p}$ the normalisation constant.

Besides, Jensen's inequality yields

$$\int_{\lambda \in \mathbb{R}^d} \frac{\mathbb{1}\{\|\mathbf{Z}_{0,p}^{1/2}\lambda\| \leq \frac{1}{4}\}}{\text{Norm}_{k,p}} \exp\left(\lambda^\top\left(S_k - 2\mathbf{Z}_{k,p}\lambda_{t,p}^*\right) - \|\lambda\|_{\mathbf{Z}_{k,p}}^2\right) d\lambda$$

$$= \int_{\mathbb{R}^d} g_{k,p}(\lambda) \exp\left(\lambda^\top\left(S_k - 2\mathbf{Z}_{k,p}\lambda_{t,p}^*\right)\right) d\lambda$$

$$\geq \exp\left(\int_{\mathbb{R}^d} g_{t,p}(\lambda)\lambda^\top\left(S_k - 2\mathbf{Z}_{k,p}\lambda_{t,p}^*\right) d\lambda\right)$$

$$= \exp\left(\left(S_k - 2\mathbf{Z}_{k,p}\lambda_{t,p}^*\right)^\top \int_{\mathbb{R}^d} g_{t,p}(\lambda)\lambda d\lambda\right)$$

$$= 1.$$

Therefore, for all $k \in \mathbb{N}^*$

$$1 \geq \bar{M}_{p,k} \geq \frac{\text{Norm}_{k,p}}{\text{Norm}_p} \exp(\lambda_{t,p}^{*\top}S_k - \|\lambda_{t,p}^*\|_{\mathbf{Z}_{k,p}}^2).$$

Markov's inequality yields

$$\delta \geq \mathbb{P}\left(\bar{M}_{p,k} \geq \frac{1}{\delta}\right)$$

$$\geq \mathbb{P}\left(\frac{\text{Norm}_{k,p}}{\text{Norm}_p} \exp(\lambda_{t,p}^{*\top}S_k - \|\lambda_{t,p}^*\|_{\mathbf{Z}_{k,p}}^2) \geq \frac{1}{\delta}\right)$$

$$= \mathbb{P}\left(\lambda_{t,p}^{*\top}S_k - \|\lambda_{t,p}^*\|_{\mathbf{Z}_{k,p}}^2 \geq \log\left(\frac{\text{Norm}_p}{\text{Norm}_{k,p}}\right) + \log(1/\delta)\right).$$

Taking $k = t$ in particular gives

$$\delta \geq \mathbb{P}\left(\max_{\|\mathbf{Z}_{0,p}^{1/2}\lambda\| \leq \frac{1}{4}} \lambda^\top S_t - \|\lambda\|_{\mathbf{Z}_{t,p}}^2 \geq \log\left(\frac{\text{Norm}_p}{\text{Norm}_{t,p}}\right) + \log(1/\delta)\right).$$

The constraint on $\lambda$ in the inner expression prevent to use the usual optimal value for subgaussian r.v. which could give a bound for $\|S_t\|_{Z_{t,p}^{-1}}^2$. Instead, we introduce

$$\lambda_{t,p} = \frac{1}{4}\frac{Z_{t,p}^{-1}S_t}{\|S_t\|_{Z_{t,p}^{-1}}},$$

for which

$$\|\mathbf{Z}_{0,p}^{1/2}\lambda_{t,p}\| \leq \frac{1}{4}\|\mathbf{Z}_{0,p}^{1/2}\mathbf{Z}_{t,p}^{-1/2}\|\frac{\|S_t\|_{\mathbf{Z}_{t,p}^{-1}}}{\|S_t\|_{\mathbf{Z}_{t,p}^{-1}}}$$

$$\leq \frac{1}{4}.$$

Then

$$\delta \geq \mathbb{P}\left(\frac{1}{4}\|S_t\|_{\mathbf{Z}_{t,p}^{-1}} - \frac{1}{16}\|S_t\|_{\mathbf{Z}_{t,p}^{-1}} \geq \log\left(\frac{\text{Norm}_p}{\text{Norm}_{t,p}}\right) + \log(1/\delta)\right)$$

$$= \mathbb{P}\left(\|S_t\|_{\mathbf{Z}_{t,p}^{-1}} \geq \frac{16}{3}\log\left(\frac{\text{Norm}_p}{\text{Norm}_{t,p}}\right) + \frac{16}{3}\log(1/\delta)\right)$$

$$\geq \mathbb{P}\left(\|S_t\|_{\mathbf{Z}_{t,p}^{-1}} \geq 6\log\left(\frac{\text{Norm}_p}{\text{Norm}_{t,p}}\right) + 6\log(1/\delta)\right).$$

$\square$

## B.2 Proof of Lemma 2

**Lemma 2.** *Let* $t \geq d(d+1)/2$, $0 < \epsilon < 1$ *and* $p \in [P_{t,\epsilon}]^d$. *Then,*

$$\log\left(\frac{\mathrm{Norm}_p}{\mathrm{Norm}_{p,t}}\right) \leq d\log(2) + \frac{1}{2}\log\left(\frac{\det(\mathbf{Z}_{t,p})}{\det(\mathbf{Z}_{0,p})}\right). \tag{23}$$

*Moreover, under event* $(t \in \mathcal{D}_p)$,

$$\log\left(\frac{\mathrm{Norm}_p}{\mathrm{Norm}_{p,t}}\right) \leq d\log(2) + \frac{1}{2}d\log(2+\epsilon). \tag{24}$$

*Proof.* Let $t \geq d(d+1)/2$, $0 < \epsilon < 1$ and $p \in [P_{t,\epsilon}]^d$. Then, following steps from Faury et al. (2020) yields

$$\mathrm{Norm}_p = \int_{\lambda \in \mathbb{R}^d} \mathbb{1}\left\{\|\mathbf{Z}_{0,p}^{1/2}\lambda\| \leq \frac{1}{2}\right\} \exp\{-\|\lambda\|_{\mathbf{Z}_{0,p}}^2\}d\lambda$$

$$= \frac{1}{\sqrt{\det(\mathbf{Z}_{0,p})}} \int_{\lambda \in \mathbb{R}^d} \mathbb{1}\left\{\|\lambda\| \leq \frac{1}{2}\right\} \exp\{-\|\lambda\|^2\}d\lambda,$$

and

$$\mathrm{Norm}_{t,p} = \int_{\lambda \in \mathbb{R}^d} \mathbb{1}\left\{\|\mathbf{Z}_{0,p}^{1/2}\lambda\| \leq \frac{1}{4}\right\} \exp\{-\|\lambda\|_{\mathbf{Z}_{t,p}}^2\}d\lambda$$

$$= \frac{1}{\sqrt{\det(\mathbf{Z}_{t,p})}} \int_{\lambda \in \mathbb{R}^d} \mathbb{1}\left\{\|\mathbf{Z}_{0,p}^{1/2}\mathbf{Z}_{t,p}^{-1/2}\lambda\| \leq \frac{1}{4}\right\} \exp\{-\|\lambda\|^2\}d\lambda.$$

Noting that $\|\mathbf{Z}_{0,p}^{1/2}\mathbf{Z}_{t,p}^{-1/2}\| \leq 1$, we deduce

$$\mathrm{Norm}_{t,p} \geq \frac{1}{\sqrt{\det(\mathbf{Z}_{t,p})}} \int_{\mathbb{R}^d} \mathbb{1}\left\{\|\lambda\| \leq \frac{1}{4}\right\} \exp\{-\|\lambda\|^2\}d\lambda.$$

Therefore,

$$\frac{\mathrm{Norm}_p}{\mathrm{Norm}_{t,p}} \leq \sqrt{\frac{\det(\mathbf{Z}_{t,p})}{\det(\mathbf{Z}_{0,p})}} \frac{\int_{\mathbb{R}^d} \mathbb{1}\left\{\|\lambda\| \leq \frac{1}{2}\right\} \exp\{-\|\lambda\|^2\}d\lambda}{\int_{\mathbb{R}^d} \mathbb{1}\left\{\|\lambda\| \leq \frac{1}{4}\right\} \exp\{-\|\lambda\|^2\}d\lambda}.$$

We treat the integrals as

$$\frac{\int_{\mathbb{R}^d} \mathbb{1}\left\{\|\lambda\| \leq \frac{1}{2}\right\} \exp\{-\|\lambda\|^2\}d\lambda}{\int_{\mathbb{R}^d} \mathbb{1}\left\{\|\lambda\| \leq \frac{1}{4}\right\} \exp\{-\|\lambda\|^2\}d\lambda} = \frac{\int_{\mathbb{R}^d} \left(\mathbb{1}\left\{\|\lambda\| \leq \frac{1}{4}\right\} + \mathbb{1}\left\{\frac{1}{4} < \|\lambda\| \leq \frac{1}{2}\right\}\right) \exp\{-\|\lambda\|^2\}d\lambda}{\int_{\mathbb{R}^d} \mathbb{1}\left\{\|\lambda\| \leq \frac{1}{4}\right\} \exp\{-\|\lambda\|^2\}d\lambda}$$

$$= 1 + \frac{\int_{\mathbb{R}^d} \mathbb{1}\left\{\frac{1}{4} < \|\lambda\| \leq \frac{1}{2}\right\} \exp\{-\|\lambda\|^2\}d\lambda}{\int_{\mathbb{R}^d} \mathbb{1}\left\{\|\lambda\| \leq \frac{1}{4}\right\} \exp\{-\|\lambda\|^2\}d\lambda}$$

$$\leq 1 + \frac{\exp(-1/16)\int_{\mathbb{R}^d} \mathbb{1}\left\{\frac{1}{4} < \|\lambda\| \leq \frac{1}{2}\right\}d\lambda}{\exp(-1/16)\int_{\mathbb{R}^d} \mathbb{1}\left\{\|\lambda\| \leq \frac{1}{4}\right\}d\lambda}$$

$$= 2^d.$$

Thus

$$\log\left(\frac{\mathrm{Norm}_p}{\mathrm{Norm}_{t,p}}\right) \leq d\log(2) + \frac{1}{2}\log\left(\frac{\det(\mathbf{Z}_{t,p})}{\det(\mathbf{Z}_{0,p})}\right)$$

$$= d\log(2) + \frac{1}{2}\log\left(\det\left(\mathbf{I} + \mathbf{Z}_{0,p}^{-1/2}\mathbf{V}_t\mathbf{Z}_{0,p}^{-1/2}\right)\right)$$

$$\leq d\log(2) + \frac{1}{2}\log\left(\prod_{i\in[d]}\left(1 + \frac{n_{t,(i,i)}\mathbf{\Sigma}_{i,i}}{(1+\epsilon)^{p_i}\mathbf{\Sigma}_{i,i} + \|B\|}\right)\right)$$

$$\leq d\log(2) + \frac{1}{2}\log\left(\prod_{i\in[d]}\left(1 + \frac{n_{t,(i,i)}}{(1+\epsilon)^{p_i}}\right)\right).$$

In particular under event $(t \in \mathcal{D}_p)$,

$$\log\left(\frac{\mathrm{Norm}_p}{\mathrm{Norm}_{t,p}}\right) \leq d\log(2) + \frac{d}{2}\log(2+\epsilon).$$

$\square$

## C  Concentration of the covariances estimations (Proposition 2)

**Proposition 2.** *Let $t \geq d(d+1)/2$ and $\delta > 0$. Then, $\mathbb{P}(\mathcal{C}_t^c) \leq \delta/(t\log(t)^2)$.*

It is a direct application of the following proposition:

**Proposition 5.** *Let $\delta \in (0,1)$. Then with probability $1 - \delta$, for all $t \geq d(d+1)/2$ and $(i,j) \in [d]^2$ "reachable",*

$$|\hat{\chi}_{t,(i,j)} - \mathbf{\Sigma}_{i,j}| \leq \frac{B_i B_j}{4}\left(\frac{5h_{t,\delta}}{\sqrt{n_{t,(i,j)}}} + \frac{h_{t,\delta}^2}{n_{t,(i,j)}} + \frac{1}{n_{t,(i,j)}^2}\right).$$

*where $h_{t,\delta} = (1 + 2\log(1/\delta) + 2\log(d(d+1)) + \log(1+t))^{1/2}$.*

*Proof.* Let $\delta > 0$, $t \geq d(d+A)/2$. We remind

$$\mathcal{C}_t = \left\{\forall(i,j)\in[d]^2 \text{ "reachable"}, \ \hat{\mathbf{\Sigma}}_{t,(i,j)} \geq \mathbf{\Sigma}_{i,j}\right\}.$$

Let $(i,j) \in [d]^2$ "reachable". Then

$$\hat{\chi}_{t,(i,j)} = \hat{\mathbf{S}}_{t,(i,j)} - \hat{\mu}_{t,i}\hat{\mu}_{t,j}$$

$$= \frac{1}{n_{t,(i,j)}}\sum_{s=1}^{t}A_{s,i}A_{s,j}Y_{s,i}Y_{s,j} - \left(\frac{1}{n_{t,i}}\sum_{s=1}^{t}A_{s,i}Y_{s,i}\right)\left(\frac{1}{n_{t,i}}\sum_{s=1}^{t}A_{s,j}Y_{s,j}\right),$$

And,

$$\hat{\chi}_{t,(i,j)} - \mathbf{\Sigma}_{i,j} = \frac{1}{n_{t,(i,j)}}\sum_{s=1}^{t}A_{s,i}A_{s,j}Y_{s,i}Y_{s,j} - \mathbf{S}_{i,j} - \left[\left(\frac{1}{n_{t,i}}\sum_{s=1}^{t}A_{s,i}Y_{s,i}\right)\left(\frac{1}{n_{t,j}}\sum_{s=1}^{t}A_{s,j}Y_{s,j}\right) - \mu_i\mu_j\right]$$

$$= \frac{1}{n_{t,(i,j)}}\sum_{s=1}^{t}A_{s,i}A_{s,j}\left[Y_{s,i}Y_{s,j} - \mathbf{S}_{i,j}\right] - \left[\left(\frac{1}{n_{t,i}}\sum_{s=1}^{t}A_{s,i}\left[Y_{s,i} - \mu_i\right]\right)\left(\frac{1}{n_{t,j}}\sum_{s=1}^{t}A_{s,j}\left[Y_{s,j} - \mu_j\right]\right)\right.$$

$$\left. + \mu_j\left(\frac{1}{n_{t,i}}\sum_{s=1}^{t}A_{s,i}\left[Y_{s,i} - \mu_i\right]\right) + \mu_i\left(\frac{1}{n_{t,j}}\sum_{s=1}^{t}A_{s,j}\left[Y_{s,j} - \mu_j\right]\right)\right].$$

A triangle inequality yields

$$|\hat{\chi}_{t,(i,j)} - \boldsymbol{\Sigma}_{i,j}| \leq \left|\frac{1}{n_{t,(i,j)}}\sum_{s=1}^{t} A_{s,i}A_{s,j}\Big[Y_{s,i}Y_{s,j} - \mathbf{S}_{i,j}\Big]\right| + \left|\frac{1}{n_{t,i}}\sum_{s=1}^{t} A_{s,i}\Big[Y_{s,i} - \mu_i\Big]\right|\left|\frac{1}{n_{t,j}}\sum_{s=1}^{t} A_{s,j}\Big[Y_{s,j} - \mu_j\Big]\right|$$

$$+ \frac{B_j}{2}\left|\frac{1}{n_{t,i}}\sum_{s=1}^{t} A_{s,i}\Big[Y_{s,i} - \mu_i\Big]\right| + \frac{B_i}{2}\left|\frac{1}{n_{t,j}}\sum_{s=1}^{t} A_{s,j}\Big[Y_{s,j} - \mu_j\Big]\right|.$$

We make repeated use of the following Lemma

**Lemma 3.** *Let* $(\mathcal{H}_t)_{t\in\mathbb{N}^*}$ *be a filtration,* $(U_t)_{t\in\mathbb{N}^*}$ *be an* $\mathcal{H}_t$ *adapted martingales bounded by* $C \in \mathbb{R}_+^*$ *with* $\mathbb{E}[U_1] = 0$*, and* $(\mathbb{1}\{V_t\})_{t\in\mathbb{N}^*}$ *be a predictable process and* $\delta > 0$*.*

*Then with probability at least* $1 - \delta$*, for all* $t$

$$\mathbb{P}\left(\frac{\sum_{s=1}^{t} \mathbb{1}\{V_s\}U_s}{1 + \sum_{s=1}^{t} \mathbb{1}\{V_s\}} > \frac{C}{\sqrt{1 + \sum_{s=1}^{t} \mathbb{1}\{V_s\}}}\sqrt{2\log(1/\delta) + \log\left(1 + \sum_{s=1}^{t} \mathbb{1}\{V_s\}\right)}\right) \leq \delta.$$

Therefore, with probability at least $1 - \delta/2$, for all $(i,j)$ and $t$,

$$\left|\frac{1}{n_{t,(i,j)}}\sum_{s=1}^{t} A_{s,i}A_{s,j}\Big[Y_{s,i}Y_{s,j} - \mathbf{S}_{i,j}\Big]\right| \leq \left|\frac{1}{n_{t,(i,j)}+1}\sum_{s=1}^{t} A_{s,i}A_{s,j}\Big[Y_{s,i}Y_{s,j} - \mathbf{S}_{i,j}\Big]\right| + \frac{B_iB_j}{4(n_{t,(i,j)}+1)}$$

$$\leq \frac{B_iB_j}{4}\frac{1}{\sqrt{n_{t,(i,j)}+1}}\sqrt{2\log(1/\delta) + 2\log(d(d+1)) + \log(1+t)}$$

$$+ \frac{B_iB_j}{4n_{t,(i,j)}}$$

$$\leq \frac{B_iB_j}{4\sqrt{n_{t,(i,j)}}}\sqrt{2\log(1/\delta) + 2\log(d(d+1)) + \log(1+t)}$$

$$+ \frac{B_iB_j}{4n_{t,(i,j)}}.$$

With probability at least $1 - \delta/2$, for all $i$ and $t$,

$$\left|\frac{1}{n_{t,i}}\sum_{s=1}^{t} A_{s,i}\Big[Y_{s,i} - \mu_i\Big]\right| \leq \left|\frac{1}{n_{t,i}+1}\sum_{s=1}^{t} A_{s,i}\Big[Y_{s,i} - \mu_i\Big]\right| + \frac{B_i}{2n_{t,(i,i)}}$$

$$\leq \frac{B_i}{2\sqrt{n_{t,(i,i)}}}\sqrt{2\log(1/\delta) + 2\log(2d) + \log(1+t)} + \frac{B_i}{2n_{t,(i,i)}}.$$

Therefore, reinjecting those expressions yields that with probability at least $1 - \delta$, for all $(i,j)$ and $t$,

$$|\hat{\chi}_{t,(i,j)} - \boldsymbol{\Sigma}_{i,j}| \leq \frac{B_iB_j}{4\sqrt{n_{t,(i,j)}}}\sqrt{2\log(1/\delta) + 2\log(d(d+1)) + \log(1+t)} + \frac{B_iB_j}{4n_{t,(i,j)}}$$

$$+ \frac{B_iB_j\big(2\log(1/\delta) + 2\log(2d) + \log(1+t)\big)}{4\sqrt{n_{t,(i,i)}n_{t,(j,j)}}} + \frac{B_iB_j}{4n_{t,(i,i)}n_{t,(j,j)}}$$

$$+ \frac{B_iB_j}{4}\left(\frac{1}{n_{t,(j,j)}\sqrt{n_{t,(i,i)}}} + \frac{1}{n_{t,(i,i)}\sqrt{n_{t,(j,j)}}}\right)\sqrt{2\log(1/\delta) + 2\log(2d) + \log(1+t)}$$

$$+ \frac{B_iB_j}{4}\left(\frac{1}{\sqrt{n_{t,(i,i)}}} + \frac{1}{\sqrt{n_{t,(j,j)}}}\right)\sqrt{2\log(1/\delta) + 2\log(2d) + \log(1+t)}$$

$$\leq \frac{B_iB_j}{4\sqrt{n_{t,(i,j)}}}\sqrt{2\log(1/\delta) + 2\log(d(d+1)) + \log(1+t)} + \frac{B_iB_j}{4n_{t,(i,j)}}$$

$$+ \frac{B_iB_j\big(2\log(1/\delta) + 2\log(2d) + \log(1+t)\big)}{4\sqrt{n_{t,(i,i)}n_{t,(j,j)}}} + \frac{B_iB_j}{4n_{t,(i,i)}n_{t,(j,j)}}$$

$$+ \frac{B_iB_j}{2}\left(\frac{1}{\sqrt{n_{t,(i,i)}}} + \frac{1}{\sqrt{n_{t,(j,j)}}}\right)\sqrt{2\log(1/\delta) + 2\log(2d) + \log(1+t)}.$$

To simplify this expression, using $n_{t,(i,j)} \leq \min\{n_{t,(i,i)}, n_{t,(j,j)}\}$ and $n_{t,(i,j)} \leq \sqrt{n_{t,(i,i)}n_{t,(j,j)}}$ yields

$$|\hat{\chi}_{t,(i,j)} - \mathbf{\Sigma}_{i,j}| \leq 5\frac{B_i B_j}{4}\frac{1}{\sqrt{n_{t,(i,j)}}}\sqrt{2\log(1/\delta) + 2\log(d(d+1)) + \log(1+t)}$$

$$+ \frac{B_i B_j}{4}\frac{1}{n_{t,(i,j)}}\Big(1 + 2\log(1/\delta) + 2\log(d(d+1)) + \log(1+t)\Big)$$

$$+ \frac{B_i B_j}{4}\frac{1}{n_{t,(i,j)}^2}.$$

Denoting $h_{t,\delta} = \Big(1 + 2\log(1/\delta) + 2\log\big(t\log(t)^2 d(d+1)\big) + \log(1+t)\Big)^{1/2}$, we have with probability at least $1 - \frac{2\delta}{d(d+1)t\log(t)^2}$

$$|\hat{\chi}_{t,(i,j)} - \mathbf{\Sigma}_{i,j}| \leq \frac{B_i B_j}{4}\left(\frac{5h_{t,\delta}}{\sqrt{n_{t,(i,j)}}} + \frac{h_{t,\delta}^2}{n_{t,(i,j)}} + \frac{1}{n_{t,(i,j)}^2}\right).$$

A union bound yields the desired results. $\qquad\qquad\qquad\qquad\qquad\qquad\qquad\qquad\square$

## C.1 Proof for Lemma 3

**Lemma 3.** *Let $(\mathcal{H}_t)_{t\in\mathbb{N}^*}$ be a filtration, $(U_t)_{t\in\mathbb{N}^*}$ be an $\mathcal{H}_t$ adapted martingales bounded by $C \in \mathbb{R}_+^*$ with $\mathbb{E}[U_1] = 0$, and $(\mathbb{1}\{V_t\})_{t\in\mathbb{N}^*}$ be a predictable process and $\delta > 0$.*

*Then with probability at least $1 - \delta$, for all $t$*

$$\mathbb{P}\left(\frac{\sum_{s=1}^t \mathbb{1}\{V_s\}U_s}{1 + \sum_{s=1}^t \mathbb{1}\{V_s\}} > \frac{C}{\sqrt{1 + \sum_{s=1}^t \mathbb{1}\{V_s\}}}\sqrt{2\log(1/\delta) + \log(1 + \sum_{s=1}^t \mathbb{1}\{V_s\})}\right) \leq \delta.$$

*Proof.* Let $t \geq 2$. Then $U_t$ is $C$ sub-Gaussian and for all $\lambda \in$

$$\mathbb{E}\left[\exp\left(\lambda\mathbb{1}\{V_t\}U_t - \frac{\lambda^2 C^2}{2}\mathbb{1}\{V_t\}\right)\Big|\mathcal{H}_{t-1}\right] \leq 1$$

Then $(W_t(\lambda))_{t\in\mathbb{N}^*} = \left(\exp(\lambda\sum_{s=1}^t \mathbb{1}\{V_s\}U_s - \frac{\lambda^2 C^2}{2}\sum_{s=1}^t \mathbb{1}\{V_s\})\right)_{t\in\mathbb{N}^*}$ is a supermatringale. We use the Method of Mixtures by integrating for a $\lambda \sim \mathcal{N}(0, 1/C^2)$. This yield

$$\int_{\lambda\in\mathbb{R}} \frac{C}{\sqrt{2\pi}}\exp\left(-\frac{\lambda^2 C^2}{2}\right)W_t(\lambda)d\lambda$$

$$= \frac{C}{\sqrt{2\pi}}\int_{\lambda\in\mathbb{R}}\exp\left(\lambda\sum_{s=1}^t \mathbb{1}\{V_s\}U_s - \frac{\lambda^2 C^2}{2}(1 + \sum_{s=1}^t \mathbb{1}\{V_s\})\right)d\lambda$$

$$= \frac{C}{\sqrt{2\pi}}\int_{\lambda\in\mathbb{R}}\exp\left(\frac{\left(\sum_{s=1}^t \mathbb{1}\{V_s\}U_s\right)^2}{2C^2(1 + \sum_{s=1}^t \mathbb{1}\{V_s\})}\right.$$

$$\left. - \frac{1}{2}\Big(\lambda - \frac{\sum_{s=1}^t \mathbb{1}\{V_s\}U_s}{C^2(1 + \sum_{s=1}^t \mathbb{1}\{V_s\})}\Big)^2 C^2(1 + \sum_{s=1}^t \mathbb{1}\{V_s\})\right)d\lambda$$

$$= \exp\left(\frac{\left(\sum_{s=1}^t \mathbb{1}\{V_s\}U_s\right)^2}{2C^2(1 + \sum_{s=1}^t \mathbb{1}\{V_s\})}\right)\frac{1}{\sqrt{1 + \sum_{s=1}^t \mathbb{1}\{V_s\}}}$$

$$\leq 1.$$

Therefore,

$$\mathbb{P}\left(\frac{\sum_{s=1}^t \mathbb{1}\{V_s\}U_s}{1 + \sum_{s=1}^t \mathbb{1}\{V_s\}} > \frac{C}{\sqrt{1 + \sum_{s=1}^t \mathbb{1}\{V_s\}}}\sqrt{2\log(1/\delta) + \log(1 + \sum_{s=1}^t \mathbb{1}\{V_s\})}\right) \leq \delta.$$

Using the stopping time construction from Abbasi-Yadkori et al. (2011) yields the property for all $t$. $\qquad\square$

## D  Behaviour in the high-probability events (Section 5)

The following proposition states that under some assumptions on the sequence of events $(\mathcal{E}_t)$, the regret can be bounded by problem-dependent quantities (including $\Sigma$, $T$, or $d$). They are not all explicitly stated in Proposition 6 to make it adaptive to both algorithms but are hidden in the constants.

**Proposition 6.** *Let $r \in \mathbb{N}$, $e \in (1, +\infty)^r$. Let $(\mathcal{E}_t)_{t \geq d(d+1)/2}$ be a sequence of events such that for all $t \geq d(d+1)/2$, under $\mathcal{E}_t$,*

$$\frac{\Delta_{A_{t+1}}^2}{C} \leq \sum_{i \in A_{t+1}} \frac{C_{A_{t+1},i}}{n_{t,(i,j)}} + \sum_{s \in [r]} \left[ \sum_{(i,j) \in A_{t+1}} \frac{C_s}{n_{t,(i,j)}^{e_s}} \right], \tag{25}$$

*where $C$ and $(C_s)_{s \in [r]}$ are problem-dependent positive constants. $C_{A_{t+1},i}$ is a positive constant depending on $A_{t+1}$ and $i$ so that, for all $a \in \mathcal{A}$, $C_{a,i} \leq 2m\Sigma_{i,i}$. Let $c \in \mathbb{R}_+^*$ and $(c_s)_{s \in [r]} \in (\mathbb{R}_+^*)^r$ be positive constants such that $1/c + \sum_{s \in [r]} 1/c_s = 1$.*

*Then,*

$$\sum_{t=d(d+1)/2}^{T-1} \Delta_{A_{t+1}} \mathbb{1}\{\mathcal{E}_t\}$$

$$\leq 96 c_1 C \log(m)^2 \sum_{i \in [d]} \left( \max_{a \in \mathcal{A}/i \in a} \frac{C_{a,i}}{\Delta_a} \right)$$

$$+ \sum_{s=1}^{r} \left[ \mathbb{1}\{e_s = 2\} 346 \left( c_s C C_s \log(m) \right)^{1/2} m d^2 \left( 1 + \log\left( \frac{\Delta_{\max}}{\Delta_{\min}} \right) \right) \right.$$

$$+ \mathbb{1}\{1 < e_s < 2\} 60.30^{1/e_s} \left( c_s C C_s \log(m) \right)^{1/e_s} d^2 m^{2/e_s} \Delta_{\min}^{1-2/e_s}$$

$$\left. + \mathbb{1}\{2 < e_s\} 60.30^{1/e_s} \left( c_s C C_s \log(m) \right)^{1/e_s} \frac{e_s}{e_s - 2} d^2 m^{2/e_s} \Delta_{\max}^{1-2/e_s} \right]. \tag{26}$$

*where $(\alpha_k)_{k \in \mathbb{N}^*}$, $(\beta_k)_{k \in \mathbb{N}^*}$ and $k_0 \in \mathbb{N}^*$ are defined in Appendix D.1.*

*Proof.* The proof is classical and involves a decomposition of the events $\mathcal{E}_t$ (see Kveton et al. (2015); Degenne and Perchet (2016); Perrault et al. (2020b)). By considering each of the $r$ sub-sum in Eq. (25) and designing sets of event that can happen only a finite number of times.

We introduce two sequences $(\alpha_k)_{k \in \mathbb{N}^*}$ and $(\beta_k)_{k \in \mathbb{N}^*}$, both begin at 1 and strictly decrease to 0 (see Appendix D.1 for their definitions). These sequences are introduced to be able to consider the different terms of Eq. (25) separately.

Let $(c_s)_{s \in [r]} \in (\mathbb{R}_+^*)^r$ such that $\sum_{s \in [r]} 1/c_s = 1$.

Let $t \geq d(d+1)/2$, $k \in \mathbb{N}^*$. We define the set

$$S_{t,k} = \left\{ i \in A_{t+1}, \quad n_{t,(i,i)} \leq c_1 m \alpha_k \frac{C}{\Delta_{A_{t+1}}^2} \frac{C_{A_{t+1},i}^2}{\Sigma_{i,i}^*} \right\}, \tag{27}$$

and the event

$$\mathbb{A}_{t,k} = \left\{ \sum_{i \in S_{t,k}} \frac{\Sigma_{i,i}^*}{C_{A_{t+1},i}} \geq \beta_k m; \quad \forall l < k, \sum_{i \in S_{t,l}^1} \frac{\Sigma_{i,i}}{C_{A_{t+1},i}} < \beta_l m \right\}. \tag{28}$$

A notable difference from previous approaches is the use of $\Sigma_{i,i}^*/C_{a,i}$ in $\mathbb{A}_{t,k}^1$ instead of set cardinals. This enables the explicit appearance of the $C_{a,i}$ coefficients, which will involve the $\sigma_{a,i}^2$ for the application of this proposition to our algorithms.

For $s \in [r]$, we define

$$S_{t,k}^s = \left\{ (i,j) \in A_{t+1}, \quad n_{t,(i,j)}^{e_s} \leq c_s m^2 \alpha_k \frac{C}{\Delta_{A_{t+1}}^2} C_s \right\} \tag{29}$$

and the events

$$\mathbb{A}_{t,k}^s = \left\{ |S_{t,k}^s| \geq \beta_k m^2; \quad \forall l < k, |S_{t,l}^s| < \beta_l m^2 \right\}. \tag{30}$$

The following Lemma, proven in Appendix D.2, decomposes $(\mathcal{E}_t)_{t \geq d(d+1)/2}$ using these events.

**Lemma 4.** *Let's consider the assumptions of Proposition 6. Let $\mathbb{A}_{t,k}$ and $(\mathbb{A}_{t,k}^s)_{s \in [r]}$ be the events defined in Eq.* (28) *and Eq.* (30). *Let $k_0 \in \mathbb{N}^*$ such that $0 < m\beta_{k_0} < \frac{1}{2m}$ and $t \geq d(d+1)/2$.*

$$\mathbb{1}\{\mathcal{E}_t\} \leq \sum_{k=1}^{k_0} \mathbb{1}\{\mathbb{A}_{t,k}\} + \sum_{s=1}^{r} \sum_{k=1}^{k_0} \mathbb{1}\{\mathbb{A}_{t,k}^s\}.$$

Using it, we decompose

$$\sum_{t=d(d+1)/2}^{T-1} \Delta_{A_{t+1}} \mathbb{1}\{\mathcal{E}_t\} \leq \sum_{t=d(d+1)/2}^{T-1} \left[ \sum_{k=1}^{k_0} \Delta_{A_{t+1}} \mathbb{1}\{\mathbb{A}_{t,k}\} + \sum_{s=1}^{r} \sum_{k=1}^{k_0} \Delta_{A_{t+1}} \mathbb{1}\{\mathbb{A}_{t,k}^s\} \right]$$

$$= \sum_{t=d(d+1)/2}^{T-1} \left[ \Delta_{A_{t+1}} \sum_{k=1}^{k_0} \mathbb{1}\{\mathbb{A}_{t,k}\} \right] + \sum_{s=1}^{r} \sum_{t=d(d+1)/2}^{T-1} \left[ \Delta_{A_{t+1}} \sum_{k=1}^{k_0} \mathbb{1}\{\mathbb{A}_{t,k}^s\} \right]. \tag{31}$$

We begin with the first term of Eq. (31). Let $t \geq d(d+1)/2$, and $k \in [k_0]$. Then,

$$\mathbb{A}_{t,k} = \left\{ \sum_{i \in S_{t,k}^1} \frac{\boldsymbol{\Sigma}_{i,i}}{C_{A_{t+1},i}} \geq \beta_k m; \quad \forall l < k, \sum_{i \in S_{t,l}^1} \frac{\boldsymbol{\Sigma}_{i,i}}{C_{A_{t+1},i}} < \beta_l \right\} \subseteq \left\{ \frac{1}{\beta_k m} \sum_{i \in S_{t,k}^1} \frac{\boldsymbol{\Sigma}_{i,i}}{C_{A_{t+1},i}} \geq 1 \right\}.$$

Therefore,

$$\mathbb{1}\{\mathbb{A}_{t,k}\} \leq \frac{1}{\beta_k m} \sum_{i \in [d]} \frac{\boldsymbol{\Sigma}_{i,i}}{C_{A_{t+1},i}} \mathbb{1}\left\{ \mathbb{A}_{t,k} \cap \{i \in S_{t,k}\} \right\}. \tag{32}$$

Summing over $t$ and $k$, and including the gaps yields

$$\sum_{t=d(d+1)/2}^{T-1} \Delta_{A_{t+1}} \sum_{k=1}^{k_0} \mathbb{1}\{\mathbb{A}_{t,k}^1\} \tag{33}$$

$$\leq \sum_{t=d(d+1)/2}^{T} \Delta_{A_{t+1}} \sum_{k=1}^{k_0} \frac{1}{\beta_k m} \sum_{i \in [d]} \frac{\boldsymbol{\Sigma}_{i,i}}{C_{A_{t+1},i}} \mathbb{1}\left\{ \mathbb{A}_{t,k}^1 \cap \{i \in S_{t,k}\} \right\} \leftarrow \text{ by Eq. (32)}$$

$$\leq \sum_{i \in [d]} \boldsymbol{\Sigma}_{i,i} \sum_{t=d(d+1)/2}^{T} \sum_{k=1}^{k_0} \frac{1}{\beta_k m} \frac{\Delta_{A_{t+1}}}{C_{A_{t+1},i}} \mathbb{1}\{i \in S_{t,k}\}$$

$$= \sum_{i \in [d]} \boldsymbol{\Sigma}_{i,i} \sum_{k=1}^{k_0} \frac{1}{\beta_k m} \sum_{t=d(d+1)/2}^{T} \frac{\Delta_{A_{t+1}}}{C_{A_{t+1},i}} \mathbb{1}\left\{ n_{t,(i,i)} \leq c_1 m \alpha_k \frac{C}{\left(\frac{\Delta_{A_{t+1}}}{C_{A_{t+1},i}}\right)^2 \boldsymbol{\Sigma}_{i,i}} \right\}. \leftarrow \text{ by Eq. (27)} \tag{34}$$

Let $i \in [d]$, we consider all the actions associated to it. Let $q_i \in \mathbb{N}^*$ be the number of actions associated to item $i$. Let $l \in [q_i]$, we denote $e_i^l \in \mathcal{A}$ the $l$-th action associated to item $i$, sorted by decreasing $\frac{\Delta_{e_i^l}}{C_{e_i^l,i}}$, with $\frac{C_{e_i^0,i}}{\Delta_{e_i^0}} = 0$ by convention. Then

$$\sum_{t=d(d+1)/2}^{T-1} \frac{\Delta_{A_{t+1}}}{C_{A_{t+1},i}} \mathbb{1}\left\{ n_{t,(i,i)} \leq c_1 m\alpha_k \frac{C}{\left(\frac{\Delta_{A_{t+1}}}{C_{A_{t+1},i}}\right)^2 \boldsymbol{\Sigma}_{i,i}} \right\}$$

$$\leq \sum_{t=0}^{T-1} \sum_{l=1}^{q_i} \frac{\Delta_{e_i^l}}{C_{e_i^l,i}} \mathbb{1}\left\{ n_{t,(i,i)} \leq c_1 m\alpha_k \frac{C}{\left(\frac{\Delta_{e_i^l}}{C_{e_i^l,i}}\right)^2 \boldsymbol{\Sigma}_{i,i}}, \quad A_{t+1} = e_i^l \right\}$$

$$= \sum_{t=0}^{T-1} \sum_{l=1}^{q_i} \frac{\Delta_{e_i^l}}{C_{e_i^l,i}} \mathbb{1}\left\{ n_{t,(i,i)} \frac{\boldsymbol{\Sigma}_{i,i}}{c_1 m\alpha_k C} \leq \frac{1}{\left(\frac{\Delta_{e_i^l}}{C_{e_i^l,i}}\right)^2}, \quad A_{t+1} = e_i^l \right\}$$

$$= \sum_{t=0}^{T-1} \sum_{l=1}^{q_i} \frac{\Delta_{e_i^l}}{C_{e_i^l,i}} \sum_{p=1}^{l} \mathbb{1}\left\{ \frac{1}{\left(\frac{\Delta_{e_i^{p-1}}}{C_{e_i^{p-1},i}^2}\right)^2} < n_{t,(i,i)} \frac{\boldsymbol{\Sigma}_{i,i}}{c_1 m\alpha_k C} \leq \frac{1}{\left(\frac{\Delta_{e_i^p}}{C_{e_i^p,i}^2}\right)}, \quad A_{t+1} = e_i^l \right\} \leftarrow \text{decomposing the event}$$

$$\leq \sum_{t=0}^{T-1} \sum_{l=1}^{q_i} \sum_{p=1}^{l} \frac{\Delta_{e_i^p}}{C_{e_i^p,i}} \mathbb{1}\left\{ \frac{1}{\left(\frac{\Delta_{e_i^{p-1}}}{C_{e_i^{p-1},i}^2}\right)^2} < n_{t,(i,i)} \frac{\boldsymbol{\Sigma}_{i,i}}{c_1 m\alpha_k C} \leq \frac{1}{\left(\frac{\Delta_{e_i^p}}{C_{e_i^p,i}^2}\right)}, \quad A_{t+1} = e_i^l \right\} \leftarrow \text{as } \frac{\Delta_{e_i^l}}{C_{e_i^l,i}} \leq \frac{\Delta_{e_i^p}}{C_{e_i^p,i}}$$

$$= \sum_{p=1}^{q_i} \frac{\Delta_{e_i^p}}{C_{e_i^p,i}} \sum_{t=0}^{T-1} \sum_{l=p}^{q_i} \mathbb{1}\left\{ \frac{1}{\left(\frac{\Delta_{e_i^{p-1}}}{C_{e_i^{p-1},i}^2}\right)^2} < n_{t,(i,i)} \frac{\boldsymbol{\Sigma}_{i,i}}{c_1 m\alpha_k C} \leq \frac{1}{\left(\frac{\Delta_{e_i^p}}{C_{e_i^p,i}^2}\right)}, \quad A_{t+1} = e_i^l \right\}$$

$$\leq \sum_{p=1}^{q_i} \frac{\Delta_{e_i^p}}{C_{e_i^p,i}} \sum_{t=0}^{T-1} \sum_{l=1}^{q_i} \mathbb{1}\left\{ \frac{1}{\left(\frac{\Delta_{e_i^{p-1}}}{C_{e_i^{p-1},i}^2}\right)^2} < n_{t,(i,i)} \frac{\boldsymbol{\Sigma}_{i,i}}{c_1 m\alpha_k C} \leq \frac{1}{\left(\frac{\Delta_{e_i^p}}{C_{e_i^p,i}^2}\right)}, \quad A_{t+1} = e_i^l \right\} \leftarrow \text{we extend the sum over } l$$

$$= \sum_{p=1}^{q_i} \frac{\Delta_{e_i^p}}{C_{e_i^p,i}} \sum_{t=0}^{T-1} \mathbb{1}\left\{ \frac{1}{\left(\frac{\Delta_{e_i^{p-1}}}{C_{e_i^{p-1},i}^2}\right)^2} < n_{t,(i,i)} \frac{\boldsymbol{\Sigma}_{i,i}}{c_1 m\alpha_k C} \leq \frac{1}{\left(\frac{\Delta_{e_i^p}}{C_{e_i^p,i}}\right)^2}, \quad i \in A_{t+1} \right\} \leftarrow \text{we simplify the inner sum}$$

$$\leq \sum_{p=1}^{q_i} \frac{\Delta_{e_i^p}}{C_{e_i^p,i}} \left( \left\lfloor \left(\frac{C_{e_i^p,i}}{\Delta_{e_i^p}}\right)^2 \frac{c_1 m\alpha_k C}{\boldsymbol{\Sigma}_{i,i}} \right\rfloor - \left\lfloor \left(\frac{C_{e_i^{p-1},i}}{\Delta_{e_i^{p-1}}}\right)^2 \frac{c_1 m\alpha_k C}{\boldsymbol{\Sigma}_{i,i}} \right\rfloor \right) \leftarrow \text{the event can only happen a given nbr. of times}$$

$$= \left( \left\lfloor \left(\frac{C_{e_i^{q_i},i}}{\Delta_{e_i^{q_i}}}\right)^2 \frac{c_1 m\alpha_k C}{\boldsymbol{\Sigma}_{i,i}} \right\rfloor \frac{\Delta_{e_i^{q_i}}}{C_{e_i^{q_i},i}} + \sum_{p=1}^{q_i-1} \left\lfloor \left(\frac{C_{e_i^p,i}}{\Delta_{e_i^p}}\right)^2 \frac{c_1 m\alpha_k C}{\boldsymbol{\Sigma}_{i,i}} \right\rfloor \left(\frac{\Delta_{e_i^p}}{C_{e_i^p,i}} - \frac{\Delta_{e_i^{p+1}}}{C_{e_i^{p+1},i}}\right) \right) \leftarrow \text{summation by parts}$$

$$\leq \frac{c_1 m\alpha_k C}{\boldsymbol{\Sigma}_{i,i}} \left( \frac{C_{e_i^{q_i},i}}{\Delta_{e_i^{q_i}}} + \sum_{p=1}^{q_i-1} \left(\frac{C_{e_i^p,i}}{\Delta_{e_i^p}}\right)^2 \left(\frac{\Delta_{e_i^p}}{C_{e_i^p,i}} - \frac{\Delta_{e_i^{p+1}}}{C_{e_i^{p+1},i}}\right) \right) \leftarrow \text{everything is positive}$$

$$\leq \frac{c_1 m\alpha_k C}{\boldsymbol{\Sigma}_{i,i}} \left( \frac{C_{e_i^{q_i},i}}{\Delta_{e_i^{q_i}}} + \int_{\left(\frac{\Delta_{e_i^{q_i}}}{C_{e_i^{q_i},i}}\right)}^{\left(\frac{\Delta_{e_i^1}}{C_{e_i^1,i}}\right)} \frac{1}{x^2} dx \right)$$

$$= \frac{c_1 m\alpha_k C}{\boldsymbol{\Sigma}_{i,i}} \left( \frac{C_{e_i^{q_i},i}^2}{\Delta_{e_i^{q_i}}} + \frac{C_{e_i^{q_i}}^2}{\Delta_{e_i^{q_i}}} - \frac{C_{e_i^1}^2}{\Delta_{e_i^1}} \right)$$

$$\leq \frac{2c_1 m\alpha_k C}{\boldsymbol{\Sigma}_{i,i}} \frac{C_{e_i^{q_i},i}}{\Delta_{e_i^{q_i}}}$$

$$\leq \frac{2c_1 m\alpha_k C}{\boldsymbol{\Sigma}_{i,i}} \left( \max_{a \in \mathcal{A}/i \in a} \frac{C_{a,i}}{\Delta_a} \right). \tag{35}$$

Reinjecting Eq. (35) into Eq. (34) yields

$$\sum_{t=d(d+1)/2}^{T-1} \Delta_{A_{t+1}} \sum_{k=1}^{k_0} \mathbb{1}\{\mathbb{A}_{t,k}^1\} \le \sum_{i\in[d]} \Sigma_{i,i} \sum_{k=1}^{k_0} \frac{1}{\beta_k m} \frac{2c_1 m C \alpha_k}{\Sigma_{i,i}} \Big( \max_{a\in\mathcal{A}/i\in a} \frac{C_{a,i}}{\Delta_a} \Big)$$

$$= 2c_1 C \Big( \sum_{k=1}^{k_0} \frac{\alpha_k}{\beta_k} \Big) \sum_{i\in[d]} \Big( \max_{a\in\mathcal{A}/i\in a} \frac{C_{a,i}}{\Delta_a} \Big)$$

$$= 96 c_1 C \log(m)^2 \sum_{i\in[d]} \Big( \max_{a\in\mathcal{A}/i\in a} \frac{C_{a,i}}{\Delta_a} \Big). \tag{36}$$

We treat the $r$ other terms in a similar way. Let $s \in [r]$, $t \ge d(d+1)/2$, and $k \in [k_0]$,

$$\mathbb{A}_{t,k}^s = \Big\{ |S_{t,k}^s| \ge \beta_k m^2; \quad \forall l < k, |S_{t,l}^s| < \beta_l m^2 \Big\} \subseteq \Big\{ \frac{1}{\beta_k m^2} |S_{t,k}^s| \ge 1 \Big\}.$$

Therefore,

$$\mathbb{1}\{\mathbb{A}_{t,k}^s\} \le \frac{1}{\beta_k m^2} \sum_{(i,j)\in[d]^2} \mathbb{1}\Big\{ \mathbb{A}_{t,k}^s \cap \{(i,j) \in S_{t,k}^s\} \Big\}. \tag{37}$$

Summing over $t$ and $k$ yields

$$\sum_{t=d(d+1)/2}^{T-1} \Delta_{A_{t+1}} \sum_{k=1}^{k_0} \mathbb{1}\{\mathbb{A}_{t,k}^s\}$$

$$\le \sum_{t=d(d+1)/2}^{T-1} \Delta_{A_{t+1}} \sum_{k=1}^{k_0} \frac{1}{\beta_k m^2} \sum_{(i,j)\in[d]^2} \mathbb{1}\Big\{ \mathbb{A}_{t,k}^s \cap \{(i,j) \in S_{t,k}^s\} \Big\} \leftarrow \text{ by Eq. (37)}$$

$$\le \sum_{(i,j)\in[d]^2} \sum_{t=d(d+1)/2}^{T-1} \sum_{k=1}^{k_0} \frac{1}{\beta_k m^2} \Delta_{A_{t+1}} \mathbb{1}\{(i,j) \in S_{t,k}^s\}$$

$$= \sum_{(i,j)\in[d]^2} \sum_{k=1}^{k_0} \frac{1}{\beta_k m^2} \sum_{t=d(d+1)/2}^{T-1} \Delta_{A_{t+1}} \mathbb{1}\Big\{ n_{t,(i,j)} \le m^{2/e_s} (c_s \alpha_k C C_s)^{1/e_s} \frac{1}{\Delta_{A_{t+1}}^{2/e_s}} \Big\} \cdot \leftarrow \text{ by Eq. (29)} \tag{38}$$

Let $(i,j) \in [d]^2$, we consider all the actions which are associated to it. Let $q_{(i,j)} \in \mathbb{N}^*$ be the number of actions associated to the tuple $(i,j)$. Let $l \in [q_{(i,j)}]$, this time, we denote $e_{(i,j)}^l \in \mathcal{A}$ the $l$-th action associated to tuple $(i,j)$, sorted by decreasing $\Delta_{e_{(i,j)}^l}$, with $\frac{1}{\Delta_{e_{(i,j)}^0}} = 0$ by convention. Then,

$$\sum_{t=d(d+1)/2}^{T-1} \Delta_{A_{t+1}} \mathbb{1}\Big\{ n_{t,(i,j)} \le m^{2/e_s} (c_s \alpha_k C C_s)^{1/e_s} \frac{1}{\Delta_{A_{t+1}}^{2/e_s}} \Big\}$$

$$\le \sum_{t=0}^{T-1} \sum_{l=1}^{q_{(i,j)}} \Delta_{e_{(i,j)}^l} \mathbb{1}\Big\{ n_{t,(i,j)} \le m^{2/e_s} (c_s \alpha_k C C_s)^{1/e_s} \frac{1}{\Delta_{A_{t+1}}^{2/e_s}}, \quad A_{t+1} = e_{(i,j)}^l \Big\}$$

$$= \sum_{t=0}^{T-1} \sum_{l=1}^{q_{(i,j)}} \Delta_{e_{(i,j)}^l} \mathbb{1}\Big\{ n_{t,(i,j)} m^{-2/e_s} (c_s \alpha_k C C_s)^{-1/e_s} \le \frac{1}{\Delta_{e_{(i,j)}^l}^{2/e_s}}, \quad A_{t+1} = e_{(i,j)}^l \Big\}$$

$$= \sum_{t=0}^{T-1} \sum_{l=1}^{q_{(i,j)}} \Delta_{e_{(i,j)}^l} \sum_{p=1}^{l} \mathbb{1}\Big\{ \frac{1}{\Delta_{e_{(i,j)}^{p-1}}^{2/e_s}} < n_{t,(i,j)} m^{-2/e_s} (c_s \alpha_k C C_s)^{-1/e_s} \le \frac{1}{\Delta_{e_{(i,j)}^p}^{2/e_s}}, \quad A_{t+1} = e_{(i,j)}^l \Big\}$$

$$\le \sum_{t=0}^{T-1} \sum_{p=1}^{q_{(i,j)}} \Delta_{e_{(i,j)}^p} \sum_{l=p}^{q_{(i,j)}} \mathbb{1}\Big\{ \frac{1}{\Delta_{e_{(i,j)}^{p-1}}^{2/e_s}} < n_{t,(i,j)} m^{-2/e_s} (c_s \alpha_k C C_s)^{-1/e_s} \le \frac{1}{\Delta_{e_{(i,j)}^p}^{2/e_s}}, \quad A_{t+1} = e_{(i,j)}^l \Big\}$$

$$\leq \sum_{p=1}^{q(i,j)} \Delta_{e_{(i,j)}^p} \sum_{t=0}^{T-1} \mathbb{1}\left\{ \frac{1}{\Delta_{e_{(i,j)}^{p-1}}^{2/e_s}} < n_{t,(i,j)} m^{-2/e_s}(c_s\alpha_k CC_s)^{-1/e_s} \leq \frac{1}{\Delta_{e_{(i,j)}^p}^{2/e_s}}, \quad i \in A_{t+1}\right\}$$

$$\leq \sum_{p=1}^{q(i,j)} \Delta_{e_{(i,j)}^p}\left( \left\lfloor \frac{m^{2/e_s}(c_s\alpha_k CC_s)^{1/e_s}}{\Delta_{e_{(i,j)}^p}^{2/e_s}} \right\rfloor - \left\lfloor \frac{m^{2/e_s}(c_s\alpha_k CC_s)^{1/e_s}}{\Delta_{e_{(i,j)}^{p-1}}^{2/e_s}} \right\rfloor \right)$$

$$= \left\lfloor \frac{m^{2/e_s}(c_s\alpha_k CC_s)^{1/e_s}}{\Delta_{e_{(i,j)}^{q(i,j)}}^{2/e_s}} \right\rfloor \Delta_{e_{(i,j)}^{q(i,j)}} + \sum_{p=1}^{q(i,j)-1} \left\lfloor \frac{m^{2/e_s}(c_s\alpha_k CC_s)^{1/e_s}}{\Delta_{e_{(i,j)}^p}^{2/e_s}} \right\rfloor \left( \Delta_{e_{(i,j)}^p} - \Delta_{e_{(i,j)}^{p+1}} \right)$$

$$\leq m^{2/e_s}(c_s\alpha_k CC_s)^{1/e_s} \left( \Delta_{e_{(i,j)}^{q(i,j)}} \right)^{1-2/e_s} + \sum_{p=1}^{q(i,j)-1} \left( \Delta_{e_{(i,j)}^p} \right)^{-2/e_s} \left( \Delta_{e_{(i,j)}^p} - \Delta_{e_{(i,j)}^{p+1}} \right)$$

$$\leq m^{2/e_s}(c_s\alpha_k CC_s)^{1/e_s} \left( \left( \Delta_{e_{(i,j)}^{q(i,j)}} \right)^{1-2/e_s} + \int_{\Delta_{e_{(i,j)}^{q(i,j)}}}^{\Delta_{e_{(i,j)}^1}} x^{-2/e_s} dx \right).$$

If $e_s = 2$, then

$$\sum_{t=d(d+1)/2}^{T-1} \Delta_{A_{t+1}} \mathbb{1}\left\{ n_{t,(i,j)} \leq m^{2/e_s}(c_s\alpha_k CC_s)^{1/e_s} \frac{1}{\Delta_{A_{t+1}}^{2/e_s}} \right\}$$

$$\leq m(c_s\alpha_k CC_s)^{1/2}\left( 1 + \int_{\Delta_{e_{(i,j)}^{q(i,j)}}}^{\Delta_{e_{(i,j)}^1}} x^{-1} dx \right)$$

$$\leq m(c_s\alpha_k CC_s)^{1/2}\left( 1 + \log\left( \frac{\Delta_{\max}}{\Delta_{\min}} \right) \right).$$

Reinjecting this expression into yields Eq. (38), for $e_s = 2$

$$\sum_{t=d(d+1)/2}^{T-1} \Delta_{A_{t+1}} \sum_{k=1}^{k_0} \mathbb{1}\{\mathbb{A}_{t,k}^s\} \leq \sum_{(i,j)\in[d]^2} \sum_{k\in[k_0]} \frac{1}{\beta_k m^2} m(c_s\alpha_k CC_s)^{1/2}\left( 1 + \log\left( \frac{\Delta_{\max}}{\Delta_{\min}} \right) \right)$$

$$= (c_s CC_s)^{1/2} \frac{d^2}{m}\left( 1 + \log\left( \frac{\Delta_{\max}}{\Delta_{\min}} \right) \right) \sum_{k\in[k_0]} \frac{\alpha_k^{1/2}}{\beta_k}$$

$$\leq 346\left( c_s CC_s \log(m) \right)^{1/2} m d^2\left( 1 + \log\left( \frac{\Delta_{\max}}{\Delta_{\min}} \right) \right). \qquad (39)$$

Else, for $1 < e_s < 2$, then

$$\sum_{t=d(d+1)/2}^{T-1} \Delta_{A_{t+1}} \mathbb{1}\left\{ n_{t,(i,j)} \leq m^{2/e_s}(c_s\alpha_k CC_s)^{1/e_s} \frac{1}{\Delta_{A_{t+1}}^{2/e_s}} \right\}$$

$$\leq m^{2/e_s}(c_s\alpha_k CC_s)^{1/e_s}\left( \left( \Delta_{e_{(i,j)}^{q(i,j)}} \right)^{1-2/e_s} + \int_{\Delta_{e_{(i,j)}^{q(i,j)}}}^{\Delta_{e_{(i,j)}^1}} x^{-2/e_s} dx \right)$$

$$= m^{2/e_s}(c_s\alpha_k CC_s)^{1/e_s}\left( \left( \Delta_{e_{(i,j)}^{q(i,j)}} \right)^{1-2/e_s} + \frac{e_s}{e_s-2}\left( \Delta_{e_{(i,j)}^1}^{1-2/e_s} - \Delta_{e_{(i,j)}^{q(i,j)}}^{1-2/e_s} \right) \right)$$

$$\leq m^{2/e_s}(c_s\alpha_k CC_s)^{1/e_s}\left( \left( \Delta_{e_{(i,j)}^{q(i,j)}} \right)^{1-2/e_s} - e_s\left( \Delta_{e_{(i,j)}^1}^{1-2/e_s} - \Delta_{e_{(i,j)}^{q(i,j)}}^{1-2/e_s} \right) \right)$$

$$\leq 3m^{2/e_s}(c_s\alpha_k CC_s)^{1/e_s} \Delta_{\min}^{1-2/e_s}.$$

This yield

$$\sum_{t=d(d+1)/2}^{T-1} \Delta_{A_{t+1}} \sum_{k=1}^{k_0} \mathbb{1}\{\mathbb{A}_{t,k}^s\} \leq \sum_{(i,j)\in[d]^2} \sum_{k\in[k_0]} \frac{1}{\beta_k m^2} 3m^{2/e_s}(c_s\alpha_k CC_s)^{1/e_s}\Delta_{\min}^{1-2/e_s}$$

$$= 3(c_s CC_s)^{1/e_s} d^2 m^{2/e_s-2}\Delta_{\min}^{1-2/e_s} \sum_{k\in[k_0]} \frac{\alpha_k^{1/e_s}}{\beta_k}$$

$$\leq 189.30^{1/e_s}\Big(c_s CC_s \log(m)\Big)^{1/e_s} d^2 m^{2/e_s}\Delta_{\min}^{1-2/e_s}. \qquad (40)$$

Finally, for $e_s > 2$,

$$\sum_{t=d(d+1)/2}^{T-1} \Delta_{A_{t+1}}\mathbb{1}\left\{ n_{t,(i,j)} \leq m^{2/e_s}(c_s\alpha_k CC_s)^{1/e_s}\frac{1}{\Delta_{A_{t+1}}^{2/e_s}}\right\}$$

$$\leq m^{2/e_s}(c_s\alpha_k CC_s)^{1/e_s}\left( \Big(\Delta_{e_{(i,j)}^{q(i,j)}}\Big)^{1-2/e_s} + \int_{\Delta_{e_{(i,j)}^{q(i,j)}}}^{\Delta_{e_{(i,j)}^1}} x^{-2/e_s}dx \right)$$

$$= m^{2/e_s}(c_s\alpha_k CC_s)^{1/e_s}\left( \Big(\Delta_{e_{(i,j)}^{q(i,j)}}\Big)^{1-2/e_s} + \frac{e_s}{e_s-2}\Big(\Delta_{e_{(i,j)}^1}^{1-2/e_s} - \Delta_{e_{(i,j)}^{q(i,j)}}^{1-2/e_s}\Big) \right)$$

$$\leq m^{2/e_s}(c_s\alpha_k CC_s)^{1/e_s}\frac{e_s}{e_s-2}\Big(\Delta_{\max}\Big)^{1-2/e_s},$$

and

$$\sum_{t=d(d+1)/2}^{T-1} \Delta_{A_{t+1}} \sum_{k=1}^{k_0} \mathbb{1}\{\mathbb{A}_{t,k}^s\} \leq \sum_{(i,j)\in[d]^2} \sum_{k\in[k_0]} \frac{1}{\beta_k m^2} m^{2/e_s}(c_s\alpha_k CC_s)^{1/e_s}\frac{e_s}{e_s-2}\Big(\Delta_{\max}\Big)^{1-2/e_s}$$

$$= (c_s CC_s)^{1/e_s}\frac{e_s}{e_s-2}d^2 m^{2/e_s-2}\Delta_{\max}^{1-2/e_s} \sum_{k\in[k_0]} \frac{\alpha_k^{1/e_s}}{\beta_k}$$

$$\leq 63.30^{1/e_s}\Big(c_s CC_s \log(m)\Big)^{1/e_s}\frac{e_s}{e_s-2}d^2 m^{2/e_s}\Delta_{\max}^{1-2/e_s}. \qquad (41)$$

All in all, we reinject Eq. (36), Eq. (39), Eq. (40) and Eq. (41) into Eq. (31), yielding

$$\sum_{t=d(d+1)/2}^{T-1} \Delta_{A_{t+1}}\mathbb{1}\{\mathcal{E}_t\}$$

$$\leq \sum_{t=d(d+1)/2}^{T-1}\left[ \Delta_{A_{t+1}} \sum_{k=1}^{k_0} \mathbb{1}\{\mathbb{A}_{t,k}\}\right] + \sum_{s=1}^{r} \sum_{t=d(d+1)/2}^{T-1}\left[ \Delta_{A_{t+1}} \sum_{k=1}^{k_0} \mathbb{1}\{\mathbb{A}_{t,k}^s\}\right]$$

$$\leq 96c_1 C\log(m)^2 \sum_{i\in[d]}\Big( \max_{a\in\mathcal{A}/i\in a} \frac{C_{a,i}}{\Delta_a}\Big)$$

$$+ \sum_{s=1}^{r}\Big[ \mathbb{1}\Big\{e_s = 2\Big\} 346\Big(c_s CC_s \log(m)\Big)^{1/2} md^2\Big(1 + \log\Big(\frac{\Delta_{\max}}{\Delta_{\min}}\Big)\Big)$$

$$+ \mathbb{1}\Big\{1 < e_s < 2\Big\}63.30^{1/e_s}\Big(c_s CC_s \log(m)\Big)^{1/e_s} d^2 m^{2/e_s}\Delta_{\min}^{1-2/e_s}$$

$$+ \mathbb{1}\Big\{2 < e_s\Big\}63.30^{1/e_s}\Big(c_s CC_s \log(m)\Big)^{1/e_s}\frac{e_s}{e_s-2}d^2 m^{2/e_s}\Delta_{\max}^{1-2/e_s}\Big]. \qquad (42)$$

$\square$

## D.1 Definition of the sequences $(\alpha_k)$ and $(\beta_k)$

Let $\beta = 1/5$, $x > 0$. We define $\beta_0 = \alpha_0 = 1$. For $k \geq 1$, we define

$$\beta_k = \beta^k, \qquad \alpha_k = x\beta^k . \tag{43}$$

Let's first look for an adequate $k_0$ for Lemma 4, taking $1 \leq k_0 = \lceil \frac{2\log(\sqrt{2}m)}{\log(1/\beta)} + 1 \rceil \leq (2\log(m)+3)$ is sufficient to have $0 < m\beta_{k_0} < \frac{1}{2m}$. This choice particularly yields

$$
\begin{aligned}
\left( \sum_{k=1}^{k_0-1} \frac{\beta_{k-1} - \beta_k}{\alpha_k} + \frac{\beta_{k_0-1}}{\alpha_{k_0}} \right) &= \left( \sum_{k=1}^{k_0-1} \frac{1-\beta}{\beta} + \frac{1}{\beta} \right) \frac{1}{x} \\
&= \left( (k_0 - 1)\frac{1-\beta}{\beta} + \frac{1}{\beta} \right) \frac{1}{x} \\
&= \left( 4k_0 + 1 \right) \frac{1}{x} \\
&< 1 ,
\end{aligned}
\tag{44}
$$

for $x = 4k_0 + 2$.

Besides,

$$\sum_{k=1}^{k_0} \frac{\alpha_k}{\beta_k} = (4k_0 + 2)k_0 \leq 16\log(m)^2 + 52\log(m) + 42 \leq 48\log(m)^2 \tag{45}$$

as $m \geq 5$. Let $c \in \mathbb{R}$, $c > 1$. Then

$$
\begin{aligned}
\sum_{k=1}^{k_0} \frac{\alpha_k^{1/c}}{\beta_k} &= (4k_0 + 2)^{1/c} \sum_{k=1}^{k_0} (\beta^{1/c-1})^k \\
&\leq (8\log(m) + 14)^{1/c} \sum_{k=1}^{k_0} (5^{\frac{c-1}{c}})^k \\
&\leq 30^{1/c} \log(m)^{1/c} \sum_{k=1}^{k_0} 5^k \\
&= 30^{1/c} \log(m)^{1/c} 5 \frac{5^{k_0} - 1}{5 - 1} \\
&= 30^{1/c} \log(m)^{1/c} \frac{5}{4}(5^{k_0} - 1) \\
&= 30^{1/c} \log(m)^{1/c} \frac{5}{4}\left( \exp\left( \log(5)\left( \frac{2\log(\sqrt{2}m)}{\log(5)} + 2 \right) \right) - 1 \right) \\
&= 30^{1/c} \log(m)^{1/c} \frac{5}{4}\left( 50m^2 - 1 \right) \\
&\leq 63m^2 \left( 30^{1/c} \log(m)^{1/c} \right) \\
&\leq 63.30^{1/c} m^2 \log(m)^{1/c} .
\end{aligned}
$$

## D.2 Proof of Lemma 4

**Lemma 4.** *Let's consider the assumptions of Proposition 6. Let $\mathbb{A}_{t,k}$ and $(\mathbb{A}_{t,k}^s)_{s \in [r]}$ be the events defined in Eq. (28) and Eq. (30). Let $k_0 \in \mathbb{N}^*$ such that $0 < m\beta_{k_0} < \frac{1}{2m}$ and $t \geq d(d+1)/2$.*

$$\mathbb{1}\{\mathcal{E}_t\} \leq \sum_{k=1}^{k_0} \mathbb{1}\{\mathbb{A}_{t,k}\} + \sum_{s=1}^{r} \sum_{k=1}^{k_0} \mathbb{1}\{\mathbb{A}_{t,k}^s\} .$$

*Proof.* Let's consider the assumptions of 6, $\mathbb{A}_{t,k}$ and $(\mathbb{A}^s_{t,k})_{s\in[r]}$ be the events defined in Eq. (28) and Eq. (30). Let $k_0 \in \mathbb{N}^*$ such that $0 < m\beta_{k_0} < \frac{1}{2m}$ and $t \geq d(d+1)/2$.

We first prove that the events for $k \geq k_0$ cannot happen. Let $k \geq k_0$,

$$\mathbb{A}_{t,k} = \left\{ \sum_{i\in S^1_{t,k}} \frac{\boldsymbol{\Sigma}^*_{i,i}}{C_{A_{t+1},i}} \geq \beta_k m; \quad \forall l < k, \sum_{i\in S^1_{t,l}} \frac{\boldsymbol{\Sigma}_{i,i}}{C_{A_{t+1},i}} < \beta_l m \right\}.$$

As $\beta_k m < \beta_{k_0} m < \frac{1}{2m} \leq \min_{i,a} \frac{\boldsymbol{\Sigma}_{i,i}}{C_{a,i}}$ and $(S^1_{t,l})_l$ is a decreasing sequence of sets, $\sum_{i\in S_{t,k_0}} \frac{\boldsymbol{\Sigma}_{i,i}}{C_{A_{t+1},i}} < \beta_{k_0} m$ imply $S_{t,k_0} = \emptyset$ and $\sum_{i\in S_{t,k}} \frac{\boldsymbol{\Sigma}^*_{i,i}}{C_{A_{t+1}}} = 0 < \beta_k m$. Therefore, $\mathbb{A}_{t,k}$ cannot happen and we denote

$$\mathbb{A}_t = \bigcup_{k\geq 1} \mathbb{A}_{t,k} = \bigcup_{k\in[k_0]} \mathbb{A}_{t,k} = \bigcup_{k\in[k_0]} \left\{ \sum_{i\in S_{t,k}} \frac{\boldsymbol{\Sigma}_{i,i}}{C_{A_{t+1},i}} \geq \beta_k m; \quad \forall l < k, \sum_{i\in S^1_{t,l}} \frac{\boldsymbol{\Sigma}_{i,i}}{C_{A_{t+1},i}} < \beta_l m \right\}.$$

Likewise, for $k > k_0$ and $s \in [r]$,

$$\mathbb{A}^s_{t,k} = \left\{ |S^s_{t,k}| \geq \beta_k m^2; \quad \forall l < k, |S^s_{t,k}| < \beta_l m^2 \right\}.$$

As $\beta_{k_0} m^2 < 1/2 < 1$ and $(S^s_{t,l})_l$ is a decreasing sequence of sets, then $|S^s_{t,k_0}| < \beta_{k_0} m^2$ imply $S^s_{t,k_0} = \emptyset$ and $|S^s_{t,k}| = 0 < \beta_k m^2$. Therefore, $\mathbb{A}^s_{t,k}$ cannot happen and we denote

$$\mathbb{A}^s_t = \bigcup_{k\geq 1} \mathbb{A}^s_{t,k} = \bigcup_{k\in[k_0]} \mathbb{A}^s_{t,k} = \bigcup_{k\in[k_0]} \left\{ |S^s_{t,k}| \geq \beta_k m^2; \quad \forall l < k, |S^s_{t,k}| < \beta_l m^2 \right\}.$$

The idea is now to prove that

$$\left( \mathbb{A}_t \cup \bigcup_{s=1}^r \mathbb{A}^s_t \right)^c = \mathbb{A}^c_t \cap \cap_{s=1}^r \left( \mathbb{A}^s_t \right)^c \subseteq \mathcal{E}^c_t.$$

We begin by considering $(\mathbb{A}^1_t)^c$,

$$(\mathbb{A}_t)^c = \cap_{k=1}^{k_0} (\mathbb{A}_{t,k})^c$$

$$= \cap_{k=1}^{k_0} \left( \left\{ \sum_{i\in S_{t,k}} \frac{\boldsymbol{\Sigma}_{i,i}}{C_{A_{t+1},i}} < \beta_k m \right\} \bigcup_{l=1}^{k-1} \left\{ \sum_{i\in S^1_{t,l}} \frac{\boldsymbol{\Sigma}_{i,i}}{C_{A_{t+1},i}} \geq \beta_l m \right\} \right)$$

$$= \cap_{k=1}^{k_0} \left\{ \sum_{i\in S^1_{t,k}} \frac{\boldsymbol{\Sigma}_{i,i}}{C_{A_{t+1},i}} < \beta_k m \right\}. \tag{46}$$

Then, under $(\mathbb{A}_t)^c$, denoting $S_{t,0} = A_{t+1}$, as $S_{t,k_0} = \emptyset$ and the sets $S_{t,k}$ are decreasing with respect to $k$,

$$\sum_{i\in A_{t+1}} \frac{C_{A_{t+1},i}}{n_{t,(i,i)}} = \sum_{k=1}^{k_0} \sum_{i\in S^1_{t,k-1}\backslash S^1_{t,k}} \frac{C_{A_{t+1},i}}{n_{t,(i,i)}}$$

$$\leq \sum_{k=1}^{k_0} \sum_{i\in S^1_{t,k-1}\backslash S^1_{t,k}} C_{A_{t+1},i} \frac{1}{3m\alpha_k} \frac{\Delta^2_{A_{t+1}}}{C} \frac{\boldsymbol{\Sigma}^*_{i,i}}{C^2_{A_{t+1},i}} \leftarrow \text{ by Eq. (27)}$$

$$= \frac{\Delta^2_{A_{t+1}}}{c_1 mC} \sum_{k=1}^{k_0} \frac{1}{\alpha_k} \sum_{i\in S_{t,k-1}\backslash S_{t,k}} \frac{\boldsymbol{\Sigma}_{i,i}}{C_{A_{t+1},i}}$$

$$= \frac{\Delta^2_{A_{t+1}}}{c_1 mC} \sum_{k=1}^{k_0} \frac{1}{\alpha_k} \left( \sum_{i\in S^1_{t,k-1}} \frac{\boldsymbol{\Sigma}_{i,i}}{C_{A_{t+1},i}} - \sum_{i\in S^1_{t,k}} \frac{\boldsymbol{\Sigma}_{i,i}}{C_{A_{t+1},i}} \right)$$

$$
= \frac{\Delta^2_{A_{t+1}}}{c_1 mC} \sum_{k=0}^{k_0-1} \frac{1}{\alpha_{k+1}} \left( \sum_{i \in S^1_{t,k}} \frac{\Sigma_{i,i}}{C_{A_{t+1},i}} - \sum_{i \in S^1_{t,k+1}} \frac{\Sigma_{i,i}}{C_{A_{t+1},i}} \right)
$$

$$
= \frac{\Delta^2_{A_{t+1}}}{c_1 mC} \left( \frac{1}{\alpha_1} \sum_{i \in S^1_{t,0}} \frac{\Sigma_{i,i}}{C_{A_{t+1},i}} + \sum_{k=1}^{k_0-1} \left( \frac{1}{\alpha_{k+1}} - \frac{1}{\alpha_k} \right) \sum_{i \in S^1_{t,k}} \frac{\Sigma_{i,i}}{C_{A_{t+1},i}} \right)
$$

$$
< \frac{\Delta^2_{A_{t+1}}}{c_1 mC} \left( \frac{m}{\alpha_1} + \sum_{k=1}^{k_0-1} m\beta_k \left( \frac{1}{\alpha_{k+1}} - \frac{1}{\alpha_k} \right) \right) \quad \leftarrow S_{t,0} = A_{t+1} \text{ and Eq. (46)}
$$

$$
= \frac{\Delta^2_{A_{t+1}}}{c_1 C} \left( \sum_{k=1}^{k_0-1} \frac{\beta_{k-1} - \beta_k}{\alpha_k} + \frac{\beta_{k_0-1}}{\alpha_{k_0}} \right)
$$

$$
\leq \frac{1}{c} \frac{\Delta^2_{A_{t+1}}}{C} . \quad \leftarrow \text{ by Eq. (44)} \tag{47}
$$

Likewise for $s \in [r]$,

$$
(\mathbb{A}^s_t)^c = \cap_{k=1}^{k_0} \left\{ |S^s_{t,k}| < \beta_k m^2 \right\} . \tag{48}
$$

Denoting $S^s_{t,0} = A_{t+1} \times A_{t+1}$, as $S^s_{t,k_0} = \emptyset$,

$$
\sum_{(i,j) \in A_{t+1}} \frac{C_s}{n^{e_s}_{t,(i,j)}} = \sum_{k=1}^{k_0} \sum_{(i,j) \in S_{t,k-1} \setminus S_{t,k}} \frac{C_s}{n^{e_s}_{t,i}}
$$

$$
\leq \sum_{k=1}^{k_0} \sum_{(i,j) \in S_{t,k-1} \setminus S_{t,k}} C_s \frac{1}{c_2 m^2 \alpha_k} \frac{\Delta^2_{A_{t+1}}}{C} \frac{1}{C_s} \quad \leftarrow \text{ by Eq. (29)}
$$

$$
= \frac{\Delta^2_{A_{t+1}}}{c_s m^2 C} \sum_{k=1}^{k_0} \frac{1}{\alpha_k} \left( |S_{t,k-1}| - |S_{t,k}| \right)
$$

$$
= \frac{\Delta^2_{A_{t+1}}}{c_s m^2 C} \sum_{k=0}^{k_0-1} \frac{1}{\alpha_{k+1}} \left( |S_{t,k}| - |S_{t,k+1}| \right)
$$

$$
= \frac{\Delta^2_{A_{t+1}}}{c_s m^2 C} \left( \frac{|S_{t,0}|}{\alpha_1} + \sum_{k=1}^{k_0-1} |S_{t,k}| \left( \frac{1}{\alpha_{k+1}} - \frac{1}{\alpha_k} \right) \right)
$$

$$
< \frac{\Delta^2_{A_{t+1}}}{c_s m^2 C} \left( \frac{1}{\alpha_1} m^2 + \sum_{k=1}^{k_0-1} \beta_k m^2 \left( \frac{1}{\alpha_{k+1}} - \frac{1}{\alpha_k} \right) \right) \quad \leftarrow \text{ by Eq. (48)}
$$

$$
= \frac{\Delta^2_{A_{t+1}}}{c_2 C} \left( \sum_{k=1}^{k_0-1} \frac{\beta_{k-1} - \beta_k}{\alpha_k} + \frac{\beta_{k_0-1}}{\alpha_{k_0}} \right)
$$

$$
\leq \frac{1}{c_2} \frac{\Delta^2_{A_{t+1}}}{C} . \quad \leftarrow \text{ by Eq. (44)} \tag{49}
$$

Therefore, under $\mathbb{A}^c_t \cap \cap_{s=1}^{r} \left( \mathbb{A}^s_t \right)^c$, summing Eq. (47) and Eq. (49)

$$
\sum_{i \in A_{t+1}} \frac{C_{A_{t+1},i}}{n_{t,(i,j)}} + \sum_{s \in [r]} \left[ \sum_{(i,j) \in A_{t+1}} \frac{C_s}{n^{e_s}_{t,(i,j)}} \right] < \left( \frac{1}{c} + \sum_{s \in [r]} \frac{1}{c_s} \right) \frac{\Delta^2_{A_{t+1}}}{C}
$$

$$
= \frac{\Delta^2_{A_{t+1}}}{C} ,
$$

which contradict Eq. (25) and thus imply $\mathcal{E}_t^c$. By contraposition, we have proved that $\mathcal{E}_t$ imply $\mathbb{A}_t \cap \bigcup_{s=1}^r \left( \mathbb{A}_t^s \right)$. Therefore,

$$\mathbb{1}\{\mathcal{E}_t\} \leq \sum_{k=1}^{k_0} \mathbb{1}\{\mathbb{A}_{t,k}\} + \sum_{s=1}^r \sum_{k=1}^{k_0} \mathbb{1}\{\mathbb{A}_{t,k}^s\}.$$

$\square$

# E  Details for `OLS-UCB-C` (Section 5.2)

## E.1  Proof of Proposition 3

**Proposition 3.** *Let $\delta > 0$. Then, `OLS-UCB-C` yields*

$$\mathbb{E}\left[ \sum_{t=d(d+1)/2}^{T-1} \Delta_{A_{t+1}} \mathbb{1}\{\mathcal{G}_t \cap \mathcal{C}_t\} \right] = O\left( \log(T)^2 \log(m)^2 \sum_{i=1}^d \max_{a \in \mathcal{A}/i \in a} \frac{\sigma_{a,i}^2}{\Delta_a} \right),$$

*as $T \to \infty$, where $\sigma_{a,i}^2 = \sum_{j \in a} (\mathbf{\Sigma}_{i,j})_+$.*

*Proof.* The objective is to use Proposition 6.

**Proposition 6.** *Let $r \in \mathbb{N}$, $e \in (1, +\infty)^r$. Let $(\mathcal{E}_t)_{t \geq d(d+1)/2}$ be a sequence of events such that for all $t \geq d(d+1)/2$, under $\mathcal{E}_t$,*

$$\frac{\Delta_{A_{t+1}}^2}{C} \leq \sum_{i \in A_{t+1}} \frac{C_{A_{t+1},i}}{n_{t,(i,j)}} + \sum_{s \in [r]} \left[ \sum_{(i,j) \in A_{t+1}} \frac{C_s}{n_{t,(i,j)}^{e_s}} \right], \tag{25}$$

*where $C$ and $(C_s)_{s \in [r]}$ are problem-dependent positive constants. $C_{A_{t+1},i}$ is a positive constant depending on $A_{t+1}$ and $i$ so that, for all $a \in \mathcal{A}$, $C_{a,i} \leq 2m\mathbf{\Sigma}_{i,i}$. Let $c \in \mathbb{R}_+^*$ and $(c_s)_{s \in [r]} \in (\mathbb{R}_+^*)^r$ be positive constants such that $1/c + \sum_{s \in [r]} 1/c_s = 1$.*

*Then,*

$$\sum_{t=d(d+1)/2}^{T-1} \Delta_{A_{t+1}} \mathbb{1}\{\mathcal{E}_t\}$$

$$\leq 96 c_1 C \log(m)^2 \sum_{i \in [d]} \left( \max_{a \in \mathcal{A}/i \in a} \frac{C_{a,i}}{\Delta_a} \right)$$

$$+ \sum_{s=1}^r \left[ \mathbb{1}\{e_s = 2\} 346 \left( c_s C C_s \log(m) \right)^{1/2} m d^2 \left( 1 + \log\left( \frac{\Delta_{\max}}{\Delta_{\min}} \right) \right) \right.$$

$$+ \mathbb{1}\{1 < e_s < 2\} 60.30^{1/e_s} \left( c_s C C_s \log(m) \right)^{1/e_s} d^2 m^{2/e_s} \Delta_{\min}^{1-2/e_s}$$

$$\left. + \mathbb{1}\{2 < e_s\} 60.30^{1/e_s} \left( c_s C C_s \log(m) \right)^{1/e_s} \frac{e_s}{e_s - 2} d^2 m^{2/e_s} \Delta_{\max}^{1-2/e_s} \right]. \tag{26}$$

*where $(\alpha_k)_{k \in \mathbb{N}^*}$, $(\beta_k)_{k \in \mathbb{N}^*}$ and $k_0 \in \mathbb{N}^*$ are defined in Appendix D.1.*

We need to check that its hypotheses are satisfied. Let $t \geq d(d+1)/2$ and $\delta > 0$, then we have the Lemma.

**Lemma 5.** *Let $\delta > 0$ and $t \geq d(d+1)/2$. Then under $\{\mathcal{G}_t \cap \mathcal{C}_t\}$, `OLS-UCB-C` satisfies*

$$\frac{\Delta_{A_{t+1}}^2}{f_{T,\delta}^2} \sum_{i \in A_{t+1}} \frac{4\bar{\sigma}_{A_{t+1},i}^2}{n_{t,(i,i)}} + \sum_{(i,j) \in A_{t+1}} \frac{(4d + h_{t,\delta}^2)\|B\|_\infty^2}{n_{t,(i,j)}^2} + \sum_{(i,j) \in A_{t+1}} \frac{5h_{t,\delta}\|B\|_\infty^2}{n_{t,(i,j)}^{3/2}} + \sum_{(i,j) \in A_{t+1}} \frac{\|B\|_\infty^2}{n_{t,(i,j)}^3},$$

*where $\bar{\sigma}_{A_{t+1},i}^2 = 2 \sum_{j \in A_{t+1}/\mathbf{\Sigma}_{j,j} \leq \mathbf{\Sigma}_{i,i}} (\mathbf{\Sigma}_{i,j})_+ \leq 2\sigma_{A_{t+1},i}^2$.*

Therefore, we can choose $r = 3$, $e = (2, 3/2, 3)$ and $(\mathcal{E}_t) = (\mathcal{G}_t \cap \mathcal{C}_t)$. Taking $c = (4, 4, 4, 4)$ and identifying the rest of the coefficients yields that OLS-UCB-C satisfies

$$
\sum_{t=d(d+1)/2}^{T-1} \Delta_{A_{t+1}} \mathbb{1}\{\mathcal{G}_t \cap \mathcal{C}_t\}
$$

$$
\leq 384 f_{T,\delta}^2 \log(m)^2 \sum_{i \in [d]} \left( \max_{a \in \mathcal{A}/i \in a} \frac{\bar{\sigma}_{a,i}^2}{\Delta_a} \right)
$$

$$
+ 692 \|B\|_\infty f_{T,\delta} (4d + h_{t,\delta}^2)^{1/2} \log(m)^{1/2} \left( 1 + \log\left( \frac{\Delta_{\max}}{\Delta_{\min}} \right) \right)
$$

$$
+ 1460 \|B\|_\infty^{4/3} f_{T,\delta}^{4/3} h_{t,\delta}^{2/3} \log(m)^{2/3} d^2 m^{2/3} \Delta_{\min}^{-1/3}
$$

$$
+ 296 \|B\|_\infty^{2/3} f_{T,\delta}^{2/3} \log(m)^{1/3} d^2 m^{2/3} \Delta_{\max}^{1/3} .
$$

As

$$
f_{t,\delta} = 6 \log(1/\delta) + 6 \Big( \log(t) + (d+2) \log(\log(t)) \Big) + 3d \Big( 2\log(2) + \log(1+e) \Big),
$$

$$
h_{t,\delta} = \Big( 1 + 2\log(1/\delta) + 2\log \big( t \log(t)^2 d(d+1) \big) + \log(1+t) \Big)^{1/2},
$$

we deduce

$$
\sum_{t=d(d+1)/2}^{T-1} \Delta_{A_{t+1}} \mathbb{1}\{\mathcal{G}_t \cap \mathcal{C}_t\} = O\left( \log(T)^2 \log(m)^2 \sum_{i \in [d]} \left( \max_{a \in \mathcal{A}/i \in a} \frac{\bar{\sigma}_{a,i}^2}{\Delta_a} \right) \right). \tag{50}
$$

$\square$

### E.2  Proof of Lemma 5

**Lemma 5.** *Let $\delta > 0$ and $t \geq d(d+1)/2$. Then under $\{\mathcal{G}_t \cap \mathcal{C}_t\}$, OLS-UCB-C satisfies*

$$
\frac{\Delta_{A_{t+1}}^2}{f_{T,\delta}^2} \sum_{i \in A_{t+1}} \frac{4\bar{\sigma}_{A_{t+1},i}^2}{n_{t,(i,i)}} + \sum_{(i,j) \in A_{t+1}} \frac{(4d + h_{t,\delta}^2) \|B\|_\infty^2}{n_{t,(i,j)}^2} + \sum_{(i,j) \in A_{t+1}} \frac{5h_{t,\delta} \|B\|_\infty^2}{n_{t,(i,j)}^{3/2}} + \sum_{(i,j) \in A_{t+1}} \frac{\|B\|_\infty^2}{n_{t,(i,j)}^3},
$$

*where $\bar{\sigma}_{A_{t+1},i}^2 = 2 \sum_{j \in A_{t+1}/\Sigma_{j,j} \leq \Sigma_{i,i}} (\Sigma_{i,j})_+ \leq 2\sigma_{A_{t+1},i}^2$.*

*Proof.* Let $t \geq d(d+1)/2$ and $\delta > 0$. OLS-UCB-C statisfies the following Lemma.

**Lemma 6.** *Let $t \geq d(d+1)/2$ and $\delta > 0$. Then for OLS-UCB-C, under the event $\{\mathcal{G}_t \cap \mathcal{C}_t\}$,*

$$
\Delta_{A_{t+1}} \leq f_{t,\delta} \big( \|N_t^{-1} A_{t+1}\|_{Z_t} + \|N_t^{-1} A_{t+1}\|_{\hat{Z}_t} \big) .
$$

Therefore, under $\{\mathcal{G}_t \cap \mathcal{C}_t\}$, and

$$
0 \leq \Delta_{A_{t+1}} \leq f_{t,\delta} \big( \|N_t^{-1} A_{t+1}\|_{Z_t} + \|N_t^{-1} A_{t+1}\|_{\hat{Z}_t} \big)
$$

$$
\Delta_{A_{t+1}}^2 \leq f_{t,\delta}^2 \big( \|N_t^{-1} A_{t+1}\|_{Z_t} + \|N_t^{-1} A_{t+1}\|_{\hat{Z}_t} \big)^2
$$

$$
\leq 2 f_{t,\delta}^2 \big( \|N_t^{-1} A_{t+1}\|_{Z_t}^2 + \|N_t^{-1} A_{t+1}\|_{\hat{Z}_t}^2 \big)
$$

$$
\frac{\Delta_{A_{t+1}}^2}{2 f_{t\delta}^2} \leq \|N_t^{-1} A_{t+1}\|_{Z_t}^2 + \|N_t^{-1} A_{t+1}\|_{\hat{Z}_t}^2 . \tag{51}
$$

From, here, we develop the right-hand side,

$$\|\mathbf{N}_t^{-1} A_{t+1}\|_{\mathbf{Z}_t}^2 = A_{t+1}^\top \mathbf{N}_t^{-1} \mathbf{Z}_t \mathbf{N}_t^{-1} A_{t+1}$$

$$= \sum_{(i,j) \in A_{t+1}} \frac{(\mathbf{Z}_t)_{i,j}}{n_{t,(i,i)} n_{t,(j,j)}} \, .$$

As $\mathbf{Z}_t = \sum_{s=1}^{t} \mathbf{d}_{A_s} \mathbf{\Sigma}^* \mathbf{d}_{A_s} + \mathbf{d}_{\mathbf{\Sigma}^*} \mathbf{N}_t + \|B\|^2 \mathbf{I}$, we get

$$\|\mathbf{N}_t^{-1} A_{t+1}\|_{\mathbf{Z}_t}^2 = \sum_{(i,j) \in A_{t+1}} \frac{n_{t,(i,j)} \mathbf{\Sigma}_{i,j}}{n_{t,(i,i)} n_{t,(j,j)}} + \sum_{i \in A_{t+1}} \frac{n_{t(i,i)} \mathbf{\Sigma}_{i,i}}{n_{t,(i,i)}^2} + \sum_{i \in A_{t+1}} \frac{\|B\|^2}{n_{t,(i,i)}^2}$$

$$\leq \sum_{i \in A_{t+1}} \left( 2 \sum_{j \in A_{t+1} / \mathbf{\Sigma}_{j,j} \leq \mathbf{\Sigma}_{i,i}} \frac{n_{t,(i,j)} \mathbf{\Sigma}_{i,j}}{n_{t,(i,i)} n_{t,(j,j)}} \right) + \sum_{i \in A_{t+1}} \frac{\|B\|^2}{n_{t,(i,i)}^2} \, ,$$

by rearranging terms.

Now as for all $(i,j) \in [d]^2$, $n_{t,(i,j)} \leq \min\{n_{t,(i,i)}, n_{t,(j,j)}\}$, then

$$\|\mathbf{N}_t^{-1} A_{t+1}\|_{\mathbf{Z}_t}^2 \leq \sum_{i \in A_{t+1}} \frac{1}{n_{t,(i,i)}} \left( 2 \sum_{j \in A_{t+1} / \mathbf{\Sigma}_{j,j} \leq \mathbf{\Sigma}_{i,i}} \mathbf{\Sigma}_{i,j} \right) + \sum_{i \in A_{t+1}} \frac{\|B\|^2}{n_{t,(i,i)}^2} \, .$$

Denoting $\bar{\sigma}_{A_{t+1},i}^2 = 2 \sum_{j \in A_{t+1} / \mathbf{\Sigma}_{j,j} \leq \mathbf{\Sigma}_{i,i}} (\mathbf{\Sigma}_{i,j})_+$ yields

$$\|\mathbf{N}_t^{-1} A_{t+1}\|_{\mathbf{Z}_t}^2 \leq \sum_{i \in A_{t+1}} \frac{\bar{\sigma}_{A_{t+1},i}^2}{n_{t,(i,i)}} + \sum_{i \in A_{t+1}} \frac{\|B\|^2}{n_{t,(i,i)}^2} \, . \tag{52}$$

The second term from the right-hand side of Eq. (51) is developed in the same manner but involves more terms.

$$\|\mathbf{N}_t^{-1} A_{t+1}\|_{\hat{\mathbf{Z}}_t}^2 = A_{t+1}^\top \mathbf{N}_t^{-1} \hat{\mathbf{Z}}_t \mathbf{N}_t^{-1} A_{t+1}$$

$$= \sum_{(i,j) \in A_{t+1}} \frac{(\hat{\mathbf{Z}}_t)_{i,j}}{n_{t,(i,i)} n_{t,(j,j)}} \, .$$

We remind that $\hat{\mathbf{Z}}_t = \sum_{s=1}^{t} \mathbf{d}_{A_s} \hat{\mathbf{\Sigma}}_t \mathbf{d}_{A_s} + \mathbf{d}_{\hat{\mathbf{\Sigma}}_t} \mathbf{N}_t + \|B\|^2 \mathbf{I}$ where for all $(i,j) \in [d]^2$,

$$\hat{\mathbf{\Sigma}}_{t,(i,j)} = \hat{\chi}_{t,(i,j)} + \frac{B_i B_j}{4} \left( \frac{5 h_{t,\delta}}{\sqrt{n_{t,(i,j)}}} + \frac{h_{t,\delta}^2}{n_{t,(i,j)}} + \frac{1}{n_{t,(i,j)}^2} \right) \, .$$

Being under the event $\mathcal{C}$, Proposition 2 yields $\hat{\Sigma}_{t,(i,j)} \leq \Sigma_{i,j} + \frac{B_i B_j}{4}\left(\frac{5h_{t,\delta}}{\sqrt{n_{t,(i,j)}}} + \frac{h_{t,\delta}^2}{n_{t,(i,j)}} + \frac{1}{n_{t,(i,j)}^2}\right)$.
Then,

$$
\begin{aligned}
\|\mathbf{N}_t^{-1}A_{t+1}\|_{\hat{\mathbf{Z}}_t}^2 &\leq \sum_{(i,j)\in A_{t+1}} \frac{n_{t,(i,j)}\Sigma_{i,j}}{n_{t,(i,i)}n_{t,(j,j)}} \\
&\quad + \sum_{(i,j)\in A_{t+1}} \frac{n_{t,(i,j)}}{n_{t,(i,i)}n_{t,(j,j)}}\frac{B_i B_j}{4}\left(\frac{5h_{t,\delta}}{\sqrt{n_{t,(i,j)}}} + \frac{h_{t,\delta}^2}{n_{t,(i,j)}} + \frac{1}{n_{t,(i,j)}^2}\right) \\
&\quad + \sum_{i\in A_{t+1}} \frac{n_{t(i,i)}\Sigma_{i,i}}{n_{t,(i,i)}^2} + \sum_{i\in A_{t+1}} \frac{B_i^2}{4n_{t,(i,i)}}\left(\frac{5h_{t,\delta}}{\sqrt{n_{t,(i,i)}}} + \frac{h_{t,\delta}^2}{n_{t,(i,i)}} + \frac{1}{n_{t,(i,i)}^2}\right) \\
&\quad + \sum_{i\in A_{t+1}} \frac{\|B\|^2}{n_{t,(i,i)}^2} \\
&\leq \sum_{i\in A_{t+1}} \frac{\bar{\sigma}_{A_{t+1},i}^2}{n_{t,(i,i)}} + \sum_{i\in A_{t+1}} \frac{\|B\|^2}{n_{t,(i,i)}^2} \\
&\quad + \sum_{(i,j)\in A_{t+1}} \frac{5h_{t,\delta}\|B\|_\infty^2}{4}\frac{1}{n_{t,(i,j)}^{3/2}} + \sum_{(i,j)\in A_{t+1}} \frac{h_{t,\delta}^2\|B\|_\infty^2}{4}\frac{1}{n_{t,(i,j)}^2} + \sum_{(i,j)\in A_{t+1}} \frac{\|B\|_\infty^2}{4}\frac{1}{n_{t,(i,j)}^3} \\
&\quad + \sum_{i\in A_{t+1}} \frac{5h_{t,\delta}\|B\|_\infty^2}{4}\frac{1}{n_{t,(i,i)}^{3/2}} + \sum_{i\in A_{t+1}} \frac{h_{t,\delta}^2\|B\|_\infty^2}{4}\frac{1}{n_{t,(i,i)}^2} + \sum_{i\in A_{t+1}} \frac{\|B\|_\infty^2}{4}\frac{1}{n_{t,(i,i)}^3}.
\end{aligned}
$$
(53)

Reinjecting Eq. (52) and Eq. (53) into Eq. (51) yields

$$
\begin{aligned}
\frac{\Delta_{A_{t+1}}^2}{2f_{t,\delta}^2} &\leq \|\mathbf{N}_t^{-1}A_{t+1}\|_{\mathbf{Z}_t}^2 + \|\mathbf{N}_t^{-1}A_{t+1}\|_{\hat{\mathbf{Z}}_t}^2 \\
&\leq \sum_{i\in A_{t+1}} \frac{2\bar{\sigma}_{A_{t+1},i}^2}{n_{t,(i,i)}} + \sum_{(i,j)\in A_{t+1}} \frac{(2d+h_{t,\delta}^2/2)\|B\|_\infty^2}{n_{t,(i,j)}^2} \\
&\quad + \sum_{(i,j)\in A_{t+1}} \frac{5h_{t,\delta}\|B\|_\infty^2/2}{n_{t,(i,j)}^{3/2}} + \sum_{(i,j)\in A_{t+1}} \frac{\|B\|_\infty^2/2}{n_{t,(i,j)}^3}.
\end{aligned}
$$

The desired inequality just comes from $f_{T,\delta} \geq f_{t,\delta}$  $\qquad\qquad\square$

### E.3 Proof of Lemma 6

**Lemma 6.** *Let $t \geq d(d+1)/2$ and $\delta > 0$. Then for $\mathtt{OLS\text{-}UCB\text{-}C}$, under the event $\{\mathcal{G}_t \cap \mathcal{C}_t\}$,*
$$
\Delta_{A_{t+1}} \leq f_{t,\delta}\big(\|\mathbf{N}_t^{-1}A_{t+1}\|_{\mathbf{Z}_t} + \|\mathbf{N}_t^{-1}A_{t+1}\|_{\hat{\mathbf{Z}}_t}\big).
$$

*Proof.* Let $t \geq d(d+1)/2$ and $\delta > 0$. The error in estimating the mean reward for action $a$ with $\langle a, \hat{\mu}_t\rangle$ is bounded as
$$
\left|a^\top(\hat{\mu}_t - \mu)\right| \leq \|\mathbf{N}_t^{-1}a\|_{\mathbf{Z}_t}\big\|\textstyle\sum_{s=1}^t \mathbf{d}_{A_s}\eta_s\big\|_{\mathbf{Z}_t^{-1}}.
$$

The definition of $\mathcal{G}_t = \big\{\big\|\sum_{s=1}^t \mathbf{d}_{A_s}\eta_s\big\|_{\mathbf{Z}_t^{-1}} \leq f_{t,\delta}\big\}$ yields that in this event,
$$
\langle A_{t+1}, \hat{\mu}_t\rangle \leq \langle A_{t+1}, \mu\rangle + f_{t,\delta}\|\mathbf{N}_t^{-1}A_{t+1}\|_{\mathbf{Z}_t},
$$
and
$$
\langle a^*, \mu\rangle - f_{t,\delta}\|\mathbf{N}_t^{-1}a^*\|_{\mathbf{Z}_t} \leq \langle a^*, \hat{\mu}_t\rangle.
$$

By the definition of $A_{t+1}$ for $\mathtt{OLS\text{-}UCB\text{-}C}$ in (6),
$$
\langle a^*, \hat{\mu}_t\rangle + f_{t,\delta}\|\mathbf{N}_t^{-1}a^*\|_{\hat{\mathbf{Z}}_t} \leq \langle A_{t+1}, \hat{\mu}_t\rangle + f_{t,\delta}\|\mathbf{N}_t^{-1}A_{t+1}\|_{\hat{\mathbf{Z}}_t}.
$$

Combining the expressions gives

$$\langle a^*, \mu \rangle + f_{t,\delta}(\|\mathbf{N}_t^{-1} a^*\|_{\hat{\mathbf{Z}}_t} - \|\mathbf{N}_t^{-1} a^*\|_{\mathbf{Z}_t}) \leq \langle A_{t+1}, \mu \rangle + f_{t,\delta}(\|\mathbf{N}_t^{-1} A_{t+1}\|_{\mathbf{Z}_t} + \|\mathbf{N}_t^{-1} A_{t+1}\|_{\hat{\mathbf{Z}}_t}).$$

Now, we use the fact that under $\mathcal{C}_t$, $\hat{\mathbf{Z}}_t$ uses coefficient-wise upper bounds of $\mathbf{\Sigma}$, which yields that

$$\|\mathbf{N}_t^{-1} a^*\|_{\mathbf{Z}_t}^2 \leq \|\mathbf{N}_t^{-1} a^*\|_{\hat{\mathbf{Z}}_t}^2.$$

Rearranging terms the desired result. $\qquad \square$

### E.4 Proof of the gap-free bound

**Theorem 1.** *Let $T \in \mathbb{N}^*$ and $\delta > 0$.*

*Then,* OLS-UCB-C *(Alg. 2) satisfies the* gap-dependent *regret upper bound*

$$\mathbb{E}[R_T] = \tilde{O}\left( \log(m)^2 \sum_{i=1}^{d} \max_{a \in \mathcal{A}/i \in a, \Delta_a > 0} \frac{\sigma_{a,i}^2}{\Delta_a} \right),$$

*where $\sigma_{a,i}^2 = \sum_{j \in a} \max\{\mathbf{\Sigma}_{i,j}, 0\}$, and the* gap-free *regret upper bound*

$$\mathbb{E}[R_T] = \tilde{O}\left( \log(m)\sqrt{T} \sqrt{\sum_{i=1}^{d} \max_{a \in \mathcal{A}/i \in a} \sigma_{a,i}^2} \right).$$

*Proof.* Let $\Delta > 0$, then

$$\sum_{t=d(d+1)}^{T-1} \Delta_{A_{t+1}} \mathbb{1}\{\mathcal{G}_t \cap \mathcal{C}_t\} = \sum_{t=d(d+1)}^{T-1} \Delta_{A_{t+1}} \mathbb{1}\left\{ \mathcal{G}_t \cap \mathcal{C}_t \cap (\Delta_{A_{t+1}} \leq \Delta) \right\}$$

$$+ \sum_{t=d(d+1)}^{T-1} \Delta_{A_{t+1}} \mathbb{1}\left\{ \mathcal{G}_t \cap \mathcal{C}_t \cap (\Delta_{A_{t+1}} > \Delta) \right\}$$

$$\leq T\Delta + \sum_{t=d(d+1)}^{T-1} \Delta_{A_{t+1}} \mathbb{1}\left\{ \mathcal{G}_t \cap \mathcal{C}_t \cap (\Delta_{A_{t+1}} > \Delta) \right\}.$$

Adapting Proposition 6 to account for $\Delta_{A_{t+1}} > \Delta$ yields

$$\sum_{t=0}^{T-1} \Delta_{A_{t+1}} \mathbb{1}\{\mathcal{G}_t \cap \mathcal{C}_t \cap (\Delta_{A_{t+1}} > \Delta)\} \lesssim \frac{1}{\Delta} \log(T)^2 \log(m)^2 \sum_{i \in [d]} \left( \max_{a \in \mathcal{A}/i \in a} \sigma_{a,i}^2 \right).$$

where $\lesssim$ is an inequality up to constant factors (when $T$ varies).

Balancing $T\Delta$ and $\frac{1}{\Delta} \log(T)^2 \log(m)^2 \sum_{i \in [d]} \left( \max_{a \in \mathcal{A}/i \in a} \sigma_{a,i}^2 \right)$ yields

$$\mathbb{E}[R_T] = O\left( \log(m) \log(T)\sqrt{T} \sqrt{\sum_{i \in [d]} \max_{a \in \mathcal{A}/i \in a} \sigma_{a,i}^2} \right).$$

$\qquad \square$

## F Details for COS-V (Section 5.3)

### F.1 Proof for Lemma 7

**Lemma 7.** *Let $\delta > 0$. Then* COS-V *satisfies*

$$\sum_{s=d(d+A)/2}^{T} \mathbb{P}(\mathcal{H}_t^c) \leq \delta \sum_{t=1}^{T} \frac{1}{t \log(t)^2}.$$

*Proof.* Let $\delta > 0$, $t \geq d(d+1)/2$. We remind

$$\mathcal{H}_t = \left\{ \forall i \in [d], \; \left| \left( \hat{\mu}_{t,i} + (1 + g_{t,\delta}) f_{t,\delta} \frac{(\hat{\mathbf{Z}}_{t,i})^{1/2}}{n_{t,i}} \right) - \tilde{\mu}_{t,i} \right| \leq g_{t,\delta} f_t \frac{(\hat{\mathbf{Z}}_{t,i})^{1/2}}{n_{t,i}} \right\}, \qquad (54)$$

where $g_{t,\delta} = \left( 2 \log \left( 2dt \log(t)^2 \right) + \log(1/\delta) \right)^{1/2}$.

Conditionally to $\mathcal{F}_t = \sigma(A_1, Y_1, \ldots, A_t, Y_t)$, for all $i \in [d]$

$$\tilde{\mu}_{t,i} \sim \mathcal{N} \left( \hat{\mu}_{t,i} + (1 + g_{t,\delta}) f_{t,\delta} \frac{\hat{\mathbf{Z}}_{t,(i,i)}^{1/2}}{n_{t,(i,i)}}, \; f_{t,\delta}^2 \frac{\hat{\mathbf{Z}}_{t,(i,i)}}{n_{t,(i,i)}^2} \right).$$

Let $i \in a^*$. Then Gaussian concentration yields

$$\mathbb{P}_{\mathcal{F}_t} \left( \left| \left( \hat{\mu}_{t,i} + (1 + g_{t,\delta}) f_{t,\delta} \frac{\hat{\mathbf{Z}}_{t,(i,i)}^{1/2}}{n_{t,(i,i)}} \right) - \tilde{\mu}_{t,i} \right| > \sqrt{2 \log(2dt \log(t)^2/\delta)} f_{t,\delta} \frac{\hat{\mathbf{Z}}_{t,(i,i)}^{1/2}}{n_{t,(i,i)}} \right) \leq \frac{\delta}{dt \log(t)^2},$$

and

$$\mathbb{P} \left( \left| \left( \hat{\mu}_{t,i} + (1 + g_{t,\delta}) f_{t,\delta} \frac{\hat{\mathbf{Z}}_{t,(i,i)}^{1/2}}{n_{t,(i,i)}} \right) - \tilde{\mu}_{t,i} \right| > \sqrt{2 \log(2dt \log(t)^2/\delta)} f_{t,\delta} \frac{\hat{\mathbf{Z}}_{t,(i,i)}^{1/2}}{n_{t,(i,i)}} \right) \leq \frac{\delta}{dt \log(t)^2}.$$

by integration.

A union bound on $i \in [d]$ and $t \geq d(d+1)/2$ yields the result

$$\sum_{t=d(d+1)/2}^{T} \mathbb{P}(\mathcal{H}_t^c) \leq \sum_{t \in [T]} \frac{\delta}{t(\log(t)^2}.$$

$\square$

## F.2 Proof for Proposition 4

**Proposition 4.** *Let $\delta > 0$. Then* COS-V *yields*

$$\mathbb{E} \left[ \sum_{t=d(d+1)}^{T-1} \Delta_{A_{t+1}} \mathbb{1}\{\mathcal{G}_t \cap \mathcal{C}_t \cap \mathcal{H}_t\} \right] = O \left( \log(T)^3 \log(m)^2 \left( \sum_{i=1}^{d} \frac{m\mathbf{\Sigma}_{i,i}}{\Delta_{i,\min}} \right) \right).$$

*Proof.* Let $\delta > 0$. We first make use of the following Lemma.

**Lemma 8.** *Let $t \geq d(d+1)/2$, $\delta > 0$. Then for* COS-V, *under $\{\mathcal{G}_t \cap \mathcal{C}_t \cap \mathcal{H}_t\}$,*

$$\frac{\Delta_{A_{t+1}}}{f_{T,\delta}^2 g_{T,\delta}^2} \leq \sum_{i \in A_{t+1}} \frac{40m\mathbf{\Sigma}_{i,i}}{n_{t,(i,i)}} + \sum_{i \in A_{t+1}} \frac{29mdh_{t,\delta}^2 \|B\|_\infty^2}{n_{t,(i,i)}^2} \qquad (55)$$

$$+ \sum_{i \in A_{t+1}} \frac{45mh_{t,\delta} \|B\|_\infty^2}{n_{t,(i,i)}^{3/2}} + \sum_{i \in A_{t+1}} \frac{9m\|B\|_\infty^2}{n_{t,(i,i)}^3}. \qquad (56)$$

This enables to use a "modified" version of Proposition 6, which do not consider covariances.

**Proposition 7.** *Let $r \in \mathbb{N}$, $e \in (1, +\infty)^r$. Let $(\mathcal{E}_t)_{t \geq d(d+1)/2}$ be a sequence of events such that for all $t \geq d(d+1)/2$, under $\mathcal{E}_t$,*

$$\frac{\Delta_{A_{t+1}}^2}{C} \leq \sum_{i \in A_{t+1}} \frac{C_i}{n_{t,(i,j)}} + \sum_{s \in [r]} \left[ \sum_{i \in A_{t+1}} \frac{C_s}{n_{t,(i,j)}^{e_s}} \right] \qquad (57)$$

*where $C$ and $(C_s)_{s \in [r]}$ are problem-dependent positive constants. $C_i$ is a positive constant depending on $i$ so that, $C_i \leq 2m\mathbf{\Sigma}_{i,i}$. Let $c \in \mathbb{R}_+^*$ and $(c_s)_{s \in [r]} \in (\mathbb{R}_+^*)^r$ be positive constants such that $1/c + \sum_{s \in [r]} 1/c_s = 1$.*

*Then,*

$$\sum_{t=d(d+1)/2}^{T-1} \Delta_{A_{t+1}} \mathbb{1}\{\mathcal{E}_t\}$$

$$\leq 96 c_1 C \log(m)^2 \sum_{i \in [d]} \left( \frac{C_i}{\Delta_{i,\min}} \right)$$

$$+ \sum_{s=1}^{r} \left[ \mathbb{1}\{e_s = 2\} 346 \left( c_s C C_s \log(m) \right)^{1/2} m^{3/2} d \left( 1 + \log \left( \frac{\Delta_{\max}}{\Delta_{\min}} \right) \right) \right.$$

$$+ \mathbb{1}\{1 < e_s < 2\} 60.30^{1/e_s} \left( c_s C C_s \log(m) \right)^{1/e_s} d m^{1+1/e_s} \Delta_{\min}^{1-2/e_s}$$

$$+ \mathbb{1}\{2 < e_s\} 60.30^{1/e_s} \left( c_s C C_s \log(m) \right)^{1/e_s} \frac{e_s}{e_s - 2} d m^{1+1/e_s} \Delta_{\max}^{1-2/e_s} \bigg], \quad (58)$$

*where $(\alpha_k)_{k \in \mathbb{N}^*}$, $(\beta_k)_{k \in \mathbb{N}^*}$ and $k_0 \in \mathbb{N}^*$ are defined in Appendix D.1.*

Applied to `COS-V`, this yields

$$\sum_{t=d(d+1)/2}^{T-1} \Delta_{A_{t+1}} \mathbb{1}\{\mathcal{G}_t \cap \mathcal{C} \cap \mathcal{H}\}$$

$$\leq 15360 f_{T,\delta}^2 g_{T,\delta}^2 \log(m)^2 \sum_{i \in [d]} \left( \frac{m \boldsymbol{\Sigma}_{i,i}}{\Delta_{i,\min}} \right)$$

$$+ 3727 f_{T,\delta} g_{T,\delta} h_{T,\delta} (\log(m))^{1/2} \|B\|_\infty m^2 d^2 \left( 1 + \log \left( \frac{\Delta_{\max}}{\Delta_{\min}} \right) \right)$$

$$+ 7329 (f_{T,\delta} g_{T,\delta})^{4/3} h_{T,\delta}^{2/3} \log(m)^{2/3} \|B\|_\infty^{4/3} m^{5/3} d \Delta_{\min}^{-1/3}$$

$$+ 3745 (f_{T,\delta} g_{T,\delta})^{2/3} \log(m)^{1/3} \|B\|_\infty^{2/3} m^{4/3} d \Delta_{\max}^{1/3},$$

where

$$f_{t,\delta} = 6 \log(1/\delta) + 6 \left( \log(t) + (d+2) \log(\log(t)) \right) + 3d \left( 2 \log(2) + \log(1+e) \right),$$
$$h_{t,\delta} = (1 + 2 \log(1/\delta) + 2 \log(d(d+1)) + \log(1+t))^{1/2},$$
$$g_{t,\delta} = (1 + \log(2dt \log(t)^2) + \log(1/\delta))^{1/2}.$$

We deduce

$$\mathbb{E}\left[ \sum_{t=d(d+1)}^{T-1} \Delta_{A_{t+1}} \mathbb{1}\{\mathcal{G}_t \cap \mathcal{C} \cap \mathcal{H}\} \right] = O\left( \log(T)^3 \log(m)^2 \left( \sum_{i=1}^{d} \frac{m \boldsymbol{\Sigma}_{i,i}}{\Delta_{i,\min}} \right) \right).$$

□

## F.3 Proof for Lemma 8

**Lemma 8.** *Let $t \geq d(d+1)/2$, $\delta > 0$. Then for `COS-V`, under $\{\mathcal{G}_t \cap \mathcal{C}_t \cap \mathcal{H}_t\}$,*

$$\frac{\Delta_{A_{t+1}}}{f_{T,\delta}^2 g_{T,\delta}^2} \leq \sum_{i \in A_{t+1}} \frac{40 m \boldsymbol{\Sigma}_{i,i}}{n_{t,(i,i)}} + \sum_{i \in A_{t+1}} \frac{29 m d h_{t,\delta}^2 \|B\|_\infty^2}{n_{t,(i,i)}^2} \quad (55)$$

$$+ \sum_{i \in A_{t+1}} \frac{45 m h_{t,\delta} \|B\|_\infty^2}{n_{t,(i,i)}^{3/2}} + \sum_{i \in A_{t+1}} \frac{9m \|B\|_\infty^2}{n_{t,(i,i)}^3}. \quad (56)$$

*Proof.* Let $t \geq d(d+1)/2$ and $\delta > 0$. Then

$$
\begin{aligned}
\Delta_{A_{t+1}} &= \langle a^* - A_{t+1}, \mu \rangle \\
&= \langle a^*, \mu - \tilde{\mu}_t \rangle + \langle a^* - A_{t+1}, \tilde{\mu}_t \rangle + \langle A_{t+1}, \tilde{\mu}_t - \mu \rangle \\
&\leq \langle a^*, \mu - \tilde{\mu}_t \rangle + \langle A_{t+1}, \tilde{\mu}_t - \mu \rangle
\end{aligned}
$$

by definition of $A_{t+1}$.

Besides,

$$
\begin{aligned}
\langle a^*, \mu - \tilde{\mu}_t \rangle &= \sum_{i \in a^*} \mu_i - \tilde{\mu}_{t,i} \\
&= \sum_{i \in a^*} \left( \mu_i - \hat{\mu}_{t,i} + \hat{\mu}_{t,i} - \tilde{\mu}_{t,i} \right).
\end{aligned}
$$

Under $\mathcal{G}_t \cap \mathcal{C}_t$, for all $i \in a^*$,

$$
\mu_i - \hat{\mu}_{t,i} \leq f_{t,\delta} \frac{\mathbf{Z}_{t,i}^{1/2}}{n_{t,(i,i)}}.
$$

Under $\mathcal{H}_t$, for all $i \in a^*$,

$$
\begin{aligned}
\hat{\mu}_{t,i} + (1 + g_{t,\delta}) f_{t,\delta} \frac{\hat{\mathbf{Z}}_{t,i}^{1/2}}{n_{t,(i,i)}} - \tilde{\mu}_{t,i} &\leq g_{t,\delta} f_{t,\delta} \frac{\hat{\mathbf{Z}}_{t,i}^{1/2}}{n_{t,(i,i)}} \\
\hat{\mu}_{t,i} - \tilde{\mu}_{t,i} &\leq -f_{t,\delta} \frac{\hat{\mathbf{Z}}_{t,i}^{1/2}}{n_{t,(i,i)}}.
\end{aligned}
$$

Therefore, under $\{\mathcal{G}_t \cap \mathcal{H}_t \cap \mathcal{C}_t\}$,

$$
0 \leq \Delta_{A_{t+1}} \leq \langle A_{t+1}, \tilde{\mu}_t - \mu \rangle.
$$

We now develop the expression

$$
\begin{aligned}
\Delta_{A_{t+1}}^2 &\leq \left( \langle A_{t+1}, \tilde{\mu}_t - \mu \rangle \right)^2 \\
&\leq 2 \left( \langle A_{t+1}, \tilde{\mu}_t - \hat{\mu}_t \rangle \right)^2 + 2 \left( \langle A_{t+1}, \hat{\mu}_t - \mu \rangle \right)^2 \\
&\leq 2 \left( \sum_{i \in A_{t+1}} \tilde{\mu}_{t,i} - \hat{\mu}_{t,i} \right)^2 + 2 f_{t,\delta}^2 \| \mathbf{N}_t^{-1} A_{t+1} \|_{\mathbf{Z}_t}^2 \\
&\leq 2m \sum_{i \in A_{t+1}} \left( \tilde{\mu}_{t,i} - \hat{\mu}_{t,i} \right)^2 + 2 f_{t,\delta}^2 \| \mathbf{N}_t^{-1} A_{t+1} \|_{\mathbf{Z}_t}^2 \\
&\leq 2m(1 + 2g_{t,\delta})^2 f_{t,\delta}^2 \sum_{i \in A_{t+1}} \frac{\hat{\mathbf{Z}}_{t,(i,i)}}{n_{t,(i,i)}^2} + 2 f_{t,\delta}^2 \| \mathbf{N}_t^{-1} A_{t+1} \|_{\mathbf{Z}_t}^2 \\
&\leq 18m g_{t,\delta}^2 f_{t,\delta}^2 \sum_{i \in A_{t+1}} \frac{\hat{\mathbf{Z}}_{t,(i,i)}}{n_{t,(i,i)}^2} + 2 f_{t,\delta}^2 \| \mathbf{N}_t^{-1} A_{t+1} \|_{\mathbf{Z}_t}^2.
\end{aligned}
$$

As

$$\sum_{i \in A_{t+1}} \frac{\hat{\mathbf{Z}}_{t,(i,i)}}{n_{t,(i,i)}^2} = 2 \sum_{i \in A_{t+1}} \frac{\hat{\mathbf{\Sigma}}_{i,i}}{n_{t,(i,i)}} + \sum_{i \in A_{t+1}} \frac{d\|B\|_\infty^2}{n_{t,(i,i)}^2}$$

$$\leq 2 \sum_{i \in A_{t+1}} \frac{\mathbf{\Sigma}_{i,i}}{n_{t,(i,i)}} + \sum_{i \in A_{t+1}} \frac{d\|B\|_\infty^2}{n_{t,(i,i)}^2} + \sum_{i \in A_{t+1}} \frac{\frac{5}{2}h_{t,\delta}\|B\|_\infty^2}{n_{t,(i,i)}^{3/2}}$$

$$+ \sum_{i \in A_{t+1}} \frac{h_{t,\delta}^2\|B\|_\infty^2/2}{n_{t,(i,i)}^2} + \sum_{i \in A_{t+1}} \frac{\|B\|_\infty^2/2}{n_{t,(i,i)}^3}$$

$$= \sum_{i \in A_{t+1}} \frac{2\mathbf{\Sigma}_{i,i}}{n_{t,(i,i)}} + \sum_{i \in A_{t+1}} \frac{(d + h_{t,\delta}^2/2)\|B\|_\infty^2}{n_{t,(i,i)}^2}$$

$$+ \sum_{i \in A_{t+1}} \frac{\frac{5}{2}h_{t,\delta}\|B\|_\infty^2}{n_{t,(i,i)}^{3/2}} + \sum_{i \in A_{t+1}} \frac{\|B\|_\infty^2/2}{n_{t,(i,i)}^3},$$

and

$$\|\mathbf{N}_t^{-1} A_{t+1}\|_{\mathbf{Z}_t}^2 = \sum_{(i,j) \in A_{t+1}} \frac{n_{t,(i,j)}\mathbf{\Sigma}_{i,j}}{n_{t,(i,i)}n_{t,(j,j)}} + \sum_{i \in A_{t+1}} \frac{n_{t(i,i)}\mathbf{\Sigma}_{i,i}}{n_{t,(i,i)}^2} + \sum_{i \in A_{t+1}} \frac{\|B\|^2}{n_{t,(i,i)}^2}$$

$$\leq \sum_{(i,j) \in A_{t+1}} \frac{n_{t,(i,j)}\sqrt{\mathbf{\Sigma}_{i,i}}\sqrt{\mathbf{\Sigma}_{j,j}}}{n_{t,(i,i)}n_{t,(j,j)}} + \sum_{i \in A_{t+1}} \frac{\mathbf{\Sigma}_{i,i}}{n_{t,(i,i)}} + \sum_{i \in A_{t+1}} \frac{d\|B\|_\infty^2}{n_{t,(i,i)}^2}$$

$$\leq \sum_{(i,j) \in A_{t+1}} \frac{n_{t,(i,j)}(\mathbf{\Sigma}_{i,i} + \mathbf{\Sigma}_{j,j})/2}{n_{t,(i,i)}n_{t,(j,j)}} + \sum_{i \in A_{t+1}} \frac{\mathbf{\Sigma}_{i,i}}{n_{t,(i,i)}} + \sum_{i \in A_{t+1}} \frac{d\|B\|_\infty^2}{n_{t,(i,i)}^2}$$

$$\leq \sum_{i \in A_{t+1}} \frac{(m+1)\mathbf{\Sigma}_{i,i}}{n_{t,(i,i)}} + \sum_{i \in A_{t+1}} \frac{d\|B\|_\infty^2}{n_{t,(i,i)}^2}.$$

Therefore,

$$\Delta_{A_{t+1}}^2 \leq \sum_{i \in A_{t+1}} \frac{36mg_{t,\delta}^2 f_{t,\delta}^2 \mathbf{\Sigma}_{i,i}}{n_{t,(i,i)}} + \sum_{i \in A_{t+1}} \frac{9mg_{t,\delta}^2 f_{t,\delta}^2 (2d + h_{t,\delta}^2)\|B\|_\infty^2}{n_{t,(i,i)}^2}$$

$$+ \sum_{i \in A_{t+1}} \frac{45g_{t,\delta}^2 f_{t,\delta}^2 h_{t,\delta}\|B\|_\infty^2}{n_{t,(i,i)}^{3/2}} + \sum_{i \in A_{t+1}} \frac{9mg_{t,\delta}^2 f_{t,\delta}^2\|B\|_\infty^2}{n_{t,(i,i)}^3}$$

$$+ \sum_{i \in A_{t+1}} \frac{4mf_{t,\delta}^2 \mathbf{\Sigma}_{i,i}}{n_{t,(i,i)}} + \sum_{i \in A_{t+1}} \frac{2f_{t,\delta}^2 d\|B\|_\infty^2}{n_{t,(i,i)}^2}$$

$$\leq \sum_{i \in A_{t+1}} \frac{4mf_{t,\delta}^2 \mathbf{\Sigma}_{i,i}(9g_{t,\delta}^2 + 1)}{n_{t,(i,i)}} + \sum_{i \in A_{t+1}} \frac{f_{t,\delta}^2\|B\|_\infty^2 \left(27mdg_{t,\delta}^2 h_{t,\delta}^2 + 2d\right)}{n_{t,(i,i)}^2}$$

$$+ \sum_{i \in A_{t+1}} \frac{45g_{t,\delta}^2 f_{t,\delta}^2 h_{t,\delta}\|B\|_\infty^2}{n_{t,(i,i)}^{3/2}} + \sum_{i \in A_{t+1}} \frac{9mg_{t,\delta}^2 f_{t,\delta}^2\|B\|_\infty^2}{n_{t,(i,i)}^3}$$

$$\leq \sum_{i \in A_{t+1}} \frac{40mf_{t,\delta}^2 g_{t,\delta}^2 \mathbf{\Sigma}_{i,i})}{n_{t,(i,i)}} + \sum_{i \in A_{t+1}} \frac{29mdf_{t,\delta}^2\|B\|_\infty^2 g_{t,\delta}^2 h_{t,\delta}^2}{n_{t,(i,i)}^2}$$

$$+ \sum_{i \in A_{t+1}} \frac{45g_{t,\delta}^2 f_{t,\delta}^2 h_{t,\delta}\|B\|_\infty^2}{n_{t,(i,i)}^{3/2}} + \sum_{i \in A_{t+1}} \frac{9mg_{t,\delta}^2 f_{t,\delta}^2\|B\|_\infty^2}{n_{t,(i,i)}^3}.$$

This finally yields

$$\frac{\Delta_{A_{t+1}}}{f_{T,\delta}^2 g_{T,\delta}^2} \leq \sum_{i \in A_{t+1}} \frac{40m\Sigma_{i,i}}{n_{t,(i,i)}} + \sum_{i \in A_{t+1}} \frac{29mdh_{t,\delta}^2\|B\|_\infty^2}{n_{t,(i,i)}^2}$$

$$+ \sum_{i \in A_{t+1}} \frac{45mh_{t,\delta}\|B\|_\infty^2}{n_{t,(i,i)}^{3/2}} + \sum_{i \in A_{t+1}} \frac{9m\|B\|_\infty^2}{n_{t,(i,i)}^3}.$$

$\square$

# G   Experimental results

This section outlines some experimental results.

## G.1   Theoretical regret upper bound

In this experiment, the objective is to show the effect of the smallest suboptimality gap $\Delta_{\min}$ over theoretical gap-dependent regret upper bounds for ESCB-C and OLS-UCB-C. To that end, we sampled 100 environments with different $\Delta_{\min}$, with a constant number of items $d = 20$, a horizon of $T = 10^5$ rounds, and randomly sampled structures. We represent theoretical upper bounds with respect to $1/\Delta_{\min}$ in Fig. 1.

For readability reasoning, we have rescaled and reweighted the different components of the sums so that the leading term in the upper-bounds ($1/\Delta_{\min}$ or $1/\Delta_{\min}^2$ for ESCB-C or OLS-UCB-C) is greater/smaller than the rest, in a significant number of cases. In particular, all the theoretical upper bounds have the form

$$R_T \leq \frac{C}{\Delta_{\min}} + \frac{C'}{\Delta_{\min}^2} + C_r \text{Rest},$$

where $C$, $C'$ and $C_r$ are the tuned constants.

For OLS-UCB-C,

$$\text{Rest} = \Delta_{\max}(d(d+1)/2)$$

$$+ \|B\|_\infty f_{T,\delta}(4d + h_{t,\delta}^2)^{1/2} \log(m)^{1/2}\left(1 + \log\left(\frac{\Delta_{\max}}{\Delta_{\min}}\right)\right)$$

$$+ \|B\|_\infty^{4/3} f_{T,\delta}^{4/3} h_{t,\delta}^{2/3} \log(m)^{2/3} d^2 m^{2/3} \Delta_{\min}^{-1/3}$$

$$+ \|B\|_\infty^{2/3} f_{T,\delta}^{2/3} \log(m)^{1/3} d^2 m^{2/3} \Delta_{\max}^{1/3}.$$

For ESCB-C,

$$\text{Rest} = \Delta_{\max}(d(d+1)/2)$$

$$+ \log(T)\log(m)^2 \sum_{i \in [d]} \frac{\max_{a \in \mathcal{A}/i \in a} \bar{\sigma}_{a,i}^2}{\Delta_{i_{\min}}} + \log(T)\log(m) \sum_{i,j} \log\left(\frac{\Delta_{(i,j),\max}}{\Delta_{(i,j),\min}}\right)$$

$$+ \log(T)\log(m) \sum_{i} \log\left(\frac{\Delta_{i,\max}}{\Delta_{i,\min}}\right) + \log(T)\log(m) \sum_{i,j} \Delta_{(i,j),\min}^{-1/3}.$$

When the minimal gap is too small (right part of Fig. 1), both upper-bounds are of the magnitude of either $1/\Delta_{\min}^2$ or $1/\Delta_{\min}$ (depending on the algorithm). In this case, the theoretical regret bound of OLS-UCB-C outperforms the one of ESCB-C (green dots vs. blue dots). On the other side, when the gap is big enough, the remaining terms have more impact. In this case, ESCB-C has a better theoretical guarantee (orange dots vs. red dots).

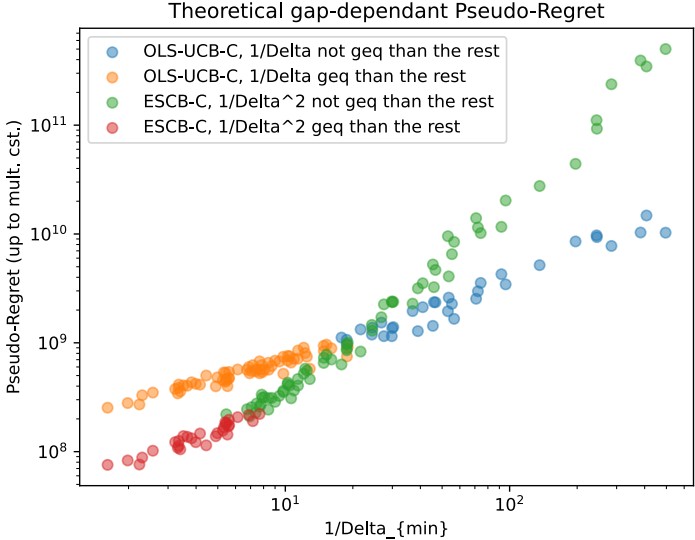

Figure 1: Evolution of regret upper bounds.

## G.2 Comparison between `ESCB-C` and `OLS-UCB-C`

We evaluate `ESCB-C` (approximated as proposed in Perrault et al., 2020b) and `OLS-UCB-C` on $d = 5$ items, $P = 10$ actions, $T = 10^5$ rounds and randomly sampled structures.

We represent the pseudo-regret evolutions in Fig. 2. The evolutions remain the same until $10^3$ rounds. After that, `ESCB-C` seemingly performs better than `OLS-UCB-C` which has a supplementary $\log(t)$ factor and is more conservative. However, just before $10^5$ rounds, we can observe a slight regime change for `ESCB-C` while the pseudo-regret of `OLS-UCB-C` continues to increase smoothly. The average regret of ESCB-C seems to have an inflexion point upward to meet the q75 curve.

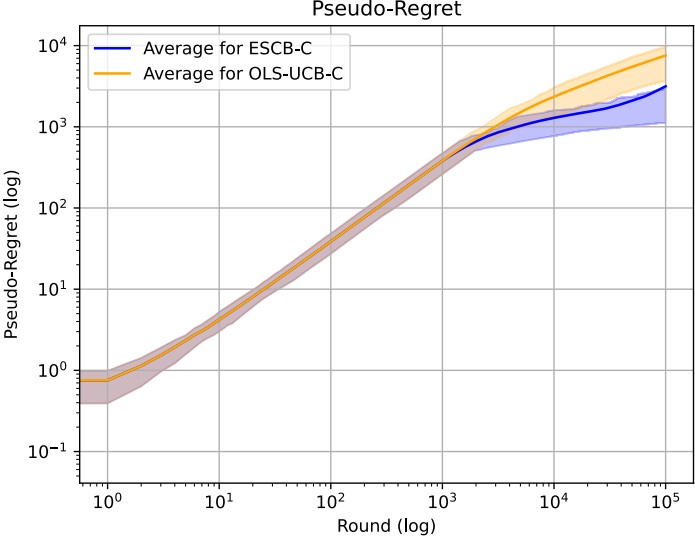

Figure 2: Pseudo-regret for `ESCB-C` and `OLS-UCB-C` for randomly sampled environments (with q25 and q75 confidence intervals).

When observing the final regret with respect to $1/\Delta_{\min}$ in Fig. 3, overall ESCB-C seems to outperform OLS-UCB-C except on some corner cases. Those cases skew the distribution for ESCB-C. Especially, for the case with the smallest suboptimality gap (the rightmost part of the figure), OLS-UCB-C outperforms ESCB-C.

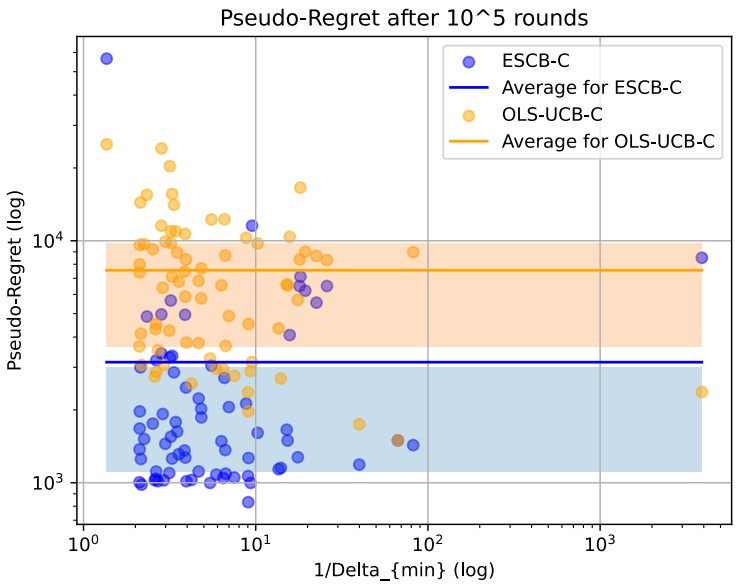

Figure 3: Pseudo-Regret with respect to $1/\Delta_{\min}$.

The evolution of the pseudo-regret in this case with the smallest suboptimality gap is presented in Fig. 4. While ESCB-C seems to fare better in the beginning, we actually see a sharp increase in its pseudo-regret before $10^5$ rounds. It could have been caused by the computational approximation of ESCB-C (described in Perrault et al. (2020b)), and/or it could be the impact of the $1/\Delta_{\min}^2$ term.

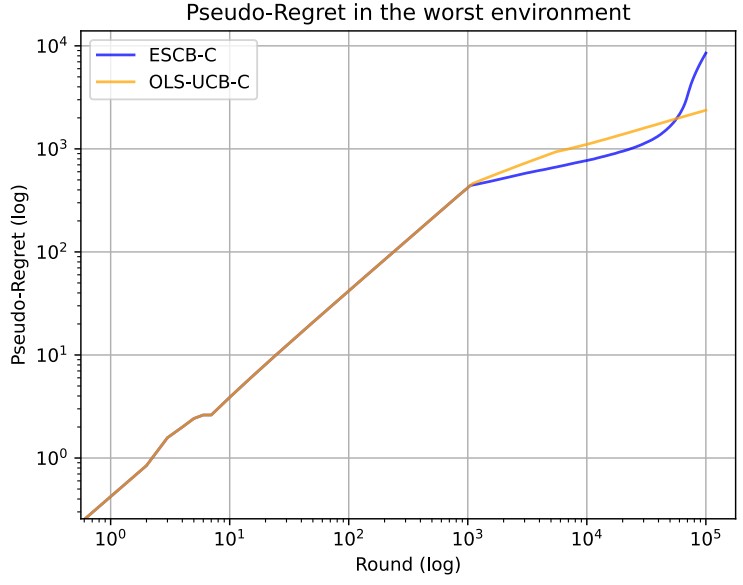

Figure 4: Pseudo-Regret in the "worst" environment.

