# OpenReview forum: "Towards Efficient and Optimal Covariance-Adaptive Algorithms for Combinatorial Semi-Bandits"
_NeurIPS.cc/2024/Conference — NeurIPS 2024 poster_

### Official Review · Reviewer_RAfE · 2024-06-20

**Soundness:** 4
**Presentation:** 3
**Contribution:** 3
**Rating:** 6
**Confidence:** 5

**Summary:**

The paper focuses on stochastic combinatorial semi-bandit problems where a player selects from a power set of actions comprising subsets of d base items. It underscores the importance of adapting to the problem structure to achieve optimal regret bounds, emphasizing the use of covariance matrix estimation to enhance performance. The proposed "optimistic" algorithms, OLS-UCB-C and COS-V, employ online covariance estimation. While COS-V prioritizes improved computational efficiency over slight optimality, inspired by Thompson Sampling, it marks the first sampling-based algorithm to achieve a $T^.5$ gap-free regret bound.

**Strengths:**

The paper is well-written.

The worst-case guarantees and instance-dependent guarantees represent a significant enhancement over existing algorithms that adapt to covariance.

2 different and relevant algorithms and analysis are proposed.

The comparison of regret rates and computational complexity between different algorithms is clear.

**Weaknesses:**

The log(T) term has a power 3 in the regret bound, which can be improved.

The approach of estimating the covariance matrix online and using it for confidence bounds is not new.

No empirical evaluation.

**Questions:**

Do you think the log(T)^3 factor can be reduced ? Why is there such a factor, whereas usually there is simply a log(T) factor ?

**Limitations:**

A sharp dependence on the horizon $T$ is more important for the instance-dependent guarantees.

---

> ### Author Rebuttal · Authors · 2024-08-06
>
> ___Concerning dependance in $T$___
>
> > The log(T) term has a power 3 in the regret bound, which can be improved.
> > Do you think the $\log(T)^3$ factor can be reduced ? Why is there such a factor, whereas usually there is simply a $log(T)$ factor ?
> > A sharp dependence on the horizon $T$ is more important for the instance-dependent guarantees.
>
> The $\log(T)$, actually has a power $2$ for OLS-UCB-C and $3$ for COS-V.  The two first exponents come from the online nature of the problem (Laplace trick) and the covariance estimations. The last one, only present in COS-V, comes from the union bounds made on the sampling distribution.
>
> ESCB-C manages a $\log(T)$ with power $1$, but suffers from more computational complexity than OLS-UCB-C and does not satisfy a $\tilde{O}(\sqrt{T})$ gap-free regret because it incurs a $1/\Delta_{min}^2$ in the gap-dependent bound. For randomized algorithms, there exists $\log(T)$ gap-dependent bounds for Combinatorial Thompson Sampling. However, they also suffer from additional terms that can be up to $1/\Delta^d$, which prevent the $\tilde{O}(\sqrt{T})$ gap-free bound.
>
> All in all, it is indeed possible to get better $\log(T)$ exponents for the gap-dependent bounds. However, known approaches do not satisfy a $\tilde{O}(\sqrt{T})$ gap-free bound at the same time. It means that on “difficult" instances, our algorithms will incur $\tilde{O}(\sqrt{T})$ regret, while others may suffer regrets that are a lot worse (Imagine $\Delta \sim 1/T$ for example).
>
> We are not aware of any straightforward way to reduce the $\log(T)$ exponents and keep the desired gap-free bounds at the same time.
>
> ___
> ___Concerning the covariance matrix___
> > The approach of estimating the covariance matrix online and using it for confidence bounds is not new.
>
> The approach is indeed not new (and it is even natural) and we do not claim to be the first. But the way to incorporate those estimators in our analysis is rather new.

---

> ### Comment · Reviewer_RAfE · 2024-08-13
>
> I read the authors’ rebuttal and the comments of other reviewers. I would like to keep my score unchanged.

---

### Official Review · Reviewer_47Sr · 2024-07-04

**Soundness:** 3
**Presentation:** 3
**Contribution:** 3
**Rating:** 6
**Confidence:** 3

**Summary:**

The paper "Towards Efficient and Optimal Covariance-Adaptive Algorithms for Combinatorial Semi-Bandits" addresses the challenge of designing algorithms that adapt to covariance in the context of stochastic combinatorial semi-bandits, where decision-makers face a set of actions exponentially large in the number of base items.

**Strengths:**

1. The paper presents new algorithms that improve on both the computational efficiency and regret bounds by adapting to the covariance structure of the environment, which is crucial for applications where the rewards are not independent across actions.

2. It provides the first gap-free regret bounds (regret bounds that do not depend on the minimum sub-optimality gap between the best and second-best actions) for these types of problems, extending the theory beyond independent rewards assumptions.

**Weaknesses:**

1. The assumption that the reward (Lines 45-46) for each base item \( Y_{t,i} \) is bounded by \( \frac{B_i}{2} \) may be strong, where reward distributions can exhibit significant variability and are not tightly bounded.

2. While the algorithms' theoretical framework and mathematical formulation are well-articulated, the lack of experimental results or simulations leaves a crucial gap.

**Questions:**

1. Concerns regarding the role of `f_t,δ` and `g_t,δ` functions in Eq. 6 and 7.
The algorithms utilize the functions `f_t,δ` and `g_t,δ` to dynamically adjust the exploration strategy. The paper implies an increase in `f_t,δ` and `g_t,δ` as \( t \) increases, which leads to an increasing exploration bonus and variance over time. This design choice raises several concerns and questions, especially from an intuitive standpoint:

     a) Increasing Exploration Over Time: Typically, one might expect that as an algorithm learns more about the environment, the need for exploration would decrease, reducing the exploration bonus and variance. The paper’s approach where `f_t,δ` increases with \( t \) seems counterintuitive. An explanation or justification for why increased exploration is necessary or beneficial as the algorithm progresses would be crucial.

     b) Design Justification for `g_t,δ` and `f_t,δ`: The paper lacks a detailed explanation for the design of `g_t,δ` and `f_t,δ`. Are there specific characteristics of the problem domain or empirical observations that suggest this approach?

   c) Impact on Algorithm’s Performance: How does the increasing trend of `f_t,δ` and `g_t,δ` impact the algorithm's overall performance across different learning stages? Some insights into how this approach compares to more traditional methods where exploration decreases over time would be beneficial.

2. The paper specifies using a Gaussian distribution in Eq. 7 for parameter sampling within the COS-V algorithm. Could the authors provide further justification for the choice of a Gaussian distribution in this context?

**Limitations:**

As described in the paper, the results and theorems are discussed with comments made especially about some limitations.

---

> ### Author Rebuttal · Authors · 2024-08-06
>
> ___Concerning the weaknesses___
>
> > The assumption that the reward (Lines 45-46) for each base item ($Y_{t,i}$) is bounded by ($\frac{B_i}{2}$)  may be strong, where reward distributions can exhibit significant variability and are not tightly bounded.
>
> In many realistic settings, a player would be able to tell what scale of values for $Y_{t, i}$ is normal or not. If they can identify an upper bound $B$, then they could use the algorithms with $min(max(Y_{t, i}, -B), B)$ instead of the "true" $Y_{t, i}$ observed.
>
> Otherwise, one could try a Doubling Trick. We initialise $B$ at a certain power of $2$, and then we restart the algorithm each time an observation surpasses it, after doubling the current value of $B$. This would add a multiplicative factor to the regret bounds, of magnitude $\log(B)$, where $B^*$ is a “true" upper bound.
>
> ___
> ___Concerning the main questions___
>
> > Concerns regarding the role of $f_{t,\delta}$ and $g_{t,\delta}$ functions in Eq. 6 and 7. The algorithms utilize the functions $f_{t,\delta}$ and $g_{t,\delta}$ to dynamically adjust the exploration strategy. The paper implies an increase in $f_{t,\delta}$ and $g_{t,\delta}$ as $t$ increases, which leads to an increasing exploration bonus and variance over time. [...]
> >> a) Increasing Exploration Over Time [...].
> b) Design Justification for $g_{t,\delta}$ and $f_{t,\delta}$ [...].
> c) Impact on Algorithm’s Performance [...].
>
> We acknowledge the fact that the $\log(t)$ increase can be counterintuitive and may add an explanation in subsequent versions to clarify it.
>
> The exploration is actually not increasing over time. The decay on the exploration bonus, comes more from the number of times each item/couple has been observed $(n_{i,j})$ rather than from $\log(t)$ (See Eq.4 and Eq.6 for example). Like for all the main variants of UCB, the exploration bonuses look like a sum of $\sqrt{\frac{\log(\delta)}{n_{i,j}}}$, which decreases as the number of time each item is chosen increases.
>
> Using $\log(t)$ (instead of $\log(T)$) ensures adaptivity to the time horizon and has been commonly used in several other papers.
>
> The justification for $f_{t, \delta}$ is explained in Section 4: it originates from the way the ellipsoids are designed. $f_{t, \delta}$ is involved in the definition of event $G_t$ in Eq.9 and its expression ensures Prop.1 is satisfied. $g_{t, \delta}$ controls the exploration made by COS-V. Its expression is derived from union bounds which enable Lemma 5 to be satisfied.
>
> > The paper specifies using a Gaussian distribution in Eq. 7 for parameter sampling within the COS-V algorithm. Could the authors provide further justification for the choice of a Gaussian distribution in this context?
>
> We think that other distributions could actually be used.
>
> Using Gaussian distributions was a natural direction as they are standard distributions, we know how to sample them efficiently, especially for given covariance matrices, and we know how to control their tails with usual tools.

---

> ### Comment · Reviewer_47Sr · 2024-08-10
>
> Thank you for the response from the authors.
> 1. I have reviewed the experiments conducted by the authors, and I am particularly curious about the regret pattern of OLS_UCBV, which shows a distinct zigzag shape: it starts with a slow increase in regret, followed by a sharp rise, and then it tends to converge. This has also piqued my curiosity about the specific settings of the randomly generated synthetic environment, and it appears that its results are inferior to those of C_UCBV and ESCB_C_approx.
>
> 2. Additionally, if it is Covariance-Adaptive, I would like to know how the fluctuations or error bars of the actual performance of the algorithm proposed by the authors compare to those of other algorithms across multiple random seeds.

---

> > ### Author Response · Authors · 2024-08-12
> >
> > We thank the reviewer for responding to our rebuttal and hope that we have adequately addressed their concerns regarding the design of our algorithms and our results. We kindly ask the reviewer to take this into consideration in their final assessment.
> >
> > We would like to emphasize again that our contributions are focused on the theoretical aspects of combinatorial bandits. Our paper is already dense with significant theoretical contributions, including new gap-free regret bounds, improved computational complexity, and a novel stochastic algorithm with analysis that combines the computational efficiency of Thompson Sampling with the analytical strengths of optimistic algorithms, all of which contribute to a deeper understanding of combinatorial bandits.
> >
> > The empirical study we provided in the rebuttal primarily serves to demonstrate that our algorithms can be implemented, achieve the expected logarithmic regret expected by the theory, and perform well in practice compared to their competitors. We acknowledge that this study alone is not sufficient, and we will strive to conduct more thorough experiments in the final version (although we do not expect it to be considered as a contribution of our current work).
> >
> > Regarding the additional questions, due to time constraints during the rebuttal phase and the high computational cost of existing combinatorial bandit algorithms, our experiments were conducted on a single synthetic dataset and should be interpreted with caution.
> >
> > > I have reviewed the experiments [...].
> >
> > Note that due to the log-log scale, the early shape of the regret curves should be interpreted with caution, as they correspond to only a few rounds $(\approx 30)$ compared to the total number of rounds $(\approx 10\,000)$. Furthermore, the curves are only valid after the measurement at $t=10$; prior to that, they mistakenly appear flat. The early behavior is primarily influenced by how the algorithms perform initial exploration and may be affected by hyper-parameter calibrations, which were not thoroughly optimized in this case.
> >
> > What is more important to observe in these experiments is the behavior at the end of the regret curves:
> >
> > - Both OLS-UCB-C and ESCB-C-approx appear to have converged and entered the logarithmic regret regime. The difference between them seems to be due to constant terms related to their exploration strategies, but it does not seem to increase over time.
> > - Towards the end, the per-round regret of C-UCBV is higher than that of OLS-UCBV, indicating that its cumulative regret will eventually be higher. This synthetic instance was "easy" enough for both algorithms to reach the logarithmic regime. However, it's important to note that in more challenging instances, ESCB-C only guarantees a worst-case regret of $O(T^{2/3})$, while our algorithm provides a stronger guarantee of $O(\sqrt{T})$.
> >
> > > Covariance-Adaptation and error bars.
> >
> > Indeed, having error bars is important, and we will include them in the final version. Unfortunately, we did not have time to include them in this preliminary experiment presented in the rebuttal. The adaptivity of our algorithms (and ESCB-C) to the covariance structure of the base items can be inferred from the per-round regret (i.e., the derivative) at the end of the plot. We observe that ESCB-C and OLS-UCB-C maintain a relatively similar distance asymptotically. In contrast, C-UCBV (C-UCB with plugged-in variance estimators) has also converged, but its final per-round regret is slightly worse than both. This is likely because C-UCBV does not account for covariance but instead assumes worst-case correlations between items. However, we must emphasize that this could be an artifact, and these experiments are not thorough enough to draw definitive conclusions.

---

> > > ### Comment · Reviewer_47Sr · 2024-08-14
> > >
> > > Thank you for your clarification. I will increase the score by one point based on this discussion.

---

> > > > ### Author Response · Authors · 2024-08-14
> > > >
> > > > Thank you for your understanding and for raising the score; we value your careful consideration of our contributions.

---

### Official Review · Reviewer_zmBr · 2024-07-09

**Soundness:** 4
**Presentation:** 4
**Contribution:** 4
**Rating:** 7
**Confidence:** 2

**Summary:**

This paper tackles the combinatorial semi-bandits problem and provides many theoretical results, including gap-free variance-dependent upper bounds for both deterministic and stochastic sampling strategies, and for the least, they also provide a corresponding lower bound.

**Strengths:**

The paper is well-written in a reader-friendly manner. It contributes a  variety of highly non-trivial theoretical results that improve upon the current state-of-the-art, with extensive proofs in the appendix.

**Weaknesses:**

The paper does not provide any empirical evaluation, it would have been interesting to see how the presented algorithms perform empirically against its competitors. But given the theoretical contributions and nature of this paper, this is okay for me.

Apart of that I have only the following rather minor issues and typos/suggestions:
- Be more precise in the formulation of the lower bound (Thm. 2): In which sense does the inequality hold, in expectation? Also, is policy restricted deterministic policies here?
- Some of the notations may be clear for most readers, but should for the sake of completeness still be properly introduced. E.g., $\mathbb{N}^\ast$, $\mathbf{I}$ and which logarithm (base) $\log$ refers to.
- In Algo 2, line 3,  does it have to be $\leq 2$ instead of $\leq 1$?
- 8: yield[s]
- 10: $\mathcal{O}(\sqrt{T})$ instead of $\sqrt{T}$?
- 28: At each round
- 108: adaptative
- 139: positive semi-definite
- 269: $\mathbb{P}(\mathcal{C}^c)$
- 272: sketch of the proof
- You oftentimes write $A\bigcap B$ instead of $A\cap B$, why?

**Questions:**

- Do you have a feeling how sensible the regret is w.r.t. the vector $B$? You assume this to be known, but I could imagine that in practice it's oftentimes unknown.
- In Table 1, do you know how $C_{1/T}^{opt}$ scales in comparison to $d^2$? That is, do you know that $OLS-UCB-C$ outperforms $ESCB-C$ regarding time complexity?
- Both your algorithms have a parameter $\delta > 0$, which is not contained in the upper bounds. Could you say how the behaviour of the regrets are w.r.t. this parameter?

**Limitations:**

The authors have adequately addressed the limitations of their work.

---

> ### Author Rebuttal · Authors · 2024-08-06
>
> ___Concerning the minor questions and remarks___
>
> We first wish to thank the reviewer for pointing out our typos and inaccuracies.
>
> > Be more precise in the formulation of the lower bound (Thm. 2): In which sense does the inequality hold, in expectation? Also, is policy restricted deterministic policies here?
>
> The theorem holds in expectation, and is applicable for both deterministic and for stochastic policies.
>
> Besides, it seems that there is a mistake in our formulation. We inverted a “for all" and a “there exists" statement. We should have written: “For any policy $\pi$, there exist a stochastic combinatorial semi-bandit such that$\dots$". We are very grateful to the reviewer for pointing out this inaccuracy, which allowed us to correct a mistake.
>
> > In Algo.2, line 3, does it have to be $\leq 2$ instead of $\leq 1$ ?
>
> It should be either $\leq 1$ or $<2$. All the coefficients indexed by $(i,j)$ are correctly defined as soon as the corresponding interaction has been observed at least twice.
>
> > You oftentimes write $A\bigcap B$ instead of $A\cap B$, why?
>
> The reason is purely aesthetic, we will uniformize the notation.
>
> ___
> ___Concerning the main questions___
>
> > Do you have a feeling how sensible the regret is w.r.t. the vector $B$? You assume this to be known, but I could imagine that in practice it's oftentimes unknown.
>
> In our analysis, $B$ appears in the negligible additive terms (w.r.t. the dependence in $T$). Using a “looser" $B$ may make the convergence a little slower, but would not impact the asymptotic regret.
>
> In practice, knowing a plausible upper bound on the rewards is sufficient. In realistic settings, we can assume that a player knows such an upper bound B_i. In that case, they can replace $Y_{t, i}$ with $min(max(Y_{t, i}, -B_i), B_i)$ and use our algorithms.
>
> Otherwise, one could try a Doubling Trick. We initialise $B$ at a certain power of $2$, and then we restart the algorithm each time an observation surpasses it, after doubling the current value of $B$. This would add a multiplicative factor to the regret bounds, of magnitude $\log(B^*)$ where $B$ is a “true" upper bound.
>
> > In Table 1, do you know how $C^{opt}_{1/T}$ scales in comparison to $d^2$ ? That is, do you know that OLS-UCB-C outperforms ESCB-C regarding time complexity?
>
> $C^{opt}_{1/T}$ scales with the horizon $T$, contrarily to ou per-rounds complexities. Ultimately, our complexity will then be better for horizons large enough.
>
> > Both your algorithms have a parameter $\delta>0$, which is not contained in the upper bounds. Could you say how the behaviour of the regrets are w.r.t. this parameter?
>
> We have formulated $f_{t, \delta}$ and $g_{t, \delta}$ so as to isolate $\delta$ from the dependence on time and dimension.
>
> Its influence shows through $\log(1/\delta)$ factors in negligible additive terms. $\delta$ is fixed at the beginning and characterizes a tradeoff when sizing the ellipsoids. It influences how quick we reach the asymptotic regret rate but not the said rate.

---

> > ### Comment · Reviewer_zmBr · 2024-08-11
> >
> > Dear authors, thank your for your rebuttal, I have no further questions.

---

### Author Rebuttal · Authors · 2024-08-06

We would like to sincerely thank the reviewers, ACs, SACs and PCs for their expertise and the time they are devoting to our submission. Their feedback and suggestions are very valuable and will be taken into account.

___General comments___

We were provided 3 high quality reviews that acknowledge the strengths of our submission, namely, the improvement over existing results by providing gap-free bounds, the clear presentation and the mathematical soundness.

We answer the specific questions of each reviewer in the per-review rebuttal.

Concerns over (the lack of) empirical evaluations have been raised by several reviewers. Our contributions being theoretical, we did not consider a priority to provide such evaluations and consider our contributions self-sufficient as is. We answer the concerns in more details in the following.

___
___Concerning empirical evaluations___

Although our contributions are mainly theoretical, we agree that some empirical evaluations could be interesting. However, we consider our submission self-sufficient as is. We detailed clear and complete proofs and strove to make the presentation clear. Their quality has been unanimously recognized as a main strength of our submission.

We are considering adding an experimental part in the Appendix for a new version, but we wish to show insightful results and illustrations to match the quality of our current main document. If the reviewers have specific demands and/or suggestions, we can try to produce them as our algorithms (and some competitors) can be easily implemented (although they all still need to be tuned).

For the sake of illustration, we provide regret curves in a PDF file, for a randomly generated synthetic environment, for OLS-UCB-C, where we see that it is comparable to other combinatorial algorithms.

---

### Decision · Program_Chairs · 2024-09-25

**Decision:**

Accept (poster)

**Comment:**

All reviewers found the paper to be technically solid and the results to be interesting, and have only raised minor concerns. All of these concerns were addressed by the author rebuttals, and thus it was easy to come to the decision that the paper should be accepted to NeurIPS 2024.